# Quantifying the impacts of human water use and climate variations on recent drying of Lake Urmia basin: the value of different sets of spaceborne and in-situ data for calibrating a hydrological model

Seyed-Mohammad Hosseini-Moghari[1], Shahab Araghinejad[1], Mohammad J. Tourian[2], Kumars Ebrahimi[1], Petra Döll[3,4]

[1]Department of Irrigation and Reclamation Engineering, University of Tehran, Karaj, Iran
[2]University of Stuttgart, Institute of Geodesy (GIS), Stuttgart, Germany
[3]Institute of Physical Geography, Goethe University Frankfurt, Frankfurt am Main, Germany
[4]Senckenberg Leibnitz Biodiversity and Climate Research Centre (SBiK-F), Frankfurt am Main, Germany

*Correspondence to:* Seyed-Mohammad Hosseini-Moghari (Hosseini_sm@ut.ac.ir)

**Abstract.** During the last decades, the endorheic Lake Urmia basin in northwestern Iran has suffered from declining groundwater tables and a very strong reduction in the volume as well as recently in the extent of Lake Urmia. For the case of Lake Urmia basin, this study explores the value of different locally and globally available observation data for adjusting a global hydrological model such that it can be reliably used for distinguishing the impacts of human water use and climate variations. The WaterGAP Global Hydrology Model (WGHM) was for the first time calibrated against multiple in-situ and spaceborne data to analyse the decreasing lake water volume, lake river inflow, loss of groundwater, and total water storage in the entire basin during 2003-2013. Then the best-performing calibration variant was run with or without considering water use to quantify the impact of human water use. Observations encompass remote-sensing based time series of annual irrigated area in the basin from MODIS, monthly total water storage anomaly (TWSA) from GRACE satellites, and monthly lake volume anomalies. In-situ observations include time series of annual inflow into the lake and basin averages of groundwater level variations based on 284 wells. In addition, local estimates of sectoral water withdrawals in 2009 and return flow fractions were utilized. Four calibration variants were set up in which the number of considered observation types was increased in a stepwise fashion. The best fit to each and all observations, including the time series of lake volume not used for calibration, was achieved if the maximum amount of observations was used for calibration. Calibration against MODIS and GRACE data alone improved simulated inflow into Lake Urmia but inflow and lake volume loss were still still overestimated, while groundwater loss was understimated and seasonality of groundwater storage was shifted as compared to observations. Lake and groundwater dynamics could only be simulated well if calibration against groundwater levels led to an adjustment the fractions of human water use from groundwater and surface water. Thus, in some basins, globally available space-born observations may not suffice for improving the simulation of human water use. According to our study, human water use was the reason for 50% of the total basin water loss of about 10 km[3] during 2003-2013, for 40% of the Lake Urmia water loss of about 8 km[3] and for up to 90% of the groundwater loss. Lake inflow was 40% less than it would have been without human water use. This study proved that even without human water use Lake Urmia would not have recovered from the significant

loss of lake water volume caused by the drought year 2008. These findings can support water management in the basin and more specifically Lake Urmia restoration plans.

## 1 Introduction

Iran is a country with arid and semi-arid climate where population growth and the government's aim of food self-sufficiency has led to increasing irrigated crop production and exploitation of surface water and groundwater resources. Climate change has resulted in increased temperatures and, in particular the northwest of the country, in decreased precipitation (Tabari and Talaee, 2011a, b) and thus decreased renewable water resources. In the last decades, numerous wetlands and lakes in Iran have dried up, and groundwater levels have strongly declined in most areas (Madani et al., 2016). The most serious disaster has occurred in the Lake Urmia basin, an interior basin in the northwest of Iran located in the three provinces West Azarbaijan, East Azarbaijan, and Kurdistan that covers an area of 52,000 km$^2$ (Fig. 1). At the downstream of the basin, 17 permanent rivers and 12 seasonal rivers discharge into the largest natural water body in Iran, Lake Urmia. Over the past two decades, climate variations and human activities (Hassanzadeh et al., 2012) have decreased inflow into the lake. Precipitation in the basin shows a decreasing trend over the period 1951-2013, with particularly low values after 1995, and evaporation has increased (Alizadeh-Choobari et al., 2016). Lake water volume is now approximately $30 \cdot 10^9$ m$^3$ below its historical maximum (ULRP, 2015a).

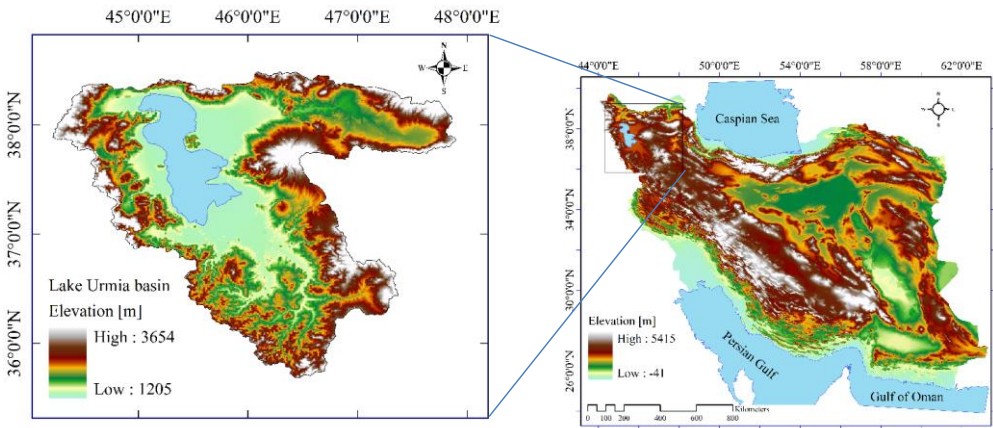

**Figure 1: Location of Lake Urmia basin.**

Lake Urmia is one of the largest hypersaline lakes in the world, which due to its ecological and natural features is a National Park, a Ramsar Site and a UNESCO Biosphere Reserve (Eimanifar and Mohebbi, 2007). It is a terminal lake that loses water only by evaporation (Hassanzadeh et al., 2012). Abbaspour and Nazaridoust (2007) estimated that inflows of at least $3 \cdot 10^9$ m$^3$/yr are needed to compensate for lake evaporation, while Alborzi et al. (2018) estimated values between $2.9 \cdot 10^9$ to $5.4 \cdot 10^9$ m$^3$/yr depending on climatic conditions. According to Alborzi et al. (2018), recovery of the lake could range from 3 to 16 years depending on climatic condition, water use reductions, and environmental releases. Inflow from groundwater to

the lake was estimated to be less than 3% of total inflow from precipitation, rivers, and groundwater (Hasemi, 2011). In the 1970s and 80s, the water table of Lake Urmia was approximately at 1,276 m above sea level and then increased to more than 1,278 m in 1995 due to a few wet years (Shadkam et al., 2016). Afterwards, the water table dropped to 1,274 m in 2003 specially because of the severe drought in 1999-2001 exacerbated by human water use (Shadkam et al., 2016). From 2003 to

2014, lake extent was approximately halved, and water level declined by another 3 m, while seasonal variability of lake water extent increased (Tourian et al., 2015) (Fig. 2). After 2015, lake extent and storage have stabilized (Fig. 3) due to the relatively high precipitation in 2015 and 2016, increased releases from reservoirs and management activities for decreasing water consumption (ULRP, 2015b).

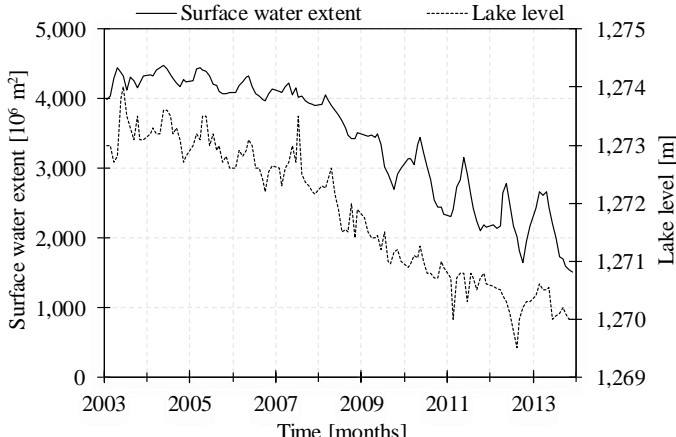

**Figure 2: Time series of surface water extent and water table elevation of Lake Urmia (data from Tourian et al., 2015).**

        Studies on various aspects of the Lake Urmia disaster abound. With decreasing lake water volume, salt concentration has increased, endangering the aquatic biota feeing birds; exposed salt layers may lead to salt storms (Pengra, 2012). Precipitation reduction, temperature increase, agricultural development including construction of man-made dams and building a causeway across the lake have been identified as the main reasons for the degradation of Lake Urmia (Abbaspour and

Nazaridoust, 2007; Zeinoddini et al., 2009; Delju et al., 2012; Jalili et al., 2012; Sima and Tajrishy, 2013; Fathian et al., 2014; Farajzadeh et al., 2014; Banihabib et al., 2015; AghaKouchak et al., 2015; Azarnivand et al., 2015; Alizadeh-Choobari et al., 2016; Ghale et al., 2018; Khazaei et al., 2019). By using Gravity Recovery And Climate Experiment (GRACE) satellite observations, altimetry data for Lake Urmia and outputs of the Global Land Data Assimilation System (GLDAS), Forootan et al. (2014) estimated the trend of groundwater storage changes in the Lake Urmia basin as -11.2 mm/yr between the years of

2005 to 2011, the largest decrease of the six investigated Iranian basins. Zarghami (2011) examined four routes to transfer the water from Aras basin in the north of Lake Urmia basin to provide an alternative for the water supply for the agricultural and drinking demands in the north of the basin. Ahmadzadeh et al. (2016) investigated the effect of irrigation system changes in the basin from the surface to pressurized systems; they found that such changes would increase water productivity but would have no effect on lake inflow and would reduce groundwater levels by 20%.

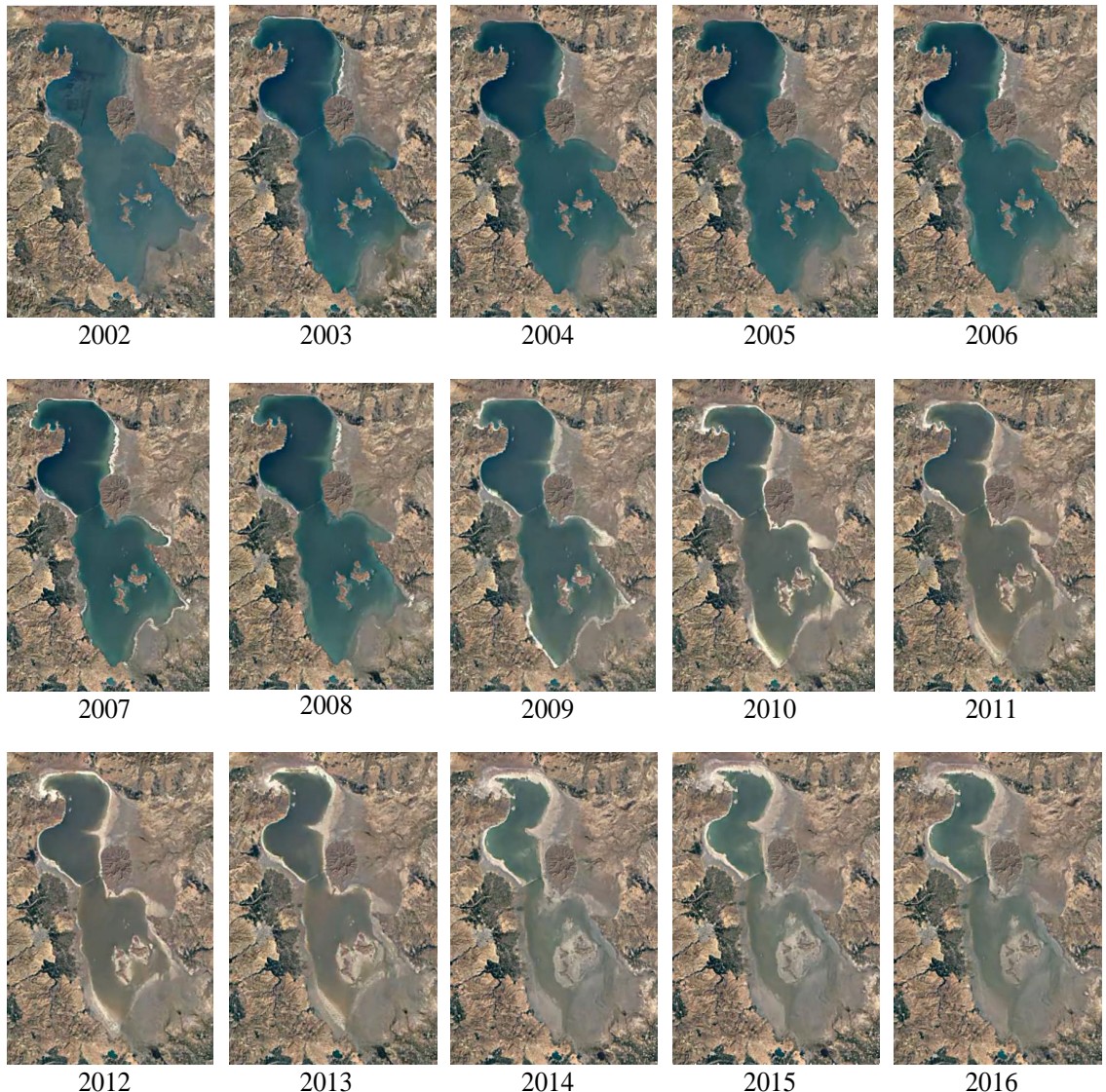

**Figure 3: Lake Urmia during the time period 2002-2016 (Google Earth Timelapse, last accessed: 28 Apr. 2018).**

Three hydrological modelling studies for Lake Urmia basin focused on quantifying the contributions of various factors on lake water volume (Hassanzadeh et al., 2012), lake inflow (Shadkam et al., 2016) or both (Chaudhari et al., 2018). Using a lumped system dynamics modelling approach and observed time series of lake water volume for model calibration, Hassanzadeh et al. (2012), determined that about 65% of lake level decline between 1997 and 2006 was due to reduced river inflow, while four major man-made reservoirs contributed 25% and diminished precipitation on the lake surface 10%. Shadkam et al. (2016) evaluated the impact of climate, irrigation with surface water and reservoirs on inflow into the lake for the period 1960-2010 using a modified version of the macro-scale gridded hydrological model Variable Infiltration Capacity (VIC)

model, which was calibrated against time series of river discharge at six observation station at the downstream end of six sub-basins draining into Lake Urmia. While the model was driven by global gridded WFDEI climate data set with a spatial resolution of 0.5°, basin-specific information on 41 reservoirs and on the temporal development of irrigated areas were taken into account. The study found that reservoirs had a very small impact on annual inflows and that climate variations accounted

for 60% of lake inflow decrease of 48% over the 50-year period. In the model, all irrigation requirements need to be fulfilled by available surface water. Therefore, reduced availability of surface water during the 2000s due to low precipitation and high temperature resulted in unfulfilled irrigated water demand and a cap on the effect of human water use in the model while in reality, groundwater abstractions occurred and even increased (Delju et al., 2012; Hesami and Amini, 2016). In addition, the modelling study of Shadkam et al. (2016) did not consider the impact of domestic and industrial water use in the basin which

can be expected to have increased during the last decades, given a population increase from 4.8 to 5.9 million from 2002 to 2010 (http://ulrp.sharif.ir/en/page/about-urmia-lake-basin, last accessed: 28 Apr. 2018). Chaudhari et al. (2018) used the output of the global HiGW-MAT model, with 1°×1° grid cell size of approx. 10,000 km$^2$, to distinguish climatic and anthropogenic contributions to the shrinkage of Urmia Lake. By running the model with and without human impacts (surface and groundwater use as well as reservoirs), they estimated that the human-induced river flow decline between 1995-2010 to account for 86% of

the observed decrease of lake volume. However, a comparison with GRACE TWSA showed that the model overestimates the decrease in TWSA in the basin between 2003 and 2010. The HiGW-MAT model was not calibrated for the Lake Urmia basin but net irrigation requirements were simulated specifically for this study based on Landsat satellite images for 5 years between 1987 and 2016. The lake water balance is not simulated by the model such that no comparison with observed lake water levels was possible. A comparison with river discharge or groundwater observations was not done either.

The aim of our study was twofold. On the one hand, we wanted to quantify, by a holistic and reliable modelling approach, the contributions of climate variations and human activities to the decrease of Lake Urmia water volume and river inflows as well as, different from previous studies, to groundwater storage and total water storage in the whole Lake Urmia basin. Such a modelling approach requires the set-up of a model that is able to simulate the impact of surface and groundwater use as well as of climate variations on these water storages and flows. The hypothesis is that if model output for all these

variables fit well to observations, then the model can be used to assess the contribution of human water use by comparing the outputs of two model variants, one with human water use and one where human water use is assumed to be zero. To achieve a good fit to observations, hydrological models need to be calibrated by comparison of observations with model output variables. While hydrological models are usually calibrated only against observations of river discharge, it is well known that a good fit of simulated and observed river discharge does not lead necessarily lead to an appropriate simulation of other flows

and storages (Beven and Freer, 2001). However, in previous hydrological modeling studies of Lake Urmia basin, model calibration was either not done at all or only using a single observation type. On the other hand, using Lake Urmia basin as a test case, we wanted to explore the value of different types of observation data for adjusting a global hydrological model by multi-observation calibration. Currently, global hydrological models are mostly uncalibrated but globally available space-born observations have increased the opportunity for model calibration at the global scale (Döll et al., 2016).

We used the state-of-the-art global hydrological model WaterGAP 2.2c (spatial resolution 0.5°×0.5°) which simulates human water uses from surface water and groundwater and how these affect river discharge, groundwater, lake water, and total water storage. In its standard version, WaterGAP is calibrated against observed mean annual river discharge at 1319 stations worldwide by adjusting 1-3 model parameters related to runoff generation and streamflow (Müller Schmied et al., 2014), but for reasons of data availability not for a station in Lake Urmia basin. A previous WaterGAP version was calibrated, for 22 large basins, against streamflow and total water storage anomalies by adjusting 6-8 parameters (Werth and Güntner, 2010). For this study on the differential impacts of climate and human water use on Lake Urmia basin, WGHM was for the first time calibrated for a specific basin by using multiple types of independent data. Multi-observation calibration included the adjustment of temporally constant model parameters as well as the adjustment of human water use input data. To understand the value of different observations or other regionally available data for understanding dynamics of water flows and storages in a basin, WGHM was calibrated sequentially by considering, in each calibration variant, an additional data type. In the first variant, only remote sensing data were used (variant RS). In-situ river discharge observations were added in variant RS_Q. In the third variant RS, discharge and groundwater level data were used (variant RS_Q_GW), and finally RS, discharge, groundwater levels as well as regional data of basin-wide total withdrawals plus estimated return flow fractions (RS_Q_GW_NA variant). Model evaluation was done by comparison of simulated lake water volume anomalies against observed anomalies. The best-performing model variant RS_Q_GW_NA was then applied to simulate the water flows and storages in Lake Urmia basin that would have occurred under naturalized conditions, i.e. without any human water use (and man-made reservoirs). By comparing the output of the naturalized run with the output of the model run with human impacts, we determined the contributions of human water use and climate variation on lake inflow and water storages in the period 2003-2013. In section 2, we describe the utilized data and the simulation setup. The results of the four calibration variants and the impacts of human water use are shown in section 3. Section 4 discusses multi-observation calibration and the analysis of human impact as well as the limitations of the study. Finally, conclusions are drawn.

## 2 Methods and data

We analyzed the 11-year period from the beginning of 2003 until the end of 2013, as both GRACE data and global climate data to drive WaterGAP where available for this period. In the following sections, WaterGAP, WaterGAP input data and observational data used for calibration as well as the calibration variants are described.

### 2.1 WaterGAP

WaterGAP is a global hydrological model for assessing water resources under the influence of humans (Döll et al., 2003; Müller Schmied et al., 2014). With a spatial resolution of 0.5°×0.5°, it simulates water abstractions and consumptive water use (so-called net abstractions, i.e. the amount of water that evapotranspirates during use and does not flow to surface water bodies and groundwater afterwards) in five sectors (irrigation, livestock, domestic, manufacturing and cooling of thermal power

plants); then net abstractions from either groundwater (NAg) or surface water bodies (NAs) are computed (Müller Schmied et al., 2014; Döll et al., 2012). Negative values of NAg occur where return flow to groundwater from irrigation with surface water is so high that water is added to groundwater storage by human water use. NA is the sum of NAg and NAs and equal to consumptive water use. Time series of NAg and NAs in each grid cells are then input to the WaterGAP Global Hydrology model WGHM that simulates their effect on water flows and storages. In WGHM, NAg and NAs are subtracted from either the groundwater or surface water bodies (lakes, reservoirs or rivers) (Müller Schmied et al., 2014).

WGHM simulates daily water storage as well as flows like evapotranspiration, groundwater recharge (Döll and Fiedler, 2008), runoff, and river discharge for all continents except Antarctica. Water is transported between grid cells according to the DDM30 drainage direction map (Döll et al., 2003). Water storage compartments encompass snow, canopy, soil, groundwater, rivers, lakes, wetlands, and man-made reservoirs (Eicker et al., 2014). Lake water storage is simulated as the difference of precipitation on the lake, evapotranspiration, inflows, and outflows. Outflow is zero for end lakes like Lake Urmia. The temporal variation of lake area, affecting precipitation on and evapotranspiration from the lake, is simulated as a non-linear function of lake water storage. WGHM contains more than 20 parameters that can be potentially be adjusted by calibration (Werth and Güntner, 2010).

WaterGAP includes a multitude of global data sets including information on irrigated areas, the fraction of irrigated areas that is equipped to be irrigated with groundwater (Siebert et al., 2010) and artificial drainage affecting return flows to surface water (Döll et al., 2012). For more information on data and model algorithms used in WaterGAP please refer to Müller Schmied et al. (2014) and Döll et al. (2014a). WGHM can be run globally or for specific basins only. In this study, it was run only for the 22 0.5° grid cells that represent the Lake Urmia basin in WGHM (Fig. 4).

WaterGAP outputs were extensively compared to in-situ streamflow observations (e.g., Döll et al., 200; Müller Schmied et al., 2014), to GRACE TWSA (Döll et al., 2012, 2014a, b) and GPS TWSA (Döll et al., 2014b). Results were shown to depend on applied climate input data sets (e.g., Müller Schmied et al., 2014, 2016; Döll et al., 2014b), model structure (Müller Schmied et al., 2014), and assumptions on water use (Döll et al. 2014a, b). Comparison of observed streamflow regime indicators (different streamflow percentiles representing statistical low and high flows) to the values computed by nine (or seven) GHMs showed that WaterGAP is one of the best fitting models (Gudmundsson et al. 2012; Tallaksen and Stahl, 2014). Prudhomme et al. (2011) concluded that "of the three global models considered here, WaterGAP is arguably best suited to reproduce most regional characteristics of large-scale high and low flow events in Europe." Regarding the fit to GRACE and GPS TWS, Döll et al. (2014b) found that WaterGAP underestimates seasonal variations of TWS on most of the land area of the globe and that seasonal maximum TWS occurs one month earlier according to WaterGAP than according to GRACE on most land areas.

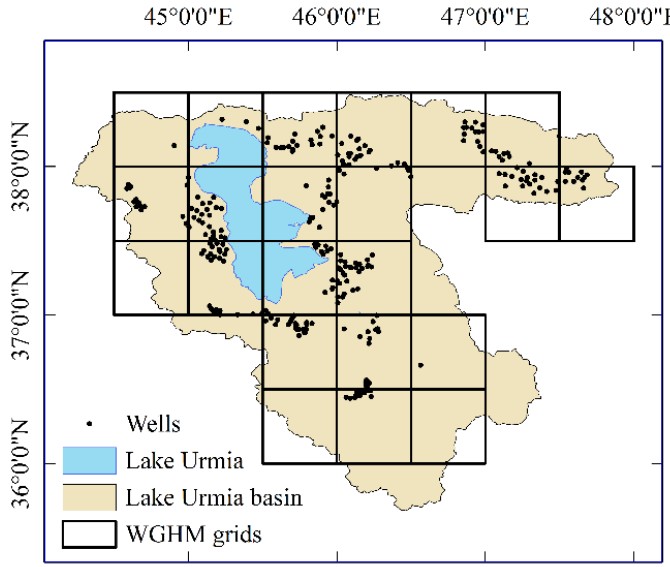

**Figure 4: Grid cells in WGHM corresponds to Lake Urmia basin along with the location of groundwater wells across the basin.**

## 2.2 Data

### 2.2.1 Remote sensing data

5    **Irrigated area in Lake Urmia basin.** Based on MODIS images, Kamali and Youneszadeh Jalili (2015) estimated annual time series of irrigated areas in Lake Urmia basin from 2001 to 2012. Considering that water management in the basin aims at preventing any increase of irrigated areas, it is assumed that irrigated area in 2013 remained at the 2012 value (Fig. 5).

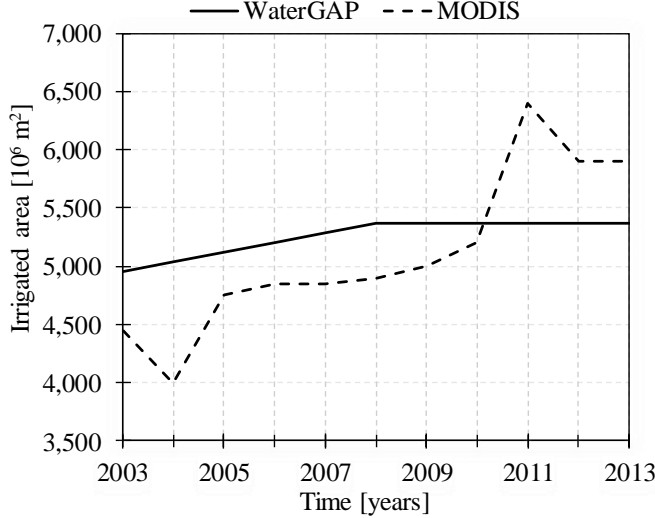

**Figure 5: Irrigated area in Lake Urmia basin assumed in WaterGAP and derived from MODIS (data from Kamali and Youneszadeh**
10    **Jalili, 2015).**

**GRACE total water storage anomalies.** GRACE satellite data allow derivation of monthly time series of total water storage anomalies (TWSA) over all continents. TWSA describes the total amount of water stored on the continents, including water storage in surface water bodies, groundwater and soil, as compared to the mean value of total water storage over a reference period. In our study CSR GRACE RL05 mascon solutions (Save et al., 2016; http://www2.csr.utexas.edu/grace/RL05_mascons.html, last accessed: 17 Jul. 2018) were used. While it is recommended GRACE data products only for areas with at least 100,000 km$^2$ (Watkins et al., 2015; Landerer and Swenson, 2012), studies by Tourian et al. (2015) and Lorenz et al. (2014) showed that signal strength or the so-called gravimetric resolution is determining the applicability of GRACE data. In fact, Lake Urmia basin has experienced an $8 \cdot 10^9$ m$^3$ change in the water volume in the last decade, which allows the use of GRACE for monitoring the changes in water storage in the basin (Tourian et al., 2015). This fact is supported by the very small gain factor of 1.0083 for the Lake Urmia basin based on Community Land Model 4 (CLM4) for spherical harmonic solutions (Landerer and Swenson, 2012), which is the factor with which signal attenuation due to leakage could be balanced. We can assume errors of the applied GRACE monthly time series of TWSA are small compared to the uncertainty of TWSA as computed by WGHM, such that model calibration against GRACE TWSA is meaningful.

### 2.2.2 Inflow into Lake Urmia

We used total annual observed inflow into the lake during 2003-2013 which was computed by the Urmia Lake Restoration Program (ULRP) based on 19 hydrometric stations around the lake (data available in http://ulrp.sharif.ir/ (In Persian), last accessed: 12 Nov. 2017). Monthly observations were not available. It was compared to the sum of simulated river discharge of all WGHM grid cells flowing into the grid cell representing Lake Urmia.

### 2.2.3 Groundwater levels

For evaluating the groundwater status in Lake Urmia basin, we used groundwater head data of 284 wells during 2003-2013 (Fig. 4). To obtain a monthly time series of average groundwater level in the basin, first the average of all groundwater level in each 0.5° grid cell was calculated and then the average values of all grid cells (see Strassberg et al., 2009).

### 2.2.4 Water withdrawals and consumptive uses

There are no water withdrawals time series data in Lake Urmia basin. However, water withdrawals in the Lake Urmia basin for 2009 was reported to be $4,825 \cdot 10^6$ m$^3$ (ULRP, 2015c) of which 89% is used for irrigation (Table 1). 57% of the withdrawn water is taken from surface water, the rest from groundwater. According to the report of Mahab Ghodss Consulting Engineering (2013), 16% of the water withdrawn for irrigation returns to groundwater and only 2% to surface water bodies, while the respective values for industrial and domestic water withdrawals are 50% and 10%. In this study, observed consumptive irrigation use was computed by subtracting total return flow from total water withdrawals for irrigation. Thus, it was set to 82% of water withdrawals for irrigation, while observed consumptive use in the domestic/industry sector was set to 40% of

sectoral water withdrawals. The sum of consumptive water use in all sectors is the so-called total net abstraction (NA) from either surface water bodies or groundwater.

**Table 1: Water withdrawals in Lake Urmia basin in 2009 [10⁶ m³] (data from URLP, 2015c).**

| Source | Sector | | | Total |
|---|---|---|---|---|
| | Agricultural | Domestic | Industry | |
| Surface water | 2424 | 276 | 33 | 2733 |
| Groundwater | 1867 | 190 | 35 | 2092 |
| Total | 4291 | 466 | 68 | 4825 |

### 2.2.5 Climate

The 0.5° gridded EartH2Observe, WFDEI and ERA-Interim Data Merged and Bias-corrected for ISIMIP (EWEMBI) dataset (Lange, 2016) was used as forcing data set. EWEMBI includes daily climate data for 1979 to 2013. For EWEMBI, ERA-Interim Reanalysis Data were bias-corrected with monthly observation data on temperature, precipitation and the number of wet days as well as daily radiation data. We compared, for the period 2003-2013, basin-average monthly precipitation and temperature values of EWEMBI dataset with those derived as the mean over monthly values observed at 143 rain gauges and six temperature gauging stations. The correlation coefficient (CC), Nash-Sutcliffe efficiency (NSE), and Willmott's refined index of agreement (Willmott et al., 2012) were 0.985, 0.946, and 0.897, respectively, for precipitation, and 0.996, 0.983, and 0.941 respectively, for temperature.

### 2.2.6 Lake volume

Based on remote sensing data for lake extent and water table elevation as well as on in-situ bathymetry data, a time series of monthly water volume in Lake Urmia for the period 2003-2013 was generated by Tourian et al. (2015) (their Fig. 9). It was used for evaluation of the model variants.

### 2.3 Calibration variants

Calibration was done by trial-and-error. It included the modification time series of irrigated area, of NAg and NAs, with different multipliers for individual years, as well as the modification of a maximum of seven temporally constant model parameters or, in case of spatially heterogeneous parameters, multipliers. Modifications were done homogeneously for the whole basin. Months with assumed irrigation in Lake Urmia basin according to WaterGAP correspond to the actual irrigation months (Apr. and Oct.) in the basin according to Saemian et al. (2015), Thus, no correction of the seasonality was needed in the calibration process. Fig. 6 shows a schematic of the calibration process for the four calibration variants. Please note that the identified parameter combinations are not the only ones that would lead to a good fit to observations.

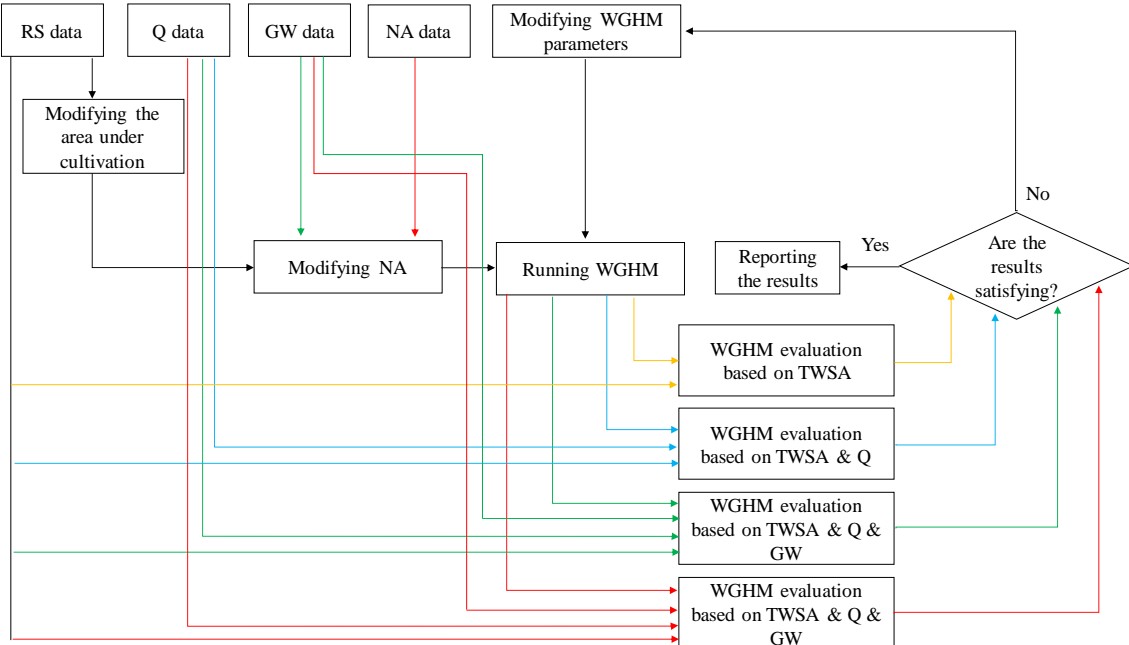

**Figure 6: Flowchart for the four calibration variants. The black line is common in all variants, the mustard, blue, green and red lines represent calibration based on RS data (RS variant), RS data and inflow data (RS_Q variant), RS, inflow and groundwater level data (RS_Q_GW variant), and RS, inflow, groundwater level and net abstraction data (RS_Q_GW_NA variant), respectively.**

## 2.3.1 RS variant: Calibration using remote sensing data

Irrigated area in Lake Urmia basin used in the standard version of WaterGAP is larger than the MODIS-based irrigated area until 2010, and smaller afterwards (Fig. 4). The largest differences, in 2004 and 2011, exceed 20%, or 1,000 km$^2$, and the strongly increasing trend is not represented in WaterGAP. The constant value of irrigated area in WaterGAP is due to the fact that the Food and Agricultural Organization of the UN does not provide more recent estimates of irrigated area in Iran (see http://www.fao.org/nr/water/aquastat, last accessed: 13 Feb. 2018). To utilize the MODIS-based time series, consumptive irrigation water use in the whole basin of WaterGAP in year i was first adjusted by multiplying it by a correction factor CF1(i), with:

$$CF1(i) = \frac{Area_{irri}^{MODIS}(i)}{Area_{irri}^{WG}(i)} \tag{1}$$

where $Area_{irri}^{MODIS}(i)$ is irrigated area from MODIS in year i and $Area_{irri}^{WG}(i)$ is irrigated area from WaterGAP database. The modified consumptive irrigation use was then added to the consumptive use of WaterGAP for the other sectors to obtain an updated basin-wide NA for each year. Then, modified monthly NAg and NAs in year i were calculated by multiplying, for each grid cell, the standard WaterGAP NAg and NAs values with the ratio of modified over standard basin-wide NA in year i. Then, WGHM was run with the modified NAg and NAs time series, and a small number of WGHM parameters was varied until achieving a good fit to monthly time series of basin-average GRACE TWSA (Fig. 6, yellow lines).

### 2.3.2 RS_Q variant: Calibration using remote sensing data and inflow into the lake

Model parameters of WGHM driven by modified NAs and NAg from the RS variant were adjusted to achieve a good fit for both GRACE TWSA and the time series of annual total inflows to Lake Urmia (Fig. 6, blue lines).

### 2.2.3 RS_Q_GW variant: Calibration using remote sensing data, inflow into the lake, and groundwater level

Since WGHM does not compute groundwater level but only groundwater storage, and there is no good information of basin-wide specific yield that would allow a translation of observed groundwater level variations into storage variations, model calibration in this variant aimed at optimizing the fit between the monthly time series of normalized basin-average observed groundwater levels (calculated by subtracting the mean and dividing by the standard deviation) to the monthly time series of normalized WGHM groundwater storage. To achieve a good fit to groundwater levels, and at the same time to GRACE TWSA and observed inflow into the lake, NAg and NAs as adjusted in variant RS had to be further modified. Keeping total NA(i) constant, correction factors $\alpha(i)$ and $\beta(i)$ were determined , with:

$$NA(i) = \alpha(i) \times NAs(i) + \beta(i) \times NAg(i) \tag{2}$$

and new values of temporally constant model parameters were identified (Fig. 6, green lines).

### 2.3.4 RS_Q_GW_NA variant: Calibration using remote sensing data, inflow into the lake, groundwater level, and net abstractions

In the most involved calibration variant, statistical data on water withdrawals in 2009 (Table 1) was used together with information on return flow to compute a consumptive irrigation water use $Cu_{irri}^{Obs}$ in the basin of $3,520 \cdot 10^6$ m³. To estimate irrigation use in all other years, with different climatic conditions, the per area consumptive irrigation water use from WaterGAP was used to compute, for each year, a climatic correction factor CF2(i) as

$$CF2(i) = \left( \frac{Cu_{irri}^{WG}(i)}{Area_{irri}^{WG}(i)} - \frac{Cu_{irri}^{WG}(2009)}{Area_{irri}^{WG}(2009)} \right) \tag{3}$$

where $CF2(i)$ is represents the difference in the per area consumptive irrigation use in year i and the year 2009, $Cu_{irri}^{WG}(i)$ is consumptive irrigation use in year i obtained in standard WaterGAP. Finally, Eq. 4 was used for estimating water consumption time series over Urmia basin:

$$Cu_{irri}(i) = \left( \frac{Area_{irri}^{MODIS}(i)}{Area_{irri}^{MODIS}(2009)} \right) \times Cu_{irri}^{Obs}(2009) + CF2(i) \times Area_{irri}^{MODIS}(i) \tag{4}$$

where $Cu_{irri}(i)$ is consumptive irrigation water use in year i. Unlike in the RS_Q_GW variant, consumptive use of the other sectors was added based on withdrawal data in Table 1 and a return flow fraction of 60%, resulting in total NA. Then, new values for correction factors $\alpha(i)$ and $\beta(i)$ (Eq. 2) were identified by trial-and-error, and model parameters were modified to obtain a good fit to the data also used in the RS_Q_GW variant (Fig. 6, red lines).

**2.4 Performance indicators**

Performance of the calibration variants of WGHM was evaluated using CC, NSE, root mean square error (RMSE), relative absolute error (RAE), and Kling Gupta efficiency (KGE, Gupta et al., 2009) with

$$CC = \frac{Cov\,(Obs.Sim)}{\sigma_{obs} \times \sigma_{Sim}} \tag{5}$$

$$NSE = 1 - \frac{\sum_{t=1}^{T}(Sim_{(t)} - Obs_{(t)})^2}{\sum_{t=1}^{T}(Obs_{(t)} - \overline{Obs})^2} \tag{6}$$

$$RMSE = \sqrt{\frac{1}{T}\sum_{t=1}^{T}\left(Obs_{(t)} - Sim_{(t)}\right)^2} \tag{7}$$

$$RAE = \frac{\sum_{t=1}^{T}|Obs_{(t)} - Sim_{(t)}|}{\sum_{t=1}^{T}|Obs_{(t)} - \overline{Obs}|} \tag{8}$$

$$KGE = 1 - \sqrt{(CC-1)^2 - \left(\frac{\sigma_{Sim}}{\sigma_{obs}} - 1\right)^2 + (\frac{\overline{Sim}}{\overline{Obs}} - 1)^2} \tag{9}$$

where *Cov* is covariance function, *Obs* is observed value, *Sim* is simulated value, *t* refers to time counter and *T* is the period length. Optimum values of CC, NSE and KGE are 1, and of RMSE and RE are 0. Trends and overall behaviour of the time series were also analysed.

**3 Results**

**3.1 Multi-observation calibration**

In variants RS and RS_Q, annual time series of irrigated area in Lake Urmia basin derived from MODIS (Fig. 4), which were applied in all four calibration variants, lead to a more strongly increasing trend of NA (consumptive water use) and NAs, as compared to the standard WaterGAP version (Fig. 7). Due to the dominant irrigation with surface water assumed in the standard version of WaterGAP, return flows from irrigation are larger than groundwater withdrawals, and there is a net recharge of groundwater by irrigation, i.e. a negative NAg. Therefore, a more strongly increasing irrigation with surface water in variants RS and RS_Q leads to return flows to groundwater that increase more strongly over time, i.e. NAg becomes increasingly negative with time (Fig. 7). Average NA in 2003-2010 decreased from 4,185·10$^6$ m$^3$/yr in the standard version to 3,815·10$^6$ m$^3$/yr, and increased from 4,233·10$^6$ m$^3$/yr to 4,781·10$^6$ m$^3$/yr in 2011-2013. However, increased net recharge of groundwater by return flows was found to be incompatible with decreasing observed groundwater levels (Fig. 8c). Positive NAg values were found to be necessary to simulate the observed lowering of groundwater levels from 2003 to 2013 Therefore, in variant RS_Q_GW, NAg and NAs were adjusted according to Eq. 2 by applying $\alpha$ and $\beta$ time series presented in Table 2. With these adjustment factors, average NAg changed from -2,294·10$^6$ m$^3$/yr in variants RS and RS_Q to 1,147·10$^6$ m$^3$/yr in variant

RS_Q_GW (Fig. 7b). Keeping annual NA constant, NAs decreased accordingly from $6,373 \cdot 10^6$ m³/yr to $2,931 \cdot 10^6$ m³/yr. Total NA slightly decreased in variant RS_Q_GW_NA as compared to the other calibrations variants.

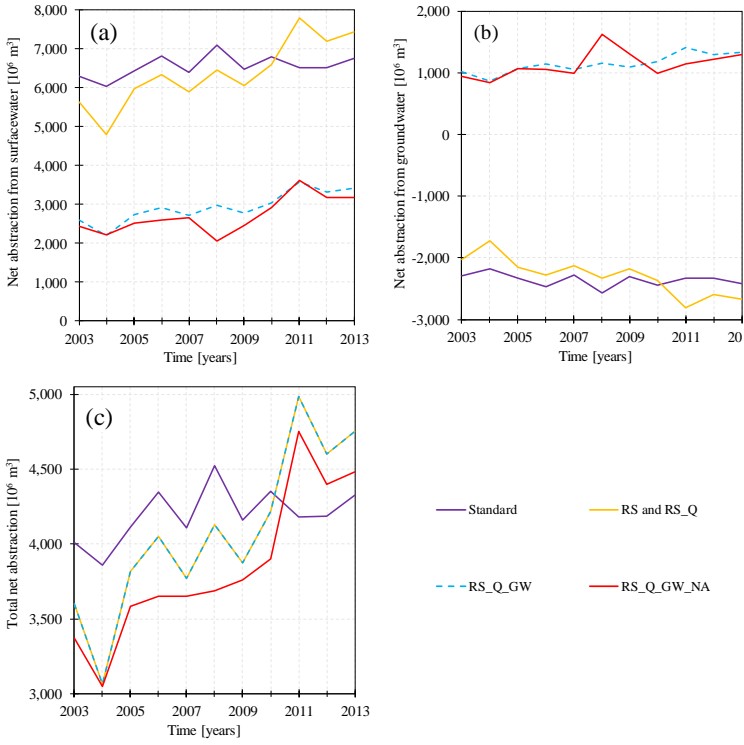

**Figure 7: Time series of net abstractions from surface water (a) and groundwater (b), as well as total net abstractions (i.e. consumptive use) (c) in Lake Urmia basin in the standard version of WaterGAP as well as the various calibration variants.**

**Table 2: Correction factors for modifying NAs and NAg (see Eq. 2).**

| Variant | RS_Q_GW | | | RS_Q_GW_NA | |
|---|---|---|---|---|---|
| Year | $\alpha$ | $\beta$ | | $\alpha$ | $\beta$ |
| 2003 | 0.47 | -0.48 | | 0.39 | -0.41 |
| 2004 | 0.46 | -0.49 | | 0.37 | -0.39 |
| 2005 | 0.46 | -0.50 | | 0.39 | -0.46 |
| 2006 | 0.46 | -0.50 | | 0.38 | -0.43 |
| 2007 | 0.46 | -0.50 | | 0.42 | -0.43 |
| 2008 | 0.45 | -0.52 | | 0.29 | -0.63 |
| 2009 | 0.46 | -0.49 | | 0.38 | -0.57 |
| 2010 | 0.47 | -0.48 | | 0.43 | -0.41 |
| 2011 | 0.47 | -0.47 | | 0.56 | -0.49 |
| 2012 | 0.46 | -0.51 | | 0.49 | -0.52 |
| 2013 | 0.45 | -0.52 | | 0.47 | -0.54 |

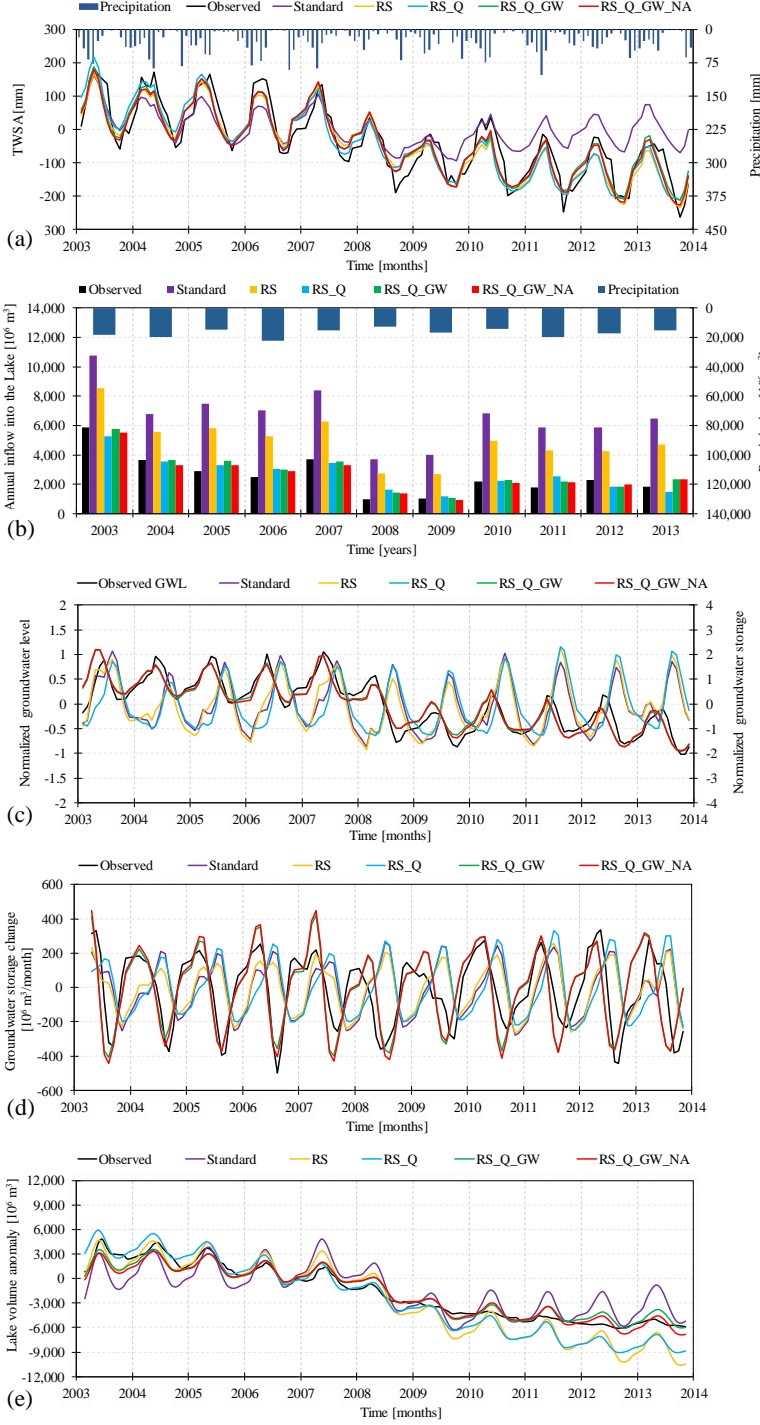

**Figure 8: Time series of monthly TWSA of GRACE and WGHM (a), annual inflow into the lake Q from observations and WGHM (b) normalized observed groundwater level and normalized groundwater storage from WGHM (c), groundwater storage change GWSC from month to month from observations and WGHM (d) and the monthly lake volume anomaly (e), for standard WaterGAP and the four calibration variants.**

Model runs driven by the different NAg and NAs of the four variants lead to the best fit to the variant-specific observational datasets if seven model parameters were re-set to the values listed in Table 3. It is emphasized that the listed parameter sets are not the only possible ones but those requiring the least number of parameters to be changed. In all four calibration variants, the minimum daily precipitation values for which groundwater recharge can occur in semi-arid regions (Döll and Fiedler, 2008) was slightly decreased (increasing groundwater recharge) and the maximum canopy storage was increased (increasing canopy evaporation). When the more observational data types were considered in the calibration process, the number of parameters that needed to be adjusted increased whereas the required parameter changes decreased.

According to GRACE observations, total water storage in Lake Urmia basin declined by $9.9 \cdot 10^9$ m$^3$ from its annual average in 2003 to its annual average in 2013, while the standard WGHM version computes a much smaller loss. According to the data of Tourian et al. (2015), about 80 % of the total water loss in the basin was due to the loss of lake water. A stronger increase of human water abstractions over time (Fig. 7a), doubling of rooting depth and thus soil water capacity and a higher maximum canopy storage everywhere in the basin, as well as an increase of maximum active lake depth of Lake Urmia from 5 m to 9 m in variant RS resulted in a good fit of WGHM TWSA to GRACE TWSA (Fig. 8a). With the larger soil and canopy water storage capacities, runoff and thus inflow into Lake Urmia decrease as compared to standard WGHM (Fig. 8b). More water could be stored in canopy, soil, and the lake at the beginning of the period such that storages could react to the decline of inflows and decrease after 2007. Still, simulated inflows into Lake Urmia computed in variant RS are still much higher than the observed values (Fig. 8b) and seasonality of groundwater levels is totally misrepresented (Figs. 8c, d).

**Table 3: WGHM parameter values adjusted by calibration in the different model variants.**

| Variant | Rooting depth multiplier | Maximum active lake depth | Runoff coefficient multiplier | Multiplier for the fraction of total runoff that becomes groundwater recharge | Maximum amount of groundwater recharge per day multiplier | Minimum amount of daily precipitation necessary in arid/semi-arid areas to get groundwater recharge [mm] | Maximum canopy storage [mm] |
|---|---|---|---|---|---|---|---|
| Standard | 1 | 5 | 1 | 1 | 1 | 12.5 | 0.3 |
| RS | 2 | 9 | 1 | 1 | 1 | 10 | 1 |
| RS_Q | 2.8 | 10 | 0.9 | 1 | 1 | 10 | 1 |
| RS_Q_GW | 3 | 9 | 0.8 | 0.5 | 4 | 10 | 1 |
| RS_Q_GW_NA | 3 | 8 | 0.8 | 0.5 | 5 | 10 | 1 |

The required reduction of computed lake inflow (Q) can be achieved (Fig. 8b) by further increasing soil water storage capacity in variant RS_Q, together with small adjustment of the runoff coefficient and active lake depth (Table 3), while the fit to GRACE TWSA remains good (Fig. 8a). However, seasonality of groundwater table fluctuations is still not simulated properly. This could only be achieved by adjusting the sources of total net abstractions. Only if net abstractions from groundwater are multiplied by approximately -0.5 (Table 2), in variant RS_Q_GW, does the seasonality of computed groundwater storage variations fit to observations (Fig. 8c). NAg in the standard, RS and RS_Q variants is negative, which means that there is artificial groundwater recharge due to irrigation by surface water during the summer irrigation months,

leading to an increase in groundwater level and storage. Groundwater level observations, however, show a decrease during this period, indicating that irrigation causes a net abstraction from groundwater. Multiplication of standard WGHM NAg by a negative value leads to a net abstraction of water from the groundwater body, and results in a seasonality of groundwater storage that fits well to the seasonality of the mean groundwater table in the basin. In addition to the NAg and NAs adjustment,

two groundwater recharge-related parameters had to be re-set in variant RS_Q_GW (Table 3). The fit to observed TWSA and lake inflow remains good (Figs. 8a, b). Use of local information on water withdrawals and return flows in variant RS_Q_GW_NA barely changed the parameter values (Table 3) and the fit to all observational data (Fig. 8).

From the results of the RS_Q_GW_NA variant, which was the most comprehensive calibration variant, we estimated the average specific yield of the aquifers in the Lake Urmia Basin, i.e. the change in groundwater storage per unit change of

10 the elevation of the groundwater table. We first divided the standard deviation of the simulated groundwater storage time series by the basin area to obtain groundwater storage variability in terms of equivalent water height and then divided this value by the standard deviation of the observed groundwater levels. This resulted in a specific yield estimate of 0.02, which is equal to the average value derived from pumping tests at 10 locations south of the lake (Hamzekhani and Aghaie, 2015). Estimated specific yield allows to compute an "observed" groundwater water storage anomaly, and thus an observed decline of

15 groundwater storage between the year 2003 and 2013 of $1.8 \cdot 10^9$ m$^3$, accounting for 18% of the observed total water storage loss in the basin. We compared the time series of simulated groundwater storage changes from month to month (GWSC) to those derived from observations of groundwater level changes. Since groundwater level observations were done only once per month and at different days, three-month moving averages were compared (Fig. 8d). Observations and model variants RS_Q_GW and RS_Q_GW_NA agree that the strongest monthly increase in groundwater storage occurs in early spring, and

20 the largest decrease in early autumn.

The performance indicators CC, NSE, RMSE, RAE, and KGE with respect to monthly TWSA (Fig. 8a), annual Q (inflow to Lake Urmia, Fig. 8b) and monthly GWSC (Fig. 8d) are presented in Table 4 for the standard version and four calibrated variants. Regarding the fit to TWSA observations, NSE increased from 0.48 in the standard version to 0.84 in the RS variant for which TWSA was the only observation considered, and increased slightly to 0.88 when groundwater

observations were taken into account in variants RS_Q_GW and RS_Q_GW_NA variants. This performance improvement is also reflected by CC, RMSE, RAE, and KGE. The performance with respect to observed inflow to the lake only improves marginally by calibration against TWSA, in variant RS. Only calibration against inflow observations strongly improves model performance, with NSE and KGE jumping from negative values for the standard variant to values around 0.9 and RAE from 3.92 to 0.30. Integration of groundwater observations again leads to a small performance improvement (see also RMSE). The

good performance shown by CC for all model variants indicates that all model variants identify correctly high and low flow years. In the case of GWSC, all performance indicators show that consideration of remote sensing and streamflow observations only do not lead to an acceptable simulation of groundwater storage. Only the two variants for which groundwater observations were taken into account lead to satisfactory performance. With a maximum NSE of 0.59 and KGE of 0.75, the fit to GWSC remains lower than the one to TWSA and lake inflow, which may also be due to the uncertainty in estimating the basin-wide

average monthly groundwater storage behavior from well observations. The most data-demanding variant RS_Q_GW_NA achieves the best fit to all three observational time series. The fit, however, is only slightly better than the fit of variant RS_Q_GW, and a much more variable time series of NAg and NAs correction coefficients (Table 2) is necessary as compared to variant RS_Q_GW (Table 2).

5   For model performance evaluation, we compared the lake volume simulated by WGHM with the observed lake volume of Tourian et al. (2015) (Fig.8e and Table 4). The standard model underestimates the decline in both lake water and TWSA, all calibrated variants simulate the TWSA trend correctly, but both variant RS and RS_Q, with worse KGE than the standard version, overestimate the decline of lake water storage, thus compensating for not decreasing sufficiently groundwater storage (Fig. 8d) due to assuming a net groundwater recharge due to surface water irrigation. Only variants RS_Q_GW and

10 RS_Q_GW_NA simulate not only the groundwater dynamics but also the decline of lake water volume correctly. NSE for the monthly lake volume anomaly is 0.68 for the standard WGHM and improves to 0.77 for RS, where GRACE TWSA could be simulated well by approximately doubling both soil and lake water storage capacity (Table 3). Including groundwater level data further improved the fit to observed lake volume, leading to a very high NSE of 0.94 or 0.95 (Table 4). We conclude that calibration of WGHM against diverse observations (that do not include lake volume observations) leads to improved simulation

15 of lake volume dynamics.

**Table 4: Performance of standard and calibrated WGHM variants with respects to observations of TWSA, inflow to lake, GWSC and lake volume anomaly**

| Phase | Variables | Criteria | Standard | RS | RS_Q | RS_Q_GW | RS_Q_GW_NA |
|---|---|---|---|---|---|---|---|
| Calibration | Monthly TWSA | CC | 0.84 | 0.93 | 0.92 | 0.94 | 0.94 |
| | | NSE | 0.48 | 0.84 | 0.83 | 0.88 | 0.88 |
| | | RMSE [mm] | 77 | 42 | 44 | 38 | 37 |
| | | RAE | 0.72 | 0.41 | 0.42 | 0.37 | 0.36 |
| | | KGE | 0.64 | 0.80 | 0.79 | 0.82 | 0.83 |
| | Annual Q | CC | 0.94 | 0.96 | 0.95 | 0.97 | 0.97 |
| | | NSE | -8.51 | -2.33 | 0.88 | 0.91 | 0.93 |
| | | RMSE [$10^6$ m$^3$/year] | 4121 | 2438 | 458 | 390 | 358 |
| | | RAE | 3.92 | 2.32 | 0.38 | 0.33 | 0.30 |
| | | KGE | -0.60 | 0.07 | 0.84 | 0.88 | 0.91 |
| | Monthly GWSC | CC | -0.14 | 0.05 | -0.31 | 0.80 | 0.82 |
| | | NSE | -0.72 | -0.39 | -1.05 | 0.55 | 0.59 |
| | | RMSE [$10^6$ m$^3$/month] | 271 | 244 | 296 | 109 | 103 |
| | | RAE | 1.28 | 1.13 | 1.42 | 0.60 | 0.58 |
| | | KGE | -0.57 | -0.44 | -0.79 | 0.71 | 0.75 |
| Evaluation | Monthly lake volume anomaly | CC | 0.82 | 0.97 | 0.99 | 0.98 | 0.97 |
| | | NSE | 0.68 | 0.77 | 0.81 | 0.94 | 0.95 |
| | | RMSE [$10^6$ m$^3$] | 1922 | 1837 | 1611 | 757 | 739 |
| | | RAE | 0.51 | 0.47 | 0.42 | 0.21 | 0.20 |
| | | KGE | 0.70 | 0.34 | 0.41 | 0.88 | 0.90 |

## 3.2 Differential impacts of human water use and climate variation on Lake Urmia basin

The impact of human water use and man-made reservoirs on water flows and storages was quantified by comparing the output

20 of WGHM in which human water use and man-made reservoirs are considered (this is normally done, now called WGHM-

ANT) with the output of a model run for naturalized conditions, where it is assumed that there are no reservoirs and no human water use (WGHM-NAT). We determined that the results of the naturalized run differ by less than 2% from a run with reservoirs but without human water use. Therefore, differences between WGHM-ANT and WGHM-NAT outputs can be considered to be caused by human water use. It should be mentioned that all simulated and observed storages (total, groundwater, lake) are not absolute values but anomalies with respect to the mean water storage during 2004-2009 (baseline period used for the provided GRACE data).

When comparing TWSA under anthropogenic and naturalized conditions in Fig. 9a, remember that TWSA in Lake Urmia basin is dominated by water storage in Lake Urmia. Seasonal TWSA variation of WGHM-ANT and WGHM-NAT do not differ much. Starting after the heavy rain in April 2007 and strongly caused by the lack of spring precipitation in 2008, both WGHM-ANT and WGHM-NAT (as well as GRACE TWSA) show a decreasing trend that is only somewhat more pronounced in WGHM-ANT (Fig. 9a). Thus, this decrease is mainly due to dry climate conditions during the well-known severe drought of 2008, with an annual precipitation of only 241 mm, i.e. 74% of the mean value for 2003-2013 (Fig. 8b). Also in the absence of human water use, total water storage would not have recovered after 2009 but would have stayed 50-100 mm below the values occurring before 2008. However, while in WGHM-NAT the minimum storage in late summer, i.e. the period with high irrigation, remains at a constant level after 2009, it decreases each year in WGHM-ANT due to consumptive increasing irrigation water use (Fig. 7c). The linear trend of WGHM-ANT and WGHM-NAT TWSA time series for the period 2003-2013 is -24.5 mm/yr (GRACE: -24.4 mm/yr) and -11.8 mm/yr, respectively. The TWSA trend for two sub-periods before and after 2008, 2003-2007 and 2009-2013 -14.2 and -16 mm/yr, respectively, for WGHM-ANT and only 0.7 and -3.85 mm/yr, respectively, for WGHM-NAT. The last mentioned trends are not significant at the 5% confidence level based on Mann-Kendall's test. According to WGHM, the basin lost, on average during 2003-2013, $1,274 \cdot 10^6$ m$^3$ water/yr, while in the absence of human water use, it would have lost $614 \cdot 10^6$ m$^3$ water/yr, i.e. 52% less. Of this total water volume, $920 \cdot 10^6$ m$^3$/yr of lake water was lost, while only $548 \cdot 10^6$ m$^3$/yr would have been lost without human water use (Fig. 9b).

The smaller decreasing trend for lake water volume under naturalized conditions is clearly caused by more inflow into the lake, even though lake evaporation is somewhat higher under naturalized inflow conditions due to the larger lake extent. While mean inflow during 2003-2013 is computed to be $4,454 \cdot 10^6$ m$^3$/yr under naturalized conditions, it decreases by 41% to $2,639 \cdot 10^6$ m$^3$/yr under anthropogenically altered conditions (Fig. 9c). The difference is only 50% of NA as only a fraction of (potential) net abstractions from surface water NAs (required to allow optimal irrigation) could be made 1) due to a lack of water availability in the surface water bodies and 2) because a fraction oft of NAg is provided a decrease in groundwater storage.

Since 2008 the inflow into the lake has never reached $3,085 \cdot 10^6$ m$^3$/yr. This is the value estimated to be the minimum environmental water requirements that compensates the amount of annual evaporation from of the lake surface (Abbaspour and Nazaridoust, 2007). Therefore, a decrease of lake water storage can be expected for the best estimate of WaterGAP of $2,639 \cdot 10^6$ m$^3$/yr. In WGHM-NAT, inflow was lower than $3,085 \cdot 10^6$ m$^3$ only in 2008 and 2009. Still, the average inflow into the lake from 2009-2013 of $3,670 \cdot 10^6$ m$^3$ would have been only enough to keep the lake from further loosing volume (needed

to compensate for lake evaporation). Thus even in the WGHM-NAT, inflow into the lake would not have been enough for a recovery to conditions between 2003 and 2007 (Fig. 9b), as during this time period, mean inflow under naturalized conditions would have been 54% larger. The ratio of inflow into the lake over precipitation in the basin varies strongly among the years, reaching a maximum value of 0.30 and 0.41 for anthropogenic and naturalized conditions, respectively, in 2003, and a minimum value of 0.11 and 0.18 in the drought year 2008. For the period 2009-2013, these ratios are, with 0.11 (ANT) and 0.22 (NAT), much smaller than the values for 2003-2007, 0.21 and 0.32. Thus, the drought year 2008 as well as the relatively small ratio of inflow into the lake over precipitation in the last five years of the study period play an equally important role as human water use in the decline of inflow and lake water storage.

While groundwater storage is estimated to decline by $251 \cdot 10^6$ m$^3$/yr during 2003-2013 in WGHM-ANT, the decline is only $27 \cdot 10^6$ m$^3$/yr in WGHM-NAT (Fig. 9d). Different from lake water storage, groundwater storage would have recovered after 2008/2009 if there had been no (increasing) net groundwater abstractions (Fig. 9d, compare Fig. 7b), even though mean groundwater recharge was on $2,579 \cdot 10^6$ m$^3$/yr during 2009-2013 as compared to $3,310 \cdot 10^6$ m$^3$/yr during 2009-2013. In WGHM, the groundwater compartment is modelled using a linear storage model where the change of groundwater storage is the difference between inflows to groundwater and outflow to surface water bodies, supplement by a prescribed outflow due to human groundwater use in case of anthropogenic conditions. Long-term average outflow from groundwater to surface water is proportional to the groundwater storage. Therefore, in case of less groundwater recharge, also the outflow to surface water bodies is decreased, while mean groundwater storage decreases only slightly, in particular in areas with a low average groundwater recharge like the Lake Urmia basin. In the absence of groundwater abstractions, the groundwater level cannot drop below the level of the surface water in WGHM. WGHM cannot simulate the case where groundwater switches from discharging groundwater to surface water bodies to receiving water from rivers and other surface water bodies. In case of groundwater abstractions, however, storage can drop below the level of the surface water, and outflow to surface water bodies ceases in this case.

In the WGHM-ANT simulations, such a drop below the surface water level, indicated by a negative water storage, value occurs in 7 out of the 22 0.5° grid cells within the basin (Fig. A1a). In 6 of these 7 grid cells, groundwater levels were stable during 2003-2007and only declined from 2008-2013, caused by increased NAg and decreased groundwater recharge in the latter part of the study period. It is these 7 cells that cause the basin groundwater decline under anthropogenic condition shown in Fig. 9d. For naturalized conditions, peak seasonal water storages decrease somewhat but minimum water storages cannot drop appreciably given the very low minimum seasonal storage values already during the relatively wet five first years of the growing period (Fig. A1b). Thus, the contribution of human water use to groundwater storage decline might therefore be overestimated as WaterGAP cannot simulate a possible drop of the groundwater table below the surface water level in the absence of groundwater abstractions. To summarize, human water use was the reason for 52% of the total water loss in the basin, for a maximum of 90% of the groundwater loss and for 40% of the Lake Urmia water loss during 2003-2013, and lake inflow was 41% less than it would have been without human water use.

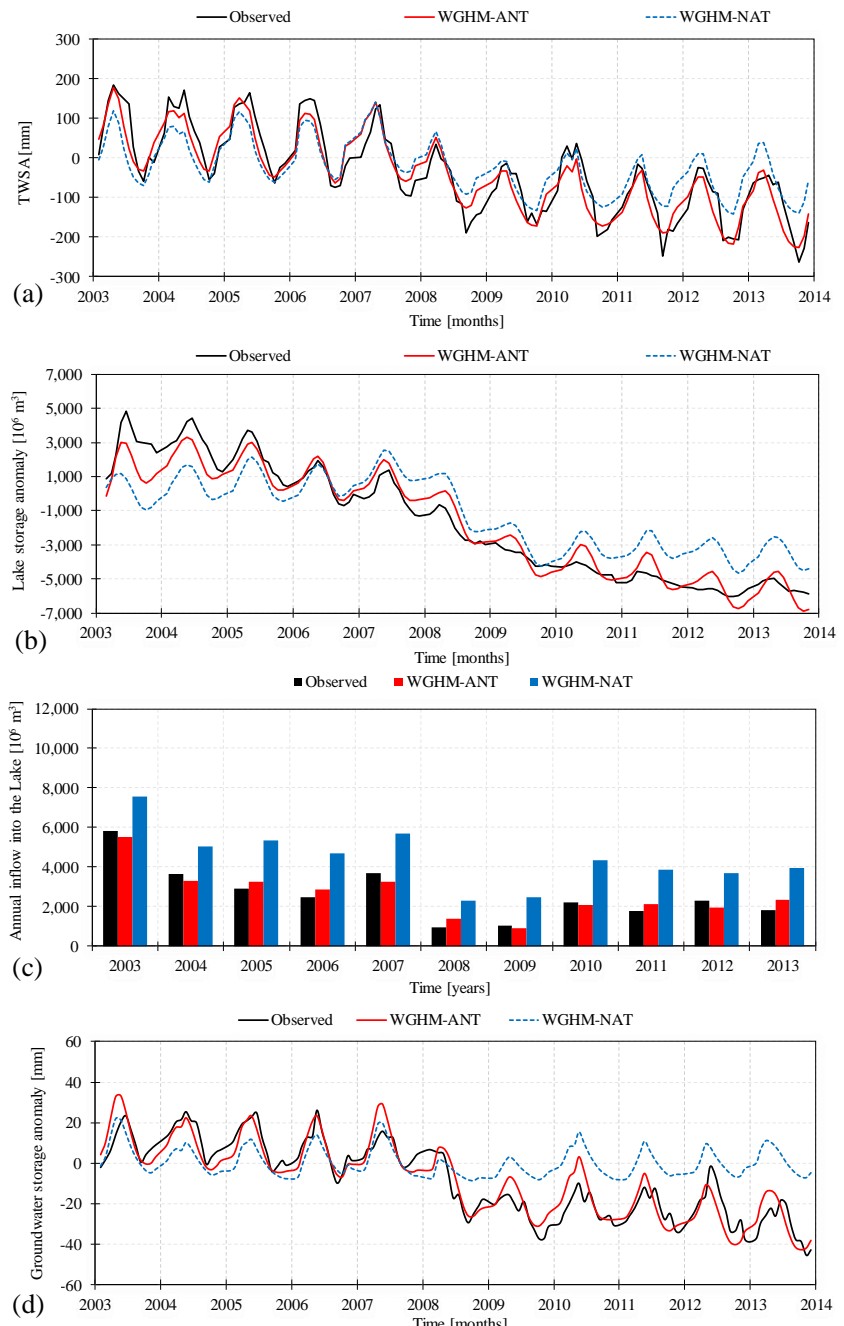

**Figure 9: Time series of simulated (variant RS_Q_GW_NA) and observed monthly TWSA (a), lake water storage anomaly (b), annual inflow into the lake (c), and monthly groundwater storage anomaly (d), under anthropogenic (WGHM-ANT) and naturalized (WGHM-NAT) conditions.**

## 4 Discussion

### 4.1 Multi-observation calibration

The output of hydrological models at all scales is uncertain as these models suffer from uncertain model inputs (e.g., climate variables or soil properties), parameter values and model structure (Döll et al., 2016). To decrease uncertainty, model calibration against independent data (e.g. observations) is performed by adjusting, for example, model parameters. While observations of river discharge are ideally suited for validating hydrological models because the point observation integrates over processes in the whole upstream basin of the gauging station, additional types of observations have to be added to avoid the well-known problem of equifinality (Beven and Freer, 2001; Döll et al., 2016). Without additional data, more than one parameter combination can lead to a good fit to e.g. observed river discharge. While e.g. total groundwater storage dynamics would be simulated very differently by model variants with the parameter sets that simulate river discharge time series equally well.

Global hydrological models suffer from a particularly high uncertainty, in particular as model inputs are uncertain. For example, climate input data are based on low-density climate observations and information on water use is often very scarce and outdated. For modelling at the global scale, it is generally not possible to obtain, the same detailed data for a specific region compared to the case that modelling this region only. Still, a global hydrological model includes all data for simulating water flows and storages in specific regions of interest everywhere on the globe, and model calibration against multiple (regional) observations is a means for improving the performance of the global model regionally. In this way, an efficient simulation of regional water flows and storages can be achieved, possibly as an alternative to a costlier setup of a regional model. More importantly, the regional-scale multi-observation calibration done in this study can serve to inform efforts for global-scale but region-specific multi-observation calibration of global hydrological models that would allow to strongly improve performance of global hydrological models at the scale that they are made for (Döll et al., 2016).

Remote sensing data are the most accessible data for calibration of global hydrological models, including TWSA from GRACE. Therefore, the model variant RS only used globally available RS data, MODIS and GRACE data products. However, MODIS data can only be used to determine the temporally variable extent of irrigated areas in dry regions of the globe such that the important adjustment of temporal dynamics of statistics-based irrigated areas is not possible everywhere. GRACE TWSA quantify the anomalies and changes of water storage aggregated over all land water storage compartments such as snow, soil, groundwater, lakes, wetlands, and rivers. Considering GRACE TWSA improved the simulation of the important water storage compartment Lake Urmia. However, the unsatisfactory simulation of inflow into Lake Urmia and of groundwater dynamics clearly shows that a good fit to observed TWSA does not guarantee a good simulation of river flows or groundwater storages. Still, calibration against TWSA did, even if only very slightly, improve model performance also with respect to lake inflow and groundwater dynamics.

By adding discharge data, the model was able to simulate TWSA and Q accurately without changing the inputs of the model and only based on modifying the parameters, mainly increasing the rooting depth further (Table 3). Interestingly, the

significant increase of the rooting depth multiplier from 2.0 to 2.8 strongly increased evapotranspiration but barely affected TWSA (Figs. 7a, b). In the case of the Lake Urmia basin, no trade-off between the fit to TWSA and river discharge exists as the performance indicators with respect to TWSA for variant RS_Q are even slightly higher than for variant RS (Table 4).

Groundwater level data were found to be necessary to identify that different from what is estimated by the standard version of WaterGAP, there is more irrigation with groundwater and less with surface water such that a net abstraction of groundwater and not artificial groundwater recharge occurs due to irrigation. Information on groundwater level dynamics with a suitable spatial density is not readily available for most regions of the globe. To simulate groundwater dynamics properly, it was not sufficient to adjust parameters of the hydrological model (in particular two groundwater recharge related model parameters, Table 3), but it was necessary to alter the fractions of net water abstractions that come from groundwater and

surface water bodies. Only then, groundwater storage decline by net groundwater abstraction was simulated, and lake water storage decline could be correctly simulated instead of being overestimated when only TWSA and lake inflow data are used for calibration. As in the case of adding lake inflow as calibration data type, no trade-off between the fits to the different data types occurred.

        Consideration of regional estimates of human water withdrawals in a specific year as well as regional estimates of

return flow fractions in variant RS_Q_GW_NA does not improve the fit to observations significantly and only leads to slight parameter adjustments. This indicates a reasonable simulation of per hectare water consumption for irrigation by the WaterGAP model. To summarize, consideration of more and more observations and other independent data results with improved fits to three type of observations, TWSA, lake inflow, and groundwater dynamics, while at the same time more and more parameters need to be adjusted (Tables 3 and 4). No trade-offs between the fits to the three observational data types

occurred in the case of the Lake Urmia basin.

        While the introduction of annually varying corrections for NAg and NAs (Eq. 2, Table 2) for variants RS_Q_GW and RS_Q_GW_NA leads to the most suitable fit to multiple observation types, it may be preferable to have instead of 11 free parameters just 1, i.e. a temporally constant $\beta$. With a temporally constant $\beta$ of -0.5 in variant RS_Q_GW, the fit to TWSA and inflow to the lake does not change at all, and groundwater storage is only slightly increased in the dry year 2008 and 2009.

Thus, given the uncertainty of observed groundwater storage variations, a temporally constant NAg correction factor is sufficient for achieving a good fit for all observations.

        To assess the potential of using observed lake volume time series as calibration target and not only for model evaluation, we also calibrated WGHM against RS observations and lake volume (RS_LV variant) and against RS, lake inflow and lake volume (RS_Q_LV variant). In the RS_LV variant, simulation of TWSA and GWSC did not change appreciably but

not only simulated lake volume anomaly but also simulated inflow into the lake greatly improved as compared to the RS variant. NSE for monthly lake volume anomaly and annual lake inflow reaches 0.95 and 0.44, respectively. Inflow into the lake is much less overestimated than in variant RS. To achieve these fits, the variant RS parameters where adjusted by increasing the rooting depth multiplier to 2.5 and setting the potential evaporation multiplier to 2. Adding lake volume observations on top of lake inflow observations in RS_Q_LV variant leads to an improved fit to lake volume observations,

with NSE increasing from 0.81 to 0.95, but the fit of observed inflow into the lake slightly worsens from 0.88 to 0.85. In this variant, the RS_Q variant parameters were used, except the maximum active lake depth was set to 9 m and the potential evaporation multiplier to 2. We conclude that in the case of the end Lake Urmia, calibration against time series of lake volume anomalies could, in the absence of inflow data, help to improve simulation of inflow, while calibration against time series of inflow could, in the absence of lake volume observation, improve simulation of lake volume anomalies. Still, calibration to both observational data types leads to the best simulation of both annual lake inflow and lake volume anomalies. However, the groundwater storage dynamics could not be improved without calibration against groundwater level dynamics.

Finally, we found that calibration aimed at optimizing the five criteria CC, NSE, RMSE, RAE and KGE with respect to monthly time series of observed total, groundwater and lake storages, with almost similar achieved performance values (Table 4), does not necessarily lead to similar estimates of total and compartmental water losses over the whole time period 2003 to 2013. For example, variants RS and RS_LV have the same values for all five performance criteria (expect KGE with 0.1 difference) with respect to TWS (not shown) but TWS loss between 2003 and 2013 is simulated to be $11.15 \cdot 10^9$ m$^3$ and $7.86 \cdot 10^9$ m$^3$, respectively (Table 5). TWS loss according to variant RS_Q_GW_NA is, with $10.04 \cdot 10^9$ m$^3$, in between and quite different, even though NSE and KGE are only 0.04 and 0.06 better, respectively. We conclude that in the case of relevant trends, the calibration criteria should include minimization of the difference between observed and simulated trends.

**Table 5. Water loss in Lake Urmia basin between 2003 and 2013 as observed and simulated by the different calibrated WGHM variants.**

| | Water loss between 2003 and 2013 [$10^9$ m$^3$] (mean annual storage in 2003 minus mean annual storage in 2013) | | | | | | |
|---|---|---|---|---|---|---|---|
| | Observed | Standard | RS | RS_LV | RS_Q | RS_Q_LV | RS_Q_GW | RS_Q_GW_NA |
| Total | 9.9 | 3.62 | 11.15 | 7.86 | 12.20 | 8.24 | 9.78 | 10.04 |
| Groundwater | 1.8 | 0.17 | 0.11 | 0.06 | 0.02 | 0.03 | 2.68 | 2.52 |
| Soil water | N.A. | 0.15 | 0.15 | 0.20 | 0.29 | 0.24 | 0.25 | 0.23 |
| Lake water | 8.0 | 3.16 | 10.76 | 7.37 | 11.83 | 7.78 | 6.62 | 7.02 |

## 4.2 Comparison to human vs. climate contribution as determined in previous studies

In order to define the lake restoration program, it is vital to know which factors contribute how much to shrinkage of the lake. All previous studies (e.g. Hassanzadeh et al., 2012; AghaKouchak et al., 2015; Ghale et al., 2018; Chaudhari et al., 2018) agreed that shrinkage is caused by both climate variations and human activities, but there is no consensus about the relative contributions. For example, Chaudari et al. (2018) concluded that human-induced changes accounted for 86% of the lake volume decline during 1995-2010, while we determined the value of 40% for 2002-2013. According to our study, human water use was the reason for 41% inflow reduction into the lake during 2003-2013 which is similar to the values of Shadkam et al. (2016) for the years 2003-2009 (comp. their Figs. 8). Discrepancies are likely due to different analysis methods but different analysis periods, as well as different conceptualizations, make a direct comparison of the estimated relative contributions difficult.

While Ghale et al. (2018) seem to support the results of Chaudhari et al. (2018), as they state that 80% of drying of Lake Urmia is due to anthropogenic impacts during 1998-2010, there statistical analysis assumes that river inflow can be considered to reflect "anthropogenic impacts" while precipitation and evaporation changes reflect climatic variations while river inflow is in reality also affected by climate variations. Also using a statistical change point analysis and without modelling, Khazaei et al. (2019) stated that given the stable conditions of precipitation and temperature, climatic changes cannot explain the dramatic decline of the lake level. They did not use in-situ data (except lake water level data) for their analysis. Based on a analysis of Standardized Precipitation Index (SPI), a drought index, AghaKouchak et al. (2015) reported there was no significant trend in droughts over the basin during past three decades and concluded from this that human activities not climatic variations are the main reason lake shrinkage. Different from our study and the modelling studies of Shadkam et al. (2016) and Chaudhari et al. (2018), these three studies consider only the dynamics of monthly and annual precipitation, not taking into account the changes in the variability of daily precipitation. During the last three decades, there was a significant increase the frequency of daily precipitation of less than 5 mm and a significant decrease in the frequency of daily precipitation of 10-15 mm, suggesting a runoff reduction even in case of constant annual precipitation (Fig. 2 in Bavil et al., 2018). Hosseini-Moghari et al. (2018) showed that an increasing frequency of days with less than 5 mm precipitation in combination with decreasing monthly precipitation has lead to the observed reduced inflow into two dams in the Lake Urmia basin that are located downstream of areas with insignificant human water use. We conclude that analyses should be done at the daily time scale or smaller.

In addition, a comprehensive modeling approach is preferable that takes into account, for example, the impacts of changing temperatures on runoff and thus river inflow and on evapotranspiration of the lake itself. Such comprehensive modelling was done by Chaudhari et al. (2018) but their uncalibrated global hydrological model that represented the basin by 5-6 cells only was not able to simulate well the flows and storages in the basin. For example, annual inflow into the lake was estimated to be $3,700 \cdot 10^6$ m$^3$ in 2003 (their Fig. 8) while observed inflow was much higher, $5,835 \cdot 10^6$ m$^3$. In 2009, observed inflow, with $1,036 \cdot 10^6$ m$^3$, was only half of the simulated one. Therefore, the very high human contribution to lake volume decline of 86% determined by Chaudhari et al. (2018) may arise from the poor performance of the uncalibrated model.

### 4.3 Limitations

Even after multi-objective calibration of a state-of-the-art comprehensive hydrological model, there remain many uncertainties that affect the accuracy of the model results. Like the results of all hydrological models, our results are affected by uncertainties in model input, model parameters, and model structure. Model parameter uncertainty was reduced by the comprehensive multi-observation calibration, albeit conditioned on just one climate input data set and using just one model (instead of the state-of-the-art multi-model ensemble approach, compare www.isimip.org, last accessed: 14 Dec. 2018). Given the low spatial model resolution (0.5°×0.5°), the model results are only valid for the basin as a whole and results for individual grid cells are very uncertain. Also due to a lack of data at the basin scale, the hydrogeology of the basin was not taken into account in the model. Information on irrigated area in each grid cell was taken from a global data set of areas equipped for irrigation from

groundwater and surface water (Siebert et al., 2010), which was adapted in this study by scaling it by basin-wide correction factors to better capture the temporal development of irrigation. Calibrated modeling results are also affected by uncertainties of the observation data. GRACE TWSA data are more reliable for larger (100,000 km$^2$ (Landerer and Swenson, 2012)) areas than the basin area. Estimation of groundwater storage changes based on water level data for unevenly distributed wells is rather uncertain due to the unknown heterogeneities in the subsurface. Evaluation results, here the good fit of simulated to "observed" lake water volume decline, are be affected by a likely underestimation of the actual decline by the "observed" value derived from remote sensing of lake water level elevation and lake water area by Tourian et al. (2015) assuming a constant bathymetry. However, there was an increase in the elevation of the lake bottom due to sedimentation and salt precipitation (Shadkam et al., 2016) so that the "observed" water volume decline was likely lower than the actual one, and our model would underestimate the lake storage decline, too.

## 5 Conclusions

This study investigated the differential impact of human water use and climate variations on water storage (total, groundwater, lake) in the Lake Urmia basin as well as on inflow into the lake during 2003-2013. This was done by utilizing the information contained in multiple types of observation data to calibrate, specifically for the Lake Urmia basin, the global hydrological model WGHM that takes into account the impact of human water use and man-made reservoirs on flows and storages. Using the best-performing model variant, the impact of human water use was determined by comparing the output of a naturalized run, where human water use was assumed to be zero, with the run with the historic water use. To understand the value of different observational data types for calibration, four calibration variants were defined where, in a step-wise fashion, basin-wide averages of 1) remote sensing data (for irrigated area and TWSA), 2) in-situ streamflow observations (for of lake inflow), 3) groundwater well data (groundwater level and storage), and 4) statistical data on water withdrawals in the basin were added. A time series of observed lake volume was used for evaluation.

We found that the time series for water demand by irrigation, as assumed in the standard WGHM version, had to be adjusted using MODIS data such that the modification of four model parameters could result in a good fit to observed TWSA. Consideration of these remote sensing data somewhat improved the dynamics of both inflow into Lake Urmia and lake water storage, inflow into the lake was still strongly overestimated by a factor of 0.92%, and groundwater dynamics should a strongly shifted seasonality. Additional calibration against observed inflow into the lake did not affect TWSA simulation and slightly improved the simulation of the lake water storage anomaly. Only by using monthly time series of mean groundwater level variations in the basins for calibration, we could adjust the fractions of human water use taken from groundwater and surface water such that seasonality of groundwater storage was simulated correctly. Only then it was possible to simulate the observed groundwater loss, and loss of lake volume was no longer overestimated. Statistical information on sectoral water withdrawals in the basin for one year as well as estimates for sectoral return flow fractions further improved the model, but only slightly.

We recommend to include, in case of relevant trends in observations, the difference between observed and simulated trends as one of the calibration criteria, not only differences between time series of daily, monthly or annual values.

The calibration exercise showed that the calibration variant for which the highest number of observational data types were used, WGHM variant RS_Q_GW_NA, showed the best fit to all observations. Certainly, no general conclusions on the worth of specific observation data types for model calibration, including trade-offs among fit to multiple data types, can be derived from this study. Lake Urmia basin is particular with respect to 1) draining into a large end lake that dominates TWSA, 2) the strong impact of human water use and 3) the fact that the standard WGHM version estimates a net recharge to the groundwater due to surface water irrigation, which had to be corrected to a net abstraction. In basins with large lakes, and in particular with end lakes, remotely sensed time series on lake area and the elevation of the lake water table should be used to estimate time series of lake water storage as these observational data can be expected to be of high value for understanding the freshwater system by hydrological model calibration. Groundwater storage cannot be observed from space but relies on in-situ observations on groundwater heads in wells but, as in the case of Lake Urmia basin, such data can be crucial for a correct understanding of the freshwater system.

Based on the good fit of WGHM variant RS_Q_GW_NA to four types of observational data, we are confident that human water use reduced lake inflow that would have occurred without human water use during 2003-2013 by about 41%. About 52% of the total water storage loss in Lake Urmia basin and only 40% of lake water loss during this time period was due to human water use, and the 48% and 60%, respectively, to climate variations. 90% of groundwater storage loss is estimated to be caused by human water use but this value may be somewhat overestimated by WGHM because climate-driven loss under naturalized conditions may be underestimated due to the simplified representation of groundwater-surface water exchanges in the model.

GRACE TWSA data indicate an increasing trend in water storage in the basin during 2014-2017 due to both less water use due to water management (ULRP, 2015b) and the wet years 2015/2016. This trend is about half as strong as the decreasing trend during 2003-2013. Further strengthening of efforts for decreasing human water use in the basin should be undertaken, while at the same time, global-scale mitigation of climate change by reducing greenhouse gas emissions to prevent strong decreases of precipitation and runoff. Our study has shown that management of the Lake Urmia basin should be based on a comprehensive assessment of all water storages and flows in the basin, including human water uses of groundwater and surface water. We recommend refining the estimated net abstractions from surface water and groundwater by a basin-wide spatially explicit quantification not only of water abstractions but also return flows to groundwater and surface water.

**Data availability**

In-situ data from "Iran Water Resources Management Company" including groundwater levels, precipitation and temperature publicly are available upon request from the corresponding author. GRACE data is available through http://www2.csr.utexas.edu/grace/RL05_mascons.html (last accessed: 17 Jul. 2018). Lake water surface extents and water

levels are available at http://hydrosat.gis.uni-stuttgart.de/php/index.php (last accessed: 17 Jul. 2018). All simulation results are available in the supplement.

**Acknowledgments**

The authors are grateful to Urmia Lake Restoration Program and Iran Water Resources Management Company for providing in-situ data. They also acknowledge the Center for Space Research (CSR) and Potsdam Institute for Climate Impact Research that provide GRACE data and global climate data, respectively. The authors graciously acknowledge Felix T. Portmann and Hannes Müller Schmied from Goethe University Frankfurt for their valuable contributions throughout the WaterGAP calibration, and thank the reviewers for their comments and suggestions that helped to improve the paper.

**Appendix A: Simulated groundwater storage in individual grid cells**

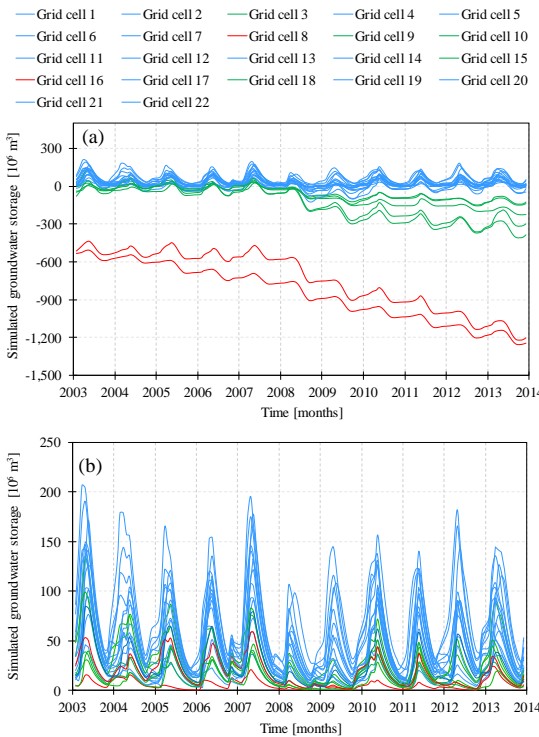

**Figure A1: Simulated groundwater storage in each of the 22 0.5° grid cells in Lake Urmia basin under anthropogenically altered (Fig. A1a) and naturalized conditions (Fig. A1b).**

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
