# Peer review of "Quantifying the impacts of human water use and climate variations on recent drying of Lake Urmia basin: the value of different sets of spaceborne and in-situ data for calibrating a hydrological model"

_Hydrology and Earth System Sciences, 2018_

## Short Comment (SC1) · 17 Sep 2018

There is a growing body of literature revolving around the human impacts of Urmia Lake. This study contributes to the ongoing debate on the shrinkage of the Urmia Lake and its ultimate fate. It assesses the impacts of climate and human water use on the desiccation of Lake Urmia in Iran using WaterGAP model. In particular, quantification of the anthropogenic effects reflected on the different parts of total water storage over

time gives a comprehensive picture of the changes in hydrology of the Lake Urmia basin caused by human interference. However, discussing and comparing results with more recent studies, such as Alizadeh et al., (2017, 2018), Chaudhari et al., (2018) and others, would add value to the manuscript. Here are some additional minor comments that the authors may want to consider during the revision.

1. The results from natural simulation in the manuscript shows a negative TWSA trend (Page 21, Lines 11-13), especially in 2009-2013; to what do the authors attribute this declining trend? Does any of the climate variables, such as precipitation and temperature, over the region show a similar declining trend? How much of the negative TWSA trend can be explained by the changes in climate variables? Even though the manuscript title says "climate variations", discussion regarding this part is currently too brief.

2.Figure 5 in the manuscript shows the WGHM grids. Significant area of the lake basin is excluded from the model domain. As the authors are estimating the total basin water storage change it is essential to encompass the entire basin.

3. The authors should use contrasting colors in figures. It is difficult to distinguish the WGHM-ANT and WGHM-NAT lines in Fig 9 due to similar colors.

References: Alizade, G. G. Y., Baykara, M. and Unal, A.: Analysis of decadal land cover changes and salinization in Urmia Lake Basin using remote sensing techniques, , (July), 1–15, 2017. Alizade, G. G. Y., Altunkaynak, A. and Unal, A.: Investigation Anthropogenic Impacts and Climate Factors on Drying up of Urmia Lake using Water Budget and Drought Analysis, Water Resour. Manag., 32(1), 325–337, doi:10.1007/s11269-017-1812-5, 2018. Chaudhari, S., Felfelani, F., Shin, S. and Pokhrel, Y.: Climate and anthropogenic contributions to the desiccation of the second largest saline lake in the twentieth century, J. Hydrol., 560, doi:10.1016/j.jhydrol.2018.03.034, 2018.

[Figure]

318, 2018.

---

## Referee Comment (RC1) · Anonymous Referee #1 · 11 Oct 2018

This manuscript uses WaterGAP Global Hydrology model to quantify the effects of human water use on inflow to Lake Urmia, lake water volume, and groundwater. The model was manually calibrated 4 for time using different observation data sets (remote sensing of irrigated area, monthly total water storage anomaly, insitu observations of stream flow, and groundwater levels from 284 wells). Strengths of the work include a focus on the pressing problem or Lake Urmia decline and identification of the effects on groundwater. With these strengths, there are also several issues that I feel need to

be addressed to accept this manuscript for publication.

1. Is the finding that humans affected lake decline new? There have been several recent studies that report this finding (Alborzi et al. 2018; Chaudhari et al. 2018; Shadkam et al. 2016) and some of these studied used the same model inputs as this work and also report groundwater changes. What is new in this work?

2. Is a global hydrologic model appropriate for a basin level analysis? The description of how the model simulates relevant processes is scant. Given resonance times, how appropriate is the temporal spacing (daily) relevant to the spatial grid size? Is it computationally efficient to run a global model for 15 or 20 grid cells of interest? There needs to be a much stronger justification for why the modeling and calibration methods are the correct approaches to use to answer the motivating questions.

3. There is a lot of focus in the text on the multiple calibration variants run with different input data sets. What was learned from this activity? How do those results effect Lake Urmia management?

4. Also, what could one potentially learn from 4 model calibrations that use different calibration data and yield four different models?

5. What are the limitations of this study?

6. The discussion of uncertainty in the results needs to go much deeper and be more specific. This uncertainty is real and likely plays a large role in the interpretation of the results.

7. I found the writing difficult to follow in numerous places, particularly the results section. There are lots of acronyms, run-on sentences, and text that digresses from the section headers or topic sentences of paragraphs. The writing here made it difficult for me to see the main results and findings of the work.

Overall I recommend decline for publication.

HESSD
Additional line-by-line comments include: pp. 2-4. The first three figures recount results from prior work. I would much prefer to see figures and tables focus on new insights gained from the work. For example, new figures that show uncertainties.

p. 3, line 5. I think "somewhat recovered" is overstated. Hard to tell from Figure 3. Maybe stabilized.

p. 3, line 18. Is the value -11.2 mm/yr correct? It seems incredibly small. In The Hashemite Kingdom of Jordan, drawdowns are 1 + m/yr, in numerous wells. In the U.S., we talk about drawdowns of ft/year.

p. 6, line 5. Only the anomalies? Or at all time periods? If the former, please explain what is meant by "anomaly", how determined, and why anomaly is the appropriate frame to discuss. I would want to calibrate a model across a range of conditions some of which might include anomalies.

p. 6, lines 18-28. I'm not familiar with WaterGAP. How does this model actually work? Explain.

p. 7. What is total water storage anomaly (TWSA)? This term seems rather central to the paper. Please explain.

p. 8, lines 13-18. This method of applying (1 - return flow multipliers) to the abstractions to estimate consumptive use assumes that water is used by only one water user. Is this a realistic assumption? If the return flow is used by another agricultural user and then again by a 3rd or 4th user, the basin-wide consumptive use fraction will be much different than the values reported. The large grid size magnifies this error. Table 1. How sensitive are study results to the values in this table?

p. 10. Lines 5-15. So the correction factors are needed because WaterGAP does not get the underlying physical hydrology correct? The correction is linear? Is the process causing the error also linear?

p. 11, line 1. Which parameters were varied to calibrate this model?

HESSD
p. 11, line 4. What is meant by optimal fit?

p. 15, line 2 and Table 3. Shouldn't these parameter values be the same across all the model variants? What is physically changing in the system that these parameter values would change across the model variants?

p. 16, lines 2-3. There could be a net groundwater abstraction but still areas where there is recharge. Is this an issue of coarse spatial resolution?

p. 18, lines 1-10. The discussion of uncertainty here is missing a fundamental point. Calibration can not help if the model structure is uncertain (or in error) or the temporal or spatial scaling of the model is mismatched to the modeled parameters of interest. This discussion also heads in a different direction than "what do we learn from the calibrations?" The text never explains what was learned. What was learned? Please discuss.

p. 18, lines 11-20. These statements are better placed in the introduction to justify the use of global hydrologic models. Still, why is a global model the appropriate choice when the domain of study is limited to one hydrologic basin (Urmia)?

p. 19, line 22. What beta?

p. 19, line 30. "much less overestimated" means what?

p. 19, line 31-32. This doesn't make sense to me. How come it is ok to change the parameter in one model variant but not others?

p. 20, lines 1-5. I would expect to see better calibration with more observational data (i.e., streamflows and lake levels).

p. 20, line 22. I'm confused. The scenario "with reservoirs but without human water use" does not fit either of the two scenarios described in the prior sentence.

p. 20, line 24. What is meant by anomalies? This term has still not been defined.

**HESSD**
p. 21, line 17, "The lower lake water loss...." What are the loss terms besides evaporation? How are these other loss terms smaller when inflow is larger? Explain.

p. 21, lines 20-23. I don't follow this explanation. There are too many NAs in this sentence. What causes the difference between the naturalized and anthropogenic scenarios?

p. 21, lines 25-28. I don't follow. What is the connection between the first part of the sentence and the second part?

p. 21, lines 28-32. Is a run-on sentence.

p. 21, lines 32 - 21. Put these ratios in context. What is desirable? Undesirable? What has implications for lake health? What values are acceptable?

p. 23, line 15. How can water storage be negative?

p. 23, line 18. This is an interesting result. It needs to be much more strongly emphasized. These cells are the locations where groundwater declines and there could be problems.

p. 23, lines 21-23. This qualification and limitation seems rather important. Why should the model results be trusted or used if the model does not get groundwater storage correct?

p. 23, lines 23-25. Run-on sentence. What is meant by the clause with maximum?

p. 24, lines 18-19. Is this result surprising? More calibration data means a better fit model. How does this result improve understanding of the Lake Urmia system?

p. 24, lines 25-28. Is this finding new? If so how? I feel the Urmia Lake Recovery Program has been working under the assumption that agricultural water use was a large contributor to lake decline and that they have been taking steps in recent years to address.

**HESSD**
p. 24, line 29. 90% of what?

p. 24, lines 19-24. I disagree. There are lots of other similar systems in the world – Great Salt Lake, Owens Lake, Dead Sea, Ural Sea, etc. each satisfy the first two criteria listed. What of these results is generalizable?

p. 24, lines 31-34. How do the model results inform the 2014-2017 trends? Also, how can climate change be constrained in this basin? Explain.

Figure 9. What is being shown in panels A, B, and D? The y-axis labels were mentioned in the text but never explained.

Figure A1a. The color scheme makes it difficult to differentiate grid cells. Use only three colors to differentiate the 3 types of storage. How can storage volume be negative? Data availability. I don't follow. If the authors do not have permission to share the data, then how can they share by author request? The HydroSat site underwritten by the University of Stuttgart is neat. What is the original source data for Urmia? Also, there is no water storage anomaly data for Urmia.

References Alborzi, A., Mirchi, A., Moftakhari, H., Mallakpour, I., Alian, S., Nazemi, A., Hassanzadeh, E., Mazdiyasni, O., Ashraf, S., Madani, K., Norouzi, H., Azarderakhsh, M., Mehran, A., Sadegh, M., Castelletti, A., and AghaKouchak, A. (2018). "Climate-informed environmental inflows to revive a drying lake facing meteorological and anthropogenic droughts." Environmental Research Letters, 13(8), 084010. http://stacks.iop.org/1748-9326/13/i=8/a=084010. Chaudhari, S., Felfelani, F., Shin, S., and Pokhrel, Y. (2018). "Climate and anthropogenic contributions to the desiccation of the second largest saline lake in the twentieth century." Journal of Hydrology, 560, 342-353. http://www.sciencedirect.com/science/article/pii/S0022169418302002. Shadkam, S., Ludwig, F., van Oel, P., Kirmit, Ç., and Kabat, P. (2016). "Impacts of climate change and water resources development on the declining inflow into Iran's Urmia Lake." Journal of Great Lakes Research, 42(5), 942-952. http://www.sciencedirect.com/science/article/pii/S0380133016301307.

---

## Author Comment (AC1) · 12 Oct 2018

Dear S. Chaudhari,

We would like to thank you for your interest on our manuscript. We will try to do our best to consider all your recommendations in the revised version. Below, we have provided a point-by-point response to your comments.

[Figure]

Comment#1: The results from natural simulation in the manuscript shows a negative TWSA trend (Page 21, Lines 11-13), especially in 2009-2013; to what do the authors attribute this declining trend? Does any of the climate variables, such as precipitation and temperature, over the region show a similar declining trend? How much of the negative TWSA trend can be explained by the changes in climate variables? Even though the manuscript title says "climate variations", discussion regarding this part is currently too brief.

Response: In the text, whenever we refer to "natural simulation" (naturalized conditions, WGHM-NAT), the simulation results are driven by climate only (in the form of daily time series of precipitation, temperature as well as shortwave and longwave down radiation) and not be human water use. So, the negative TWSA trend which you mentioned is explained totally by climate variables. The average of precipitation values in 2009-2013 was about 7.5% less than the average of precipitation values 2003-2007. About the temperature, the average of temperature in 2003-2007 and 2009-2013 were almost the same. Actually, the strong negative trend occurred in 2008 and 2009 (comp. Fig. 9) and average of precipitation during 2008-2009 was 19% less than during 2003-2007. In the revised version, we will add more information on the climate variations and also refer to the study of Bavil et al. (2018) over the Lake Urmia basin.

Reference: Bavil, S. S., Zeinalzadeh, K., and Hessari, B.: The changes in the frequency of daily precipitation in Urmia Lake basin, Iran, Theoretical and Applied Climatology, 133(1-2), 205-214, doi: 10.1007/s00704-017-2177-7, 2018.

Comment#2: Figure 5 in the manuscript shows the WGHM grids. Significant area of the lake basin is excluded from the model domain. As the authors are estimating the total basin water storage change it is essential to encompass the entire basin.

Response: We believe our procedure works well. One should not encompass much more than the entire basin area, to avoid, for example, that the precipitation input into the basin and Lake Urmia is overestimated. Given the 0.5° resolution of hydrological

model and the climate input data, the fit is quite good. We quantified the area outside the basin that is covered by the 0.5° grid cells and compared to the area of the basin that is not covered by the included grid cells; the basin area in WGHM model is only 4% larger than the actual area of the basin.

Comment#3: The authors should use contrasting colors in figures. It is difficult to distinguish the WGHM-ANT and WGHM-NAT lines in Fig 9 due to similar colors.

Response: We will change the color of WGHM-NAT in the revised version.

\*\*\*\*\*\*\*\*\*\*\*\*\*\*\*\*\*\*\*\*\*\*\*\*\*\*\*\*\*\*\*\*\*\*\*\*\*\*\*\*\*\*\*\*\*\*\*\*\*\*\*\*\*\*\*\*\*\*\*\*\*\*\*\*\*\*\*\*\*\*\*\*\*\*\*\*\*\*

Thank you very much again for your time and for providing valuable comments.

---

## Referee Comment (RC2) · Anonymous Referee #2 · 15 Oct 2018

It can be considerd as an intresting update in vast literature of Lake Urmia studies while authors tried to use a vast variety of data between 2003 to 2013 to evaluate the situation of Lake Urmia. I think a Major revisions are needed prior to evaluating its technical quality. My comments are listed as bellow

General comments: 1- A technical proof reading is needed since some of the sentences are not understandable 2- Your given figures and tables do not necessarily

indicate to the discussion you have made 3- Relative error in your models is important since figures are dimensionless.Still, given figures seems to have unacceptable errors 4- Methodology should be revised since it is not clear how you evaluated the figures in discussion

Specific comments: 1- You are suggesting that the Lake would have been vanished any way but there are a vast literature against your statement. What is your comments? You should also add sentences in the text about it 2- One of the main factors in your study is the calibration of satelite data and application of filters on data which are main issues. E.g. In GRACE data how did you manage to use 2 degree precision into such a cristal clear results? 3- Why did you use a time length between 2003 to 2013? Since the decreasing trend have already started from late 90's. 4- There are a lot of missing data in historical time series records of the region. How did you manage to remove them? are they satisfactorily acceptable methods to be applied? 5- Page 9, Line 20, Calibration: You have to give the error evaluation if you have used "try and error" method 6-Page 12, Performance criteria, Line 5:These criteria do not show relative error in your models since RMSE number is not necessarily satisfactory 7- Page 13, Figure 7: None of these figures indicate to an acceptable calibration 8- Page 14, Figure 8d: The discrepancy and error is growing in your anomalies. How do you interpret? 9- Page 18-23: Your discussions are too long, yet non of them are visuable from given tables and figures. It is a very long article, yet given informations are narrative and reader should accept your sentences without having a chance to approve it.

---

## Short Comment (SC2) · 3 Nov 2018

**Overview**

The present study aims to quantify (estimate) the impact of human water consumption—as for irrigation, livestock, domestic, manufacturing, and thermal energy production—versus (natural) climatic variability on the water balance and storage of the Lake Urmia (LU) basin and consequently the lake desiccation during the

past decades. This is indeed a curious question with high practical relevance, given the ongoing drying of the lake and scientific debates around possible causes and viable remedies. One of the strength of the study is incorporating multiple input data (both ground and remote sensing) in developing the basin's hydrologic model. The authors have also attempted to include the groundwater data which is highly important in this basin, and has been ignored in many (not all) of the previous studies. I enjoyed reading the paper, however, as the other reviewers have already pointed out there are major shortcomings that call for a major revision. In the spirit of helping the authors to improve the manuscript, I'd like to further comment on a number of—I believe—major deficiencies and questionable assumptions of the study that undermine the reliability of their results and discussion, given my own (limited) knowledge/experience in studying the lake's dynamics and desiccation [Khatami, 2013; Khatami and Berndtsson, 2013; Khazaei et al., in review]. I hope the authors would find my comments useful in highlighting the new insights and contribution of their study. :-)

My comments are cross-referenced against the manuscript's page (P) and line (L) numbers as P x L y.

**The time period of the analysis**
In their analysis, authors only considered the time period 2003-2013. It is not clear why both earlier years and the most recent years are excluded from the analysis. It is well-known that there has been significant changes in the basin and the lake water level since late 1990s. Using a statistical change point analysis, Khazaei et al. [in review] identified the year 2000 as the beginning of the period with significant changes in the lake dynamics. So, it is crucial to include all the years post 2000—the data availability is not an issue as ULRP now provides researchers with required data.

For instance, annual inflows to the lake are highly variable during this period (see the supplement figure), which you also used for calibrating the model. Given such degree of variability, the modelling results are likely sensitive to this variable, and it is important to use the entire time period.

[Figure]

**Irrigated area**

You have assumed that the irrigated area in 2013 remains the same as 2012. This assumption is wrong and known to be wrong, as we know that the irrigated area has increased since 2012. Chaudhari et al. [2018] estimated the cropland area of the basin using both MODIS and HYDE 3.1 products (also notice the difference between these two products on Fig 3). Land use change is a major driver in this basin. Expansion of the irrigated area has increased the evapotranspiration losses, and consequently lead to less runoff in the basin and in turn less inflows to the lake. Therefore, I expect a useful model to be highly sensitivity to this input, and thus improper handling of this input data could be major source of uncertainty.

**Groundwater**

It is not clear how representative the 248 groundwater (GW) wells data (section 2.23) are for the deep GW withdrawals in the basin. Failing to include deep GW abstraction can bias your results underestimating both groundwater and net abstractions in the basin (Figure 7), which seems to be the case. ULRP [2015] estimated the total amount of groundwater extraction in the basin, in 2013-2014, around 2,200 MCM, of which around 900 MCM were extraction from shallow groundwater and about 1000 MCM from deep groundwater. In your results, the estimated GW abstraction, for the same year, for the most input-comprehensive variant of RS_Q_GW_NA are around 1,200 MCM—well below the actual estimation. The groundwater extraction in the basin has also had an enormous impact on the inflowing runoff to the lake as well.

**Water consumption**

Lake Urmia is a highly regulated basin, so embedding water withdrawal/consumption in the basin's model is crucial. In doing so, however, you have used the water withdrawal records only for the year 2009. There is a high (year to year) variability in

water withdrawal/consumption in the basin, implicitly indicated by the high variability of the inflow to lake, as presented in your Figures 7 and 8. Year 2009 has one of the least amount of inflows to the lake (Figure 7) implying possibly a high water consumption in the basin. That said, given the variability of water withdrawal/consumption in the basin, including a single year is not sufficient and could lead to significant bias/uncertainty in your modelling results.

Further, in using water withdrawal/consumption records be aware of possible inconsistencies between the definition and estimation of water withdrawal/consumption/use by different water agencies/authorities or studies, and the related metrics or model fluxes. Such inconsistencies could lead to methodological fallacies and unreliability of the analysis results [Madani and Khatami, 2015].

**Model calibration and over-parametrisation**

Based on table 2, in the variant RS_W_GW the parameters $\alpha$ and $\beta$ vary between 0.45 to 0.47 and 0.47 to 0.52, respectively. That said, by introducing the NA data into the variant RS_Q_GW_NA the parameter ranges expand significantly to 0.29 to 0.56 for $\beta$ and 0.39 to 0.63 for $\beta$. This implies that the model is not benefiting from the additional information content. For instance, there is no significant improvement from RS_W_GW to RS_W_GW_NA evidenced by Figure 8 and metric values on table 4; for years 2003, 2004, 2007, and 2009 the RS_Q_GW variant is even better than the RS_Q_GW_NA in terms of estimating the annual inflow to the lake (Figure 8b). That is, the model is insensitive to adding new data NA. Instead the new information is compensated by the model parameters. It is an indication of over-parameterisation in your model, which is expected for annual multipliers, that undermines the reliability of the results. While I understand the rationale behind a year-specific parameter value, it is a serious issue. Instead, you should do a more efficient and effective parameter search (instead of manual calibration) finding one or (more desirably) a number of acceptable parameter sets. Then to ensure their reliability you can do a year-by-year evaluation of the model performance, i.e. evaluating the model performance against a

given metric for each year separately.

Also, it would be helpful to include a schematic of the model structure demonstrating its fluxes, storages, and their interconnection. This can help the reader to better understand the mechanism and process-representation of the model.

**Model calibration/evaluation and performance metrics**

It is a well-established fact that CC is an inadequate measure for model evaluation [Willmott, 1981]. It is especially redundant to use CC together with NSE, as CC is already included in the NSE metric; see the NSE decomposition by Murphy [1988] and Gupta et al. [2009]. Further, NSE puts more emphasis on the larger values, e.g. as Pushpalatha et al. [2012] showed it focuses on the top 20% of discharge flows. Therefore, calibration by NSE introduces bias. So, it's more useful to combine NSE with bias rather than RMSE. Furthermore, other metrics such as Willmott's refined index of agreement [Willmott et al., 2012] and KGE [Gupta et al., 2009] shown to be better than NSE.

Also, on P 5 L 25-29, you already explained that the standard WGHM is not calibrated for LU basin. Yet you reported the results of the standard model on Figures 7-8, and discussed it through the results and discussion. Including the standard variant in your results and discussion does not serve the manuscript any benefit other than adding to its bulk.

Further, it is also a well-known fact that (hydrologic) model cannot be validated [Konikow and Bredehoeft, 1992; Oreskes et al., 1994]. Therefore, using the term validation is both semantically and theoretically wrong. As a matter of good practice, it's been recommended to use the term evaluation instead of validation [Beven and Young, 2013]. Same comment applies to terms such as optimal values and optimal fit throughout the manuscript—there is no optimal set.

**Model evaluation and the role of uncertainties**

In this study you have not investigated the model parameter space other than four calibration variants, while the calibration is done manually. So, first, there is no way to justify that the manually calibrated parameter values are the best-performing calibrations (despite what you said e.g. on P1 L16). Further, no sensitivity nor uncertainty analysis is performed which is nowadays a requirement for publishing a modelling analysis in the hydrologic community. Without any sensitivity/uncertainty analysis the reliability of the modelling results is questionable.

Other than the model structure and parameters, the role of data uncertainty is important and should be discussed. For instance, the annual inflows are not the exact inflows to the lake. They are estimates of the last station, sometimes as far as 50 km from the lake.

**Discussion of modelling results and equifinality**

There are fundamental issues in the discussion within the first paragraph of section 3.2. First, adding a new input data into model calibration/evaluation does not necessarily decrease the modelling uncertainty. In fact, by adding each new data you're introducing a new source of uncertainty as there is also uncertainty associated with data themselves; especially in a case like LU basin where the data uncertainty is very high both for ground and remote sensing data.

The model parameter equifinality is not necessarily a problem [Savenije, 2001], in fact it can help us to improve our modelling in the face of uncertainty [Beven, 2009]. As all (hydrologic) models are wrong [Box, 1976], model ensembles—as multiple working hypotheses—are better suited than calibrated model with a single parameter set to describe/predict hydrologic systems given the uncertainties [Beven, 2012; Beven et al., 2012; Chamberlin, 1890; Clark et al., 2011; 2012].

Despite your statement, parameter equifinality does not ask for additional data! Adding data, in fact, may even exacerbate the model parameter equifinality, and one cannot make up for parameter equifinality by adjusting the parameter values. For instance, the model structure may not be able to benefit from the additional information content,

and therefore the new data is redundant (which seems to be the case comparing the results of variants RS_Q_GW and RS_Q_GQ_NA). What parameter equifinality implies is a more thorough search of the parameter and model space as well as more rigorous model evaluation schemes such as limits of acceptability approach [Beven and Binley, 2014]; even then the parameter equifinality will remain. In other words, each model variant is prone to model parameter equifinality.

Further, parts of the discussion in this section are not new lessons (e.g. limitations of global hydrologic models in paragraph 2) and are well-established in the literature. Also, section 3 is very long, sometimes discussing too much details. I think it'd serve your discussion better to provide the top 3-5 main learned lessons as bullet points early on in the section. Then briefly explain each bullet point. The rest, especially modelling technicalities which may be valuable particularly for the reproducibility of the study, could be provided as supplement. In doing so, it's particularly helpful to restructure the result discussion by, first, explicitly discussing the limitations and uncertainties associated with the modelling design and consequent results. Second, discuss what results are therefore more/less reliable given the uncertainties. Third, what are the new findings of your studies and where do the results lie within the extant litertature. Finally, wat are the organic future research questions/directions based on what you learned and what lacking in the current literature.

**On the role of human activities, climatic changes, and drought**
On P 2 L 12-16 you stated that the decreasing trend in precipitation (P) and increasing trend of temperature (T), and thus increased evaporation, has very likely to contributed to the decrease in the lake volume. This is also reported as one of the main reasons for lake degradation on P3 L12. This statement is debatable. First, in our recent analysis [Khazaei et al., in review], we showed that the decrease in P and increase of T is not considerable in explaining the shrinkage of the lake; nor the decrease in T can be associated directly with an increase in lake evaporation. The major driver of the basin, as stated before, is the land use change and the substantial expansion

of cropland areas. This has led to increase in the irrigation hence less available runoff as for the lake inflow, and also caused a major increase in evapotranspiration. The following sentence (last sentence of the paragraph) does not explicitly indicate the greater role of human activities in the lake desiccation compared to atmospheric climate change, which is the common finding of the most of the studies in this area [AghaKouchak et al., 2015; Aneseh et al., 2018; Stone, 2015; Torabi Haghighi et al., 2018; Vaheddoost and Aksoy, 2018].

On P 24 L 34 you concluded that *"climate change must be constrained to prevent strong decreases of precipitation and runoff"*. It is not clear to me what you mean by constraining climate change here. Also, as discuss previously the role of human activities are more substantial in the lake's fate than atmospheric changes.

Further, on P 3 L 2-3, you have discussed the role of drought in the lake's water level decline. First, the term drought is ambiguous, and it should be further specified what type of drought is discussed; atmospheric, hydrologic, agricultural, ecologic, or anthropogenic. Second, the analysis by AghaKouchak et al. [2015] indicated no considerable trend in droughts, at 0.05 significance level, during the past three decades. They argued that the region has undergone even more severe multi-year droughts in the past that did not cause a major change in the lake's surface area. They, therefore, cautioned against overrating the role of drought on the drying of the lake and disruption of its water balance.

This is a technical note: you have used CC and NSE (P 9 L 11-12) to cross compare precipitation and temperature records of difference sources. First, I assume by CC you meant Pearson CC, which should be explicitly mentioned. Second, both CC and NSE are sensitive measures, i.e. a few number of large outliers can significantly change their values; especially for skewed distributions. It is better to use (more) resistant alternatives such as Spearman ranked correlation (instead of Pearson correlation) and Willmott's refined index of agreement [Willmott et al., 2012] (or ideally normalised the data using a transformation such as Box-Cox, first, and then compare the time series distance).

**Other comments**

P 2 L 7: this is an unsubstantiated claim. As far as I know there is no (reliable) evidence on the degree of awareness regarding this issue. Please remove it, or provide the evidence.

P 25 L 9: It is better to explicitly acknowledge the organisations that provided you with GRACE and climate data, and the URL links if available.

P 28-29: The URLs in the ULRP references are not accurate. Please update them.

**References**

AghaKouchak, A., H. Norouzi, K. Madani, A. Mirchi, M. Azarderakhsh, A. Nazemi, N. Nasrollahi, A. Farahmand, A. Mehran, and E. Hasanzadeh (2015), Aral Sea syndrome desiccates Lake Urmia: Call for action, Journal of Great Lakes Research, 41(1), 307-311.

Aneseh, A., et al. (2018), Climate-informed environmental inflows to revive a drying lake facing meteorological and anthropogenic droughts, Environmental Research Letters, 13(8), 084010.

Beven, K. (2009), Environmental modelling: an uncertain future?, CRC Press.

Beven, K. (2012), Causal models as multiple working hypotheses about environmental processes, Comptes rendus geoscience, 344(2), 77-88.

Beven, K., and P. Young (2013), A guide to good practice in modeling semantics for authors and referees, Water Resources Research, 49(8), 5092-5098.

Beven, K., and A. Binley (2014), GLUE: 20'years on, Hydrological Processes, 28(24), 5897-5918.

Beven, K., P. Smith, I. Westerberg, and J. Freer (2012), Comment on "Pursuing the method of multiple working hypotheses for hydrological modeling" by P. Clark et al, Water Resources Research, 48(11), W11801.

Box, G. E. P. (1976), Science and Statistics, Journal of the American Statistical Association, 71(356), 791-799.

Chamberlin, T. C. (1890), The method of multiple working hypotheses, Science, 15, 92-96.

Chaudhari, S., F. Felfelani, S. Shin, and Y. Pokhrel (2018), Climate and anthropogenic contributions to the desiccation of the second largest saline lake in the twentieth century, Journal of Hydrology, 560, 342-353.

Clark, M. P., D. Kavetski, and F. Fenicia (2011), Pursuing the method of multiple working hypotheses for hydrological modeling, Water Resources Research, 47(9).

Clark, M. P., D. Kavetski, and F. Fenicia (2012), Reply to comment by K. Beven et al. on "Pursuing the method of multiple working hypotheses for hydrological modeling", Water Resources Research, 48(11).

Gupta, H. V., H. Kling, K. K. Yilmaz, and G. F. Martinez (2009), Decomposition of the mean squared error and NSE performance criteria: Implications for improving hydrological modelling, Journal of Hydrology, 377(1–2), 80-91.

Khatami, S. (2013), Nonlinear Chaotic and Trend Analyses of Water Level at Urmia Lake, Iran, M.Sc. Thesis report: TVVR 13/5012, ISSN:1101-9824, Lund: Lund University.

Khatami, S., and R. Berndtsson (2013), Urmia Lake Watershed Restoration in Iran: Short-and Long-Term Perspectives, in Proceedings of the 6th International Perspective on Water Resources  the Environment (IPWE) edited, ASCE/EWRI, Izmir, Turkey.

Khazaei, B., S. Khatami, S. H. Alemohammad, L. Rashidi, C. Wu, K. Madani, Z. Kalantari, G. Destouni, and A. Aghakouchak (in review), Climatic or regionally induced by humans? Tracing hydro-climatic and land-use changes to better understand the Lake Urmia tragedy, Journal of Hydrology.

Konikow, L. F., and J. D. Bredehoeft (1992), Ground-water models cannot be validated, Advances in Water Resources, 15(1), 75-83.

Madani, K., and S. Khatami (2015), Water for Energy: Inconsistent Assessment Standards and Inability to Judge Properly, Curr Sustainable Renewable Energy Rep, 2(1), 10-16.

Murphy, A. H. (1988), Skill Scores Based on the Mean Square Error and Their Relationships to the Correlation Coefficient, Monthly Weather Review, 116(12), 2417-2424.

Oreskes, N., K. Shrader-Frechette, and K. Belitz (1994), Verification, validation, and confirmation of numerical models in the earth sciences, Science, 263(5147), 641-646.

Pushpalatha, R., C. Perrin, N. L. Moine, and V. Andréassian (2012), A review of efficiency criteria suitable for evaluating low-flow simulations, Journal of Hydrology, 420-421, 171-182.

Savenije, H. H. G. (2001), Equifinality, a blessing in disguise?, Hydrological Processes, 15(14), 2835-2838.

Stone, R. (2015), Saving Iran's great salt lake, Science, 349(6252), 1044-1047.

Torabi Haghighi, A., N. Fazel, A. A. Hekmatzadeh, and B. Klöve (2018), Analysis of Effective Environmental Flow Release Strategies for Lake Urmia Restoration, Water Resources Management.

ULRP (2015), Groundwater and restoration of Lake Urmia, Urmia Lake Restoration Program, Sharif University of Technology.

Vaheddoost, B., and H. Aksoy (2018), Interaction of groundwater with Lake Urmia in Iran, Hydrological Processes, 32(21), 3283-3295.

Willmott, C. J. (1981), On the validation of models, Physical Geography, 2(2), 184-194.

Willmott, C. J., S. M. Robeson, and K. Matsuura (2012), A refined index of model performance, International Journal of Climatology, 32(13), 2088-2094.
* * *
[Figure]

**Fig. 1.** Annual inflow to the lake

---

## Author Comment (AC3) · 17 Dec 2018

Referee #2: It can be considered as an interesting update in vast literature of Lake Urmia studies while authors tried to use a vast variety of data between 2003 to 2013 to evaluate the situation of Lake Urmia. I think a Major revision are needed prior to evaluating its technical quality. My comments are listed as bellow.

Response: We would like to thank you for the thorough consideration and critical comments that helped us improving the manuscript. We will try to do our best to consider all your recommendations in the revised version. Below, we have provided a point-by-point response to your comments.

General comments:

Referee #2: A technical proof reading is needed since some of the sentences are not understandable.

Response: We will revise the whole manuscript and the manuscript will be checked by a native speaker.

Referee #2: Your given figures and tables do not necessarily indicate to the discussion you have made.

Response: We provided discussion on all figures and tables. We agree that some part of the discussion might not directly related to the figures and tables which is the lesson we learned from this study. However, we will re-write the result and discussion section with considering your comment.

Referee #2: Relative error in your models is important since figures are dimensionless. Still, given figures seems to have unacceptable errors.

Response: We can calculate relative error. But we think you were misled because all figures have dimension. We assume that you did not follow the different calibration variants. If you see some unacceptable errors in the figures, these errors are related to the variables which did not consider as an objective function in the calibration process (see different calibration variants).

Referee #2: Methodology should be revised since it is not clear how you evaluated the figures in discussion.

Response: We do not understand the comment well. However, we will revise the methodology section.

Specific comments:

Referee #2: You are suggesting that the Lake would have been vanished any way but there are a vast literature against your statement. What is your comments? You should also add sentences in the text about it.

Response: We do not suggest at all that the lake would have vanished without human water use. On page 21 line 28, for example, we state the following:

"Still, even in the WGHM-NAT, the average inflow into the lake from 2009-2013 of $3,670 \times 106$ m3 would have been only enough to keep the lake from further loosing volume but would not have been enough for a recovery to conditions between 2003 and 2007 (Fig. 9b),"

Referee #2: One of the main factors in your study is the calibration of satellite data and application of filters on data which are main issues. E.g. In GRACE data how did you manage to use 2degree precision into such a cristal clear results?

Response: We agree with you. We used satellite data for three objectives including the irrigated area, lake level (and extent) and TWSA. Irrigated area and lake level were taken from previous studies which they needed filters on data before application. Hence, among all used satellite data, here we only discuss GRACE data. To deal with your mentioned issue about GRACE data, we have used CSR mascon solutions product which based on Save et al. (2016) do not need any filters. Also, as we mentioned in the manuscript, we know that it is recommended GRACE data products only for areas with at least 100,000 km2 (Watkins et al., 2015; Landerer and Swenson, 2012). But studies by Tourian et al. (2015) and Lorenz et al. (2014) showed that signal strength or the so-called gravimetric resolution is determining the applicability of GRACE data. In fact, Lake Urmia basin has experienced an $8 \times 109$ m3 change in the water volume in the last decade, which allows the use of GRACE for monitoring the changes in water storage in the basin (Tourian et al., 2015). This fact is supported by the very small gain factor of 1.0083 for the Lake Urmia basin based on Community

Land Model 4 (CLM4) for spherical harmonic solutions (Landerer and Swenson, 2012), which is the factor with which signal attenuation due to leakage could be balanced. We can assume errors of the applied GRACE monthly time series of TWSA are small compared to the uncertainty of TWSA as computed by WGHM, such that model calibration against GRACE TWSA is meaningful.

Referee #2: Why did you use a time length between 2003 to 2013? Since the decreasing trend have already started from late 90's.

Response: The main reason was shortage of data. The GRACE data and irrigated area which play important roles in this study were not available for late 90's. As a results we faced substantial missing data before 2000.

Referee #2: There are a lot of missing data in historical time series records of the region. How did you manage to remove them? are they satisfactorily acceptable methods to be applied?

Response: We disagree. There are no significant missing data for 2003-2013. The method used for filling the gaps are as follows:

-Irrigated area: There is no missing data in this data (except 2013).

-GRACE data: There are some missing data (8 months) in its observations which have filled using linear interpolation.

-Inflow into the Lake: There is no missing data in the dataset of annual inflow into the lake.

-Groundwater levels: We assessed data of 635 wells then we removed the wells with more than 12 months or six consecutive months missing values. After removing the wells with significant missing values, we have worked on 284 wells. If there are missing data in the dataset of these 284 wells, we used linear interpolation (if only one month is missing) or linear regression (with the nearest well in common period) for filling the gaps in data.
[Figure]

-Water withdrawals: There is no missing data in this data.

-Precipitation: Almost there is no missing data in the studied stations between 2003-2013. Few missing data filled in comparison with near stations.

-Temperature: There is no missing data in this data.

-Lake volume: There are some missing data for lake volume which have filled using linear interpolation.

Referee #2: Page 9, Line 20, Calibration: You have to give the error evaluation if you have used "try and error" method.

Response: We used trial and error to determine the most appropriate parameters of model in each calibration of variant based on the evaluating the model error with respect to the observations used, while trying to keep the number of adapted parameters at a minimum. We provide the final errors in Table 4. We do not think that it is interesting for the reader to provide the errors/model performances for all trials (there were many).

Referee #2: Page 12, Performance criteria, Line 5: These criteria do not show relative error in your models since RMSE number is not necessarily satisfactory.

Response: We will calculate relative error in the revised version.

Referee #2: Page 13, Figure 7: None of these figures indicate to an acceptable calibration.

Response: This figure does not show any calibration results. This figure shows the inputs of model in different variants, not the model's output. Thus, there is no reason to fit the lines in this figure.

Referee #2: Page 14, Figure 8d: The discrepancy and error is growing in your anomalies. How do you interpret?

Response: It should be noted that Figure 8d shows change not anomalies. Anyway, we agree that the error in the second half of the time series is bigger than the first half. However, the difference in errors is minor. Also, discrepancies in groundwater storage (e.g. in peak of seasonality) can represent some minor discrepancy in groundwater storage changes. If you consider Figure 8c, which shows the normalized groundwater storage, there is no growing in discrepancy.

Referee #2: Page 18-23: Your discussions are too long, yet none of them are visuable from given tables and figures. It is a very long article, yet given informations are narriative and reader should accept your sentences without having a chance to approve it.

Response: We will re-write the results and discussions considering your comment.
* * *
Thank you very much again for your time and for providing valuable comments.
* * *
References

Bavil, S. S., Zeinalzadeh, K., and Hessari, B.: The changes in the frequency of daily precipitation in Urmia Lake basin, Iran, Theoretical and Applied Climatology, 133(1-2), 205-214, doi:10.1007/s00704-017-2177-7, 2018.

Farokhnia, A.: Impact of Land Use Change and Climate Variability on Urmia Basin Hydrology (Dissertation) Tarbiat Modares University, Agricultral department (in Persian), 2015.

Landerer, F. W., and Swenson, S. C.: Accuracy of scaled GRACE terrestrial water storage estimates, Water resources research, 48, doi:10.1029/2011WR011453, 2012.

Lorenz, C., Kunstmann, H., Devaraju, B., Tourian, M. J., Sneeuw, N., and Riegger, J.: Large-scale runoff from landmasses: a global assessment of the closure of the

hydrological and atmospheric water balances, Journal of Hydrometeorology, 15, 2111-2139, doi:10.1175/JHM-D-13-0157.1, 2014.

Save, H., Bettadpur, S., and Tapley, B. D.: High‐resolution CSR GRACE RL05 mascons, Journal of Geophysical Research: Solid Earth, 121, 7547-7569, doi:10.1002/2016JB013007, 2016.

Tourian, M. J., Elmi, O., Chen, Q., Devaraju, B., Roohi, S., and Sneeuw, N.: A space-borne multisensor approach to monitor the desiccation of Lake Urmia in Iran, Remote Sensing of Environment, 156, 349-360, doi:10.1016/j.rse.2014.10.006, 2015.

Watkins, M. M., Wiese, D. N., Yuan, D. N., Boening, C., and Landerer, F. W.: Improved methods for observing Earth's time variable mass distribution with GRACE using spherical cap mascons, Journal of Geophysical Research: Solid Earth, 120, 2648-2671, doi:10.1002/2014JB011547, 2015.

---

## Author Comment (AC4) · 17 Dec 2018

**Seyed-Mohammad Hosseini-Moghari et al.**

hosseini\_sm@ut.ac.ir

Received and published: 17 December 2018

Dear S. Khatami,

We would like to thank you for your interest on our manuscript and your efforts to help us improve the manuscript. Below, we have provided a point-by-point response to your comments.

Comment#1: In their analysis, authors only considered the time period 2003-2013. It is not clear why both earlier years and the most recent years are excluded from the analysis. It is well-known that there has been significant changes in the basin and the lake water level since late 1990s. Using a statistical change point analysis, Khazaei et al. [in review] identified the year 2000 as the beginning of the period with significant changes in the lake dynamics. So, it is crucial to include all the years post 2000—the data availability is not an issue as ULRP now provides researchers with required data. For instance, annual inflows to the lake are highly variable during this period (see the supplement figure), which you also used for calibrating the model. Given such degree of variability, the modelling results are likely sensitive to this variable, and it is important to use the entire time period.

Response: Data required for our analysis, except the data you mentioned, is not available before 2000. For example, GRACE satellite was only launched in 2002, and irrigated area reported by ULRP starts from 2001 and ends in 2012 (see Figure 16 in Kamali et al. (2015)). In addition, there are many gaps in groundwater data before 2000. If we want to work only on long term inflow or estimated water use based on empirical equations, we would have repeated the works were done by Shadkam et al. (2016) and Chaudhari et al. (2018), respectively. Furthermore, Iran Water Resources Management Company release data are available with two/three years' delay. Thus, in addition to irrigated area, in situ groundwater data precipitation are not available for the recent years. To sum up, be sure to perform a multi-objective calibration, we consider maximum time interval.

Comment#2: You have assumed that the irrigated area in 2013 remains the same as 2012. This assumption is wrong and known to be wrong, as we know that the irrigated area has increased since 2012. Chaudhari et al. [2018] estimated the cropland area of the basin using both MODIS and HYDE 3.1 products (also notice the difference between these two products on Fig 3). Land use change is a major driver in this basin. Expansion of the irrigated area has increased the evapotranspiration losses, and con-
sequently lead to less runoff in the basin and in turn less inflows to the lake. Therefore, I expect a useful model to be highly sensitivity to this input, and thus improper handling of this input data could be major source of uncertainty.

Response: Considering Figure 3 in Chaudhari et al. (2018), the cropland area from 2011 to 2013 did not increase everywhere increased but also decreased partially (see south part of the basin). This agrees with our data source.

Comment#3: It is not clear how representative the 248 groundwater (GW) wells data (section 2.23) are for the deep GW withdrawals in the basin. Failing to include deep GW abstraction can bias your results underestimating both groundwater and net abstractions in the basin (Figure 7), which seems to be the case. ULRP [2015] estimated the total amount of groundwater extraction in the basin, in 2013-2014, round 2,200 MCM, of which around 900 MCM were extraction from shallow groundwater and about 1000 MCM from deep groundwater. In your results, the estimated GW abstraction, for the same year, for the most input-comprehensive variant of RS\_Q\_GW\_NA are around 1,200 MCM—well below the actual estimation. The groundwater extraction in the basin has also had an enormous impact on the inflowing runoff to the lake as well.

Response: Unfortunately, we cannot find the reference you mentioned. Groundwater abstractions in WGHM are not differentiated between deep and shallow. Anyway, Figure 7 does not show total abstraction but as also described in the text, net abstraction, i.e. groundwater abstraction minus return flow from irrigation to groundwater. For example, based on Table 1, in 2009 abstraction from groundwater was about 2092 MCM (see Table 1) but net abstraction in 2009 is about 1300 MCM (see Figure 7). Hence we did not underestimate groundwater withdrawals if you consider the difference between abstraction and net abstraction.

Comment#4: Lake Urmia is a highly regulated basin, so embedding water withdrawal/consumption in the basin's model is crucial. In doing so, however, you have used the water withdrawal records only for the year 2009. There is a high (year to Interactive comment
year) variability in water withdrawal/consumption in the basin, implicitly indicated by the high variability of the inflow to lake, as presented in your Figures 7 and 8. Year 2009 has one of the least amount of inflows to the lake (Figure 7) implying possibly a high water consumption in the basin. That said, given the variability of water withdrawal/consumption in the basin, including a single year is not sufficient and could lead to significant bias/uncertainty in your modelling results. Further, in using water withdrawal/consumption records be aware of possible inconsistencies between the definition and estimation of water withdrawal/consumption/use by different water agencies/authorities or studies, and the related metrics or model fluxes. Such inconsistencies could lead to methodological fallacies and unreliability of the analysis results [Madani and Khatami, 2015].

Response: In WaterGAP, water consumption, abstraction and net abstraction are computed as a monthly time series taking into account, in case of irrigation, time series of climate forcings, and irrigated area. To improve this estimation, we included basinspecific data on the temporal development of irrigated area (Fig. 4) and the year 2009 value. We believe that this is the best to utilize the existing information. In addition, we determine year to year withdrawal aims to reach minimum difference between in situ and simulated groundwater time series and inflow into the lake. A comparison of the statistical value (Table 1) and Figure 7 indicate that this type of approach is reasonable. Finally, we aware of the definitions and possible inconsistencies.

Comment#5: Based on table 2, in the variant RS\_W\_GW the parameters  $\alpha$  and  $\beta$  vary between 0.45 to 0.47 and 0.47 to 0.52, respectively. That said, by introducing the NA data into the variant RS\_Q\_GW\_NA the parameter ranges expand significantly to 0.29 to 0.56 for  $\beta$  and 0.39 to 0.63 for  $\beta$ . This implies that the model is not benefiting from the additional information content. For instance, there is no significant improvement from RS\_W\_GW to RS\_W\_GW\_NA evidenced by Figure 8 and metric values on table 4; for years 2003, 2004, 2007, and 2009 the RS\_Q\_GW variant is even better than the RS\_Q\_GW\_NA in terms of estimating the annual inflow to the lake (Figure 8b).
That is, the model is insensitive to adding new data NA. Instead the new information is compensated by the model parameters. It is an indication of over-parameterisation in your model, which is expected for annual multipliers, that undermines the reliability of the results. While I understand the rationale behind a year-specific parameter value, it is a serious issue. Instead, you should do a more efficient and effective parameter search (instead of manual calibration) finding one or (more desirably) a number of acceptable parameter sets. Then to ensure their reliability you can do a year-by-year evaluation of the model performance, i.e. evaluating the model performance against a given metric for each year separately. Also, it would be helpful to include a schematic of the model structure demonstrating its fluxes, storages, and their interconnection. This can help the reader to better understand the mechanism and process-representation of the model.

Response: We have another viewpoint regarding this issue which a lesson we have learned. The models outputs can be changed by using parameters modification or input modification. As you mentioned, model variants RS\_W\_GW and RS\_W\_GW\_NA show similar results. It does not mean the model is insensitive to adding new data. It means that the model parameters can compensate errors in input data. Therefore, when the model is run under natural situation, the result is dubious. As a result, a holistic calibration needs a multi objective calibration. We do agree that having individual correction factors for each year can be considered as over-parameterization, and on might consider to have another model variant with temporally constant correction factor. However, this would add to the already high complexity of the manuscript, so we prefer to not show the results of such an experiment. Instead, in the revised version, we will have a new discussion section in which we will discuss the topic of over-parameterization.

WaterGAP schematic well documented in some papers. So to avoid repetition, please refer to Döll et al. (2014) who presented the WaterGAP structure in more details.

Comment#6: It is a well-established fact that CC is an inadequate measure for model

**HESSD**
evaluation [Willmott, 1981]. It is especially redundant to use CC together with NSE, as CC is already included in the NSE metric; see the NSE decomposition by Murphy [1988] and Gupta et al. [2009]. Further, NSE puts more emphasis on the larger values, e.g. as Pushpalatha et al. [2012] showed it focuses on the top 20% of discharge flows. Therefore, calibration by NSE introduces bias. So, it's more useful to combine NSE with bias rather than RMSE. Furthermore, other metrics such as Willmott's refined index of agreement [Willmott et al., 2012] and KGE [Gupta et al., 2009] shown to be better than NSE.

Response: We know that a high value of CC does not mean that the model is highly accurate. However, we disagree that "CC is an inadequate measure for model evaluation". CC is suitable for detecting synchronous behavior of two-time series, which is not done by NSE. As shown by e.g. Gupta et al. (2009), NSE reflects already the bias, i.e. if bias is large, the NSE will be low. So it seems redundant to show both the bias and the NSE. About Pushpalatha et al. (2012), please consider they have focused on low flows in daily time scale which time series has too much fluctuation. Hence, their results cannot be extended to the much less variable annual flows we analyzed (see Figure 8). Still, we could extend Table 4 by showing the three components of the KGE (bias, correlation coefficient, variability) in addition to NSE and RMSE.

Comment#7: Also, on P 5 L 25-29, you already explained that the standard WGHM is not calibrated for LU basin. Yet you reported the results of the standard model on Figures 7-8, and discussed it through the results and discussion. including the standard variant in your results and discussion does not serve the manuscript any benefit other than adding to its bulk. Further, it is also a well-known fact that (hydrologic) model cannot be validated [Konikow and Bredehoeft, 1992; Oreskes et al., 1994]. Therefore, using the term validation is both semantically and theoretically wrong. As a matter of good practice, it's been recommended to use the term evaluation instead of validation [Beven and Young, 2013]. Same comment applies to terms such as optimal values and optimal fit throughout the manuscript—there is no optimal set.
Response: One of the main goals of our manuscript, as also expressed in the title, is to determine "the value of different sets of spaceborne and in-situ data for calibrating a hydrological model". We need to investigate the likely improvement of our modelling performance after adding each new dataset. We agree "evaluation" is a better word instead of "validation". We will revise it. However, de Marsily et al. (1992) rejected the claims about the impossibility of model validation. Also about the optimal fit, we believe in reality there is no "global optimal" set for calibrating a hydrological model while we can determine "optimal" set. However, we will use "the most suitable set" phrase instead "optimal set" to prevent any misunderstanding.

Comment#8: In this study you have not investigated the model parameter space other than four calibration variants, while the calibration is done manually. So, first, there is no way to justify that the manually calibrated parameter values are the best-performing calibrations (despite what you said e.g. on P1 L16). Further, no sensitivity nor uncertainty analysis is performed which is nowadays a requirement for publishing a modelling analysis in the hydrologic community. Without any sensitivity/uncertainty analysis the reliability of the modelling results is questionable. Other than the model structure and parameters, the role of data uncertainty is important and should be discussed. For instance, the annual inflows are not the exact inflows to the lake. They are estimates of the last station, sometimes as far as 50 km from the lake.

Response: We agree with you regarding the manual calibration. But in case of multiobjective calibration, also non-manual methods of optimization may not lead to an overall optimum due to the complexity. A manual approach appeared to be more transparent to us. About the uncertainty, we agree. Uncertainty can be entered to modeling in different sections as you mentioned such as uncertainty in input data, model structure and so on. We have to accept uncertainty in in-situ input data such as your example about the inflow into the lake. These uncertainties are inseparable from nature and all the hydrologists know about it. We think our study contribute to elucidating uncertainties in understand freshwater systems by showing the different outcome/simulations
of reality when utilizing, for the four model variants, different observational data types. Ideally, one would be able to quantify uncertainty of observation and include this in the calibration process, but this appears beyond our scope. In the revised version, we will have a new discussion section in which we will discuss different type of uncertainties that affect the study outcomes.

Comment#9: There are fundamental issues in the discussion within the first paragraph of section 3.2. First, adding a new input data into model calibration/evaluation does not necessarily decrease the modelling uncertainty. In fact, by adding each new data you're introducing a new source of uncertainty as there is also uncertainty associated with data themselves; especially in a case like LU basin where the data uncertainty is very high both for ground and remote sensing data. The model parameter equifinality is not necessarily a problem [Savenije, 2001], in fact it can help us to improve our modelling in the face of uncertainty [Beven, 2009]. As all (hydrologic) models are wrong [Box, 1976], model ensembles-as multiple working hypotheses-are better suited than calibrated model with a single parameter set to describe/predict hydrologic systems given the uncertainties [Beven, 2012; Beven et al., 2012; Chamberlin, 1890; Clark et al., 2011; 2012]. Despite your statement, parameter equifinality does not ask for additional data! Adding data, in fact, may even exacerbate the model parameter equifinality, and one cannot make up for parameter equifinality by adjusting the parameter values. For instance, the model structure may not be able to benefit from the additional information content, and therefore the new data is redundant (which seems to be the case comparing the results of variants RS Q GW and RS Q GQ NA). What parameter equifinality implies is a more thorough search of the parameter and model space as well as more rigorous model evaluation schemes such as limits of acceptability approach [Beven and Binley, 2014]; even then the parameter equifinality will remain. In other words, each model variant is prone to model parameter equifinality. Further, parts of the discussion in this section are not new lessons (e.g. limitations of global hydrologic models in paragraph 2) and are well-established in the literature. Also, section 3 is very long, sometimes discussing too much details. I think it'd serve your discus-

**HESSD**
sion better to provide the top 3-5 main learned lessons as bullet points early on in the section. Then briefly explain each bullet point. The rest, especially modelling technicalities which may be valuable particularly for the reproducibility of the study, could be provided as supplement. In doing so, it's particularly helpful to restructure the result discussion by, first, explicitly discussing the limitations and uncertainties associated with the modelling design and consequent results. Second, discuss what results are therefore more/less reliable given the uncertainties. Third, what are the new findings of your studies and where do the results lie within the extant litertature. Finally, wat are the organic future research questions/directions based on what you learned and what lacking in the current literature.

Response: We disagree with your statement that adding observation data into models increase uncertainty because such data helps to understand more correctly the system behaviors. Unfortunately, we do not really understand your statements about "equifinality". While the approach for using parameter ensembles is attractive (and there is currently research done on ensemble-based calibration of WaterGA), Beven and Freer (2001) did state that equifinality leads to high uncertainties in the hydrological modeling. Further, Hamilton (2007) indicated that equifinality is only the result of parameter abuse which would prevent using a proper selection of parameters. In addition, Ebel and Loague (2006) showed that equifinality is not an unsolvable issue if we used wide range of observational data in modeling. Also Hamilton (2007) and Efstratiadis and Koutsoyiannis (2010) stated that by a multi-objective calibration we can deal with this issue. Thank you for suggesting an improved structure of section 3 that will help us for writing the revised version of the manuscript.

Comment#10: On P 2 L 12-16 you stated that the decreasing trend in precipitation (P) and increasing trend of temperature (T), and thus increased evaporation, has very likely to contributed to the decrease in the lake volume. This is also reported as one of the main reasons for lake degradation on P3 L12. This statement is debatable. First, in our recent analysis [Khazaei et al., in review], we showed that the decrease in P
and increase of T is not considerable in explaining the shrinkage of the lake; nor the decrease in T can be associated directly with an increase in lake evaporation. The major driver of the basin, as stated before, is the land use change and the substantial expansion of cropland areas. This has led to increase in the irrigation hence less available runoff as for the lake inflow, and also caused a major increase in evapotranspiration. The following sentence (last sentence of the paragraph) does not explicitly indicate the greater role of human activities in the lake desiccation compared to atmospheric climate change, which is the common finding of the most of the studies in this area [AghaKouchak et al., 2015; Aneseh et al., 2018; Stone, 2015; Torabi aghighi et al., 2018; Vaheddoost and Aksoy, 2018]. On P 24 L 34 you concluded that "climate change must be constrained to prevent strong decreases of precipitation and runoff". It is not clear to me what you mean by constraining climate change here. Also, as discuss previously the role of human activities are more substantial in the lake's fate than atmospheric changes. Further, on P 3 L 2-3, you have discussed the role of drought in the lake's water level decline. First, the term drought is ambiguous, and it should be further specified what type of drought is discussed; atmospheric, hydrologic, agricultural, ecologic, or anthropogenic. Second, the analysis by AghaKouchak et al. [2015] indicated no considerable trend in droughts, at 0.05 significance level, during the past three decades. They argued that the region has undergone even more severe multiyear droughts in the past that did not cause a major change in the lake's surface area. They, therefore, cautioned against overrating the role of drought on the drying of the lake and disruption of its water balance.

Response: We clearly stated in the first part of the introduction that both climate and human activities affected the LU basin and later review in sufficient depth the studies that, taken together, indicate just that, that it both drivers are important, and each study finds different weights for the two depending on the time frame and variables considered and the study approach. For example, as described in the manuscript, Shadkam et al. (2016) showed that the role of the climate on inflow into the lake is significant. Farokhnia (2015) who assessed the impact of land use change and cli-

**HESSD**
mate variability on Urmia Basin found that climate impact is the main reason of inflow reduction (This research is in Persian his results in English reported in Shadkam et al. (2016)). Some of the studies that found an insignificant impact of climate might have been too simplistic, such as AghaKouchak et al. (2015) who only assessed the precipitation amount while also a changes in precipitation frequency and pattern are important (which is included in our study). While the studies you mentioned considered only monthly or annual time series, the studies we mentioned above worked with daily time series. As a result, the former missed the effect of changes in pattern or precipitation frequency within a month. This changes were recently reported by Bavil et al. (2018) and Hosseini-Moghari et al. (2018) based on analysis of daily precipitation. Bavil et al. (2018) stated the frequency on precipitation event less than 5 mm/day increase significantly over basin during past decades. As a results, we can see two trends in the precipitation data. The first trend is a decreasing trend in annual precipitation and the second trend is related to ineffective precipitation events (less than 5mm/day) which have an increasing trend. Ineffective precipitation means that the precipitation value just increases surface soil moisture and after a while will evaporate. Thus, with increasing ineffective precipitation events we have less runoff with same amount of monthly precipitation. To sum up, we found that the drought studies (or precipitation trend studies in monthly or annual time scale) which assessed drought events over the basin work on monthly data. Hence, they cannot consider the change in daily precipitation pattern within a given month.

Independent of the submitted manuscript, we have in the meantime analyzed the inflow into the two main dams (Bokan and Mahabad) between 1992-2013 where human water use or cropland development is insignificant. We also analyzed the daily precipitation over the basin based on 63 rain gauge stations during 1992-2013. We calculated the anomaly of annual precipitation and depth of precipitation less than 5 mm as the ineffective precipitation (the sum of precipitation for all precipitation events less than 5 mm/day during a year) (Fig. 1a). This figure shows the ineffective precipitation increased significantly meanwhile annual precipitation decreased. This increase
in ineffective precipitation is not reflected in drought analysis at monthly or annual time scale. Fig. 1b shows inflow into the dams along with annual precipitation anomaly. As shown in the figure since 2005 onward the difference between precipitation and inflow to dams has increased. To sum up we believe that most studies which reported insignificant climate impact did not consider ineffective precipitation and change in precipitation patterns and as a result, they underestimated climate effects over the Lake Urmia basin. In addition, the impact of increased temperature was not taken into account, different from what happens if hydrological models are used for the analysis.

**[Fig. 1]**

We agree that the wording "constraining of climate change" is misleading. In the revised version, we will change it to "global-scale mitigation of climate change by reducing greenhouse gas emissions".

Comment#11: This is a technical note: you have used CC and NSE (P 9 L 11-12) to cross compare precipitation and temperature records of difference sources. First, I assume by CC you meant Pearson CC, which should be explicitly mentioned. Second, both CC and NSE are sensitive measures, i.e. a few number of large outliers can significantly change their values; especially for skewed distributions. It is better to use (more) resistant alternatives such as Spearman ranked correlation (instead of Pearson correlation) and Willmott's refined index of agreement [Willmott et al., 2012] (or ideally normalised the data using a transformation such as Box-Cox, first, and then compare the time series distance).

Response: Thank you. We will clarify in the text that Eq. 5 computes the Pearson CC. Further, Spearman ranked correlation is usually used to determine the correlation between ordinal variables. Spearman ranked correlation evaluated monotonic relationship between two variables. In monotonic relationship, the variables changes do not necessarily happen with a constant rate. Thus, for instance, if both variables are increasing, but change rates are not consistent, the Spearman coefficient equals +1
while Pearson correlation is less than +1. Therefore, when we compare the same variables such as precipitation the rate of change also is important. Hence, in our opinion, the Pearson CC is the most suitable indicator in our case as we do not correlate e.g. runoff and precipitation. In the revised version, we will therefore additionally provide the Spearmann ranked correlation coefficient and the Willmott index.

Comment#12: P 2 L 7: this is an unsubstantiated claim. As far as I know there is no (reliable) evidence on the degree of awareness regarding this issue. Please remove it, or provide the evidence.

Response: We will remove the sentence.

Comment#13: P 25 L 9: It is better to explicitly acknowledge the organisations that provided you with GRACE and climate data, and the URL links if available.

Response: We will do so.

Comment#14: P 28-29: The URLs in the ULRP references are not accurate. Please update them.

Response: We will update them in the revised version.

Thank you very much again for your time and hope our response have been resolved your ambiguities.

**References**

AghaKouchak, A., Norouzi, H., Madani, K., Mirchi, A., Azarderakhsh, M., Nazemi, A., Nasrollahi, N., Farahmand, A., Mehran, A., and Hasanzadeh, E.: Aral Sea syndrome desiccates Lake Urmia: Call for action, Journal of Great Lakes Research, 41, 307-311, doi:10.1016/j.jglr.2014.12.007, 2015.
Bavil, S. S., Zeinalzadeh, K., and Hessari, B.: The changes in the frequency of daily precipitation in Urmia Lake basin, Iran, Theoretical and Applied Climatology, 133(1-2), 205-214, doi:10.1007/s00704-017-2177-7, 2018.

Beven, K. and Freer, J.: Equifinality, data assimilation, and data uncertainty estimation in mechanistic modelling of complex environmental systems using the GLUE methodology, Journal of Hydrology, 249, 11–29, doi:10.1016/S0022-1694(01)00421-8, 2001.

Chaudhari, S., Felfelani, F., Shin, S., and Pokhrel, Y.: Climate and anthropogenic contributions to the desiccation of the second largest saline lake in the twentieth century, Journal of Hydrology, 560, 342-353, doi:10.1016/j.jhydrol.2018.03.034, 2018.

De Marsily, G., Combes, P., and Goblet, P.: Comment on 'Ground-water models cannot be validated' by LF Konikow & JD Bredehoeft, Advances in Water Resources, 15(6), 367-369, doi:10.1016/0309-1708(92)90003-K, 1992.

Döll, P., Müller Schmied, H., Schuh, C., Portmann, F. T., and Eicker, A.: Global-scale assessment of groundwater depletion and related groundwater abstractions: Combining hydrological modeling with information from well observations and GRACE satellites, Water Resources Research, 50, 5698-5720, doi:10.1002/2014WR015595, 2014.

Ebel, B. A., and Loague, K.: Physics-based hydrologic-response simulation: Seeing through the fog of equifinality. Hydrological Processes, 20(13), 2887-2900, doi:10.1002/hyp.6388, 2006.

Efstratiadis, A., and Koutsoyiannis, D.: One decade of multi-objective calibration approaches in hydrological modelling: a review, Hydrological Sciences Journal, 55(1), 58-78, doi:10.1080/02626660903526292, 2010.

Farokhnia, A.: Impact of Land Use Change and Climate Variability on Urmia Basin Hydrology (Dissertation) Tarbiat Modares University, Agricultral department (in Persian), 2015.
Gupta, H. V., Kling, H., Yilmaz, K. K., and Martinez, G. F.: Decomposition of the mean squared error and NSE performance criteria: Implications for improving hydrological modelling, Journal of Hydrology, 377(1-2), 80-91, doi: 10.1016/j.jhydrol.2009.08.003, 2009.

Hamilton, S.: Just say NO to equifinality, Hydrological Processes, 21, 1979–1980, doi:10.1002/hyp.6800, 2007.

Hosseini-Moghari, S.M., Araghinejad, S. and Ebrahimi, K.: Monthly Precipitation Assessment: a misleading tool for understanding the effects of climate change. 8th Global FRIEND-Water Conference, November 6-9, Beijing, China, 2018.

Kamali, M., and Youneszadeh Jalili, S.: Investigation of landuse changes in Lake Urmia Basin using remotely sensed images, Report of Urmia Lake Restoration Program (ULRP) (In Persian), 2015.

Pushpalatha, R., Perrin, C., Le Moine, N., and Andréassian, V.: A review of efficiency criteria suitable for evaluating low-flow simulations, Journal of Hydrology, 420, 171-182, doi: 10.1016/j.jhydrol.2011.11.055 ,2012.

Shadkam, S., Ludwig, F., van Oel, P., Kirmit, Ç., and Kabat, P.: Impacts of climate change and water resources development on the declining inflow into Iran's Urmia Lake, Journal of Great Lakes Research, 42, 942-952, doi:10.1016/j.jglr.2016.07.033, 2016.

HESSD
**HESSD**

---

## Author Response (AR1)

**Dear Editor and Reviewers,**

*We have revised the manuscript according to the insightful comments provided by the editor and reviewers as well as according to the short comments. All recommendations have been addressed in the revised manuscript. In addition to a point-by-point response which has been already uploaded in HESS website, all relevant changes made in the manuscript have been listed*

5 *below. We would like to thank you for the thorough consideration and critical comments that helped us improve our manuscript.*
* * *
Editor Comment
* * *
**Editor Comment:** You received high quality and detailed reviews to which you also provided detailed answers. Thanks to the referees and you for this discussion phase. To me the most important is the better highlight the added value of this work and

10 embed it in the vast amount of literature on Lake Urmia. Also some more details on the model and model performance is required.

**Response**: *We have highlighted the innovation of our study in comparison with the vast amount of the literature on Lake Urmia. Also, we have added some details on the model and calculated two new model performance indicators.*

- *With respect to "better highlight the added value of the work": We completely revised the abstract, reformulated the*

15 *objectives of the paper in the Introduction and included a section (section 4.2) where we discuss the shortcomings of past work on quantification of the relative impacts of humans and climate on the hydrology of Lake Urmia basin.*

- *With respect to "embed it in vast literature": We included and discussed (also in section 4.2) very recent publications.*

- *With respect to "more details on model and model performance": The WaterGAP description in section 2.1 was rewritten and extended, in particular regarding the simulation of lake dynamics and the performance of WaterGAP as*

20 *compared to other global hydrological models. In addition, following the suggestion of a reviewer, the WaterGAP model performance for Lake Urmia basin was quantified by two additional performance indicators (Table 4).*
* * *
Point-by-point response to the Referee #1
* * *
**Referee #1:** This manuscript uses WaterGAP Global Hydrology model to quantify the effects of human water use on inflow to Lake Urmia, lake water volume, and groundwater. The model was manually calibrated 4 for time using different observation

25 data sets (remote sensing of irrigated area, monthly total water storage anomaly, in-situ observations of stream flow, and groundwater levels from 284 wells). Strengths of the work include a focus on the pressing problem or Lake Urmia decline and identification of the effects on groundwater. With these strengths, there are also several issues that I feel need to be addressed to accept this manuscript for publication.

**Response:** *We would like to thank you for the thorough consideration and critical comments that helped us improving the*

30 *manuscript. We have tried to do our best to consider all your recommendations in the revised version. Below, we have provided a point-by-point response to your comments.*

**Referee #1:** Is the finding that humans affected lake decline new? There have been several recent studies that report this finding (Alborzi et al. 2018; Chaudhari et al. 2018; Shadkam et al. 2016) and some of these studied used the same model inputs as this work and also report groundwater changes. What is new in this work?

**Response**: *We agree that different studies have been conducted on the Lake Urmia for quantifying the anthropogenic and climatic impacts on shrinking of the lake. But most of these studies focused on drought events, inflow into the lake or lake levels. None of these studies considered the effect of human water use on TWSA and groundwater storage (quantitatively), even distinguishing surface water from groundwater use. Also none of previous studies used such a diverse observation data that allow understanding of the different storage compartments of the basin. Furthermore, it should be noted the study does not only aim at quantifying the different impacts of climate and human water use but equally at understanding, for the example of the investigated basin, the value of different observational data types for calibrating a hydrological model and thus understanding dynamics and flows in a basin (see page 5, lines 30ff). Our study shows that for a comprehensive hydrological modeling that captures correctly the main dynamics in a basin, a multi-objective calibration is needed, and then previous studies missed some part of a comprehensive hydrological process modeling (e.g. the specific impact of groundwater pumping as compared to surface water use). Below please find the shortcomings of the studies you mentioned.*

*Alborzi et al. (2018):*

*1) Did not use a hydrological model.*

*2) Did not have any discussion on groundwater, lake storage, total water storage under natural condition.*

*Chaudhari et al. (2018):*

*1) Used global hydrological model (HiGW-MAT) without calibration for Lake Urmia basin.*

*2) In hydrological modeling the focus is only on streamflow and there is no discussion on groundwater, lake storage, total water storage under natural condition.*

*3) Did not take into account impacts of domestic and industrial water use.*

*4) Did not use in-situ data for irrigation water requirement and used FAO Penman Monteith method for estimating irrigation water requirement.*

*5) Did not use in-situ climatic data for estimating irrigation water requirement (used 6 h atmospheric reanalysis data provided by Japanese Meteorological Agency (JMA) Climate Data Assimilation System (JCDAS) as meteorological data)*

*Shadkam et al. (2016):*

*1) Only considered a single objective calibration of VIC model based on the inflow into the lake.*

*2) In hydrological modeling the focus is only on streamflow and there is no discussion on groundwater, lake storage, total water storage under natural condition.*

*3) All irrigation requirements need to be fulfilled by available surface water.*

*4) Did not take into account the impact of domestic and industrial water use.*

*As mentioned above, none of previous study provide a comprehensive modeling such as our manuscript. We have described all modeling studies over Lake Urmia basin along with their limitations in Introduction of the revised version.*

**Referee #1:** Is a global hydrologic model appropriate for a basin level analysis?

**Response**: *Even though global hydrological models have a coarse spatial resolution, they contain a lot of different data (climate, physiographic, water use etc.) for which local information may not be available or of better quality. The study has shown that for the relatively large Lake Urmia basin, the global WGHM model can be informative after multi-observation calibration as after calibration the fit to those different types of observations is rather good.*

**Referee #1:** The description of how the model simulates relevant processes is scant.

**Response:** *Due to the fact that the length of manuscript already is long and also there are some other literatures that have described the model simulates relevant processes, we explained the model in a summarized form. In the revised version we have added the following sentence to section 2.1:*

*"WGHM simulates daily water storage as well as flows like evapotranspiration, groundwater recharge (Döll and Fiedler, 2008), runoff, and river discharge for all continents except Antarctica. Water is transported between grid cells according to the DDM30 drainage direction map (Döll et al., 2003). Water storage compartments encompass snow, canopy, soil, groundwater, rivers, lakes, wetlands, and man-made reservoirs (Eicker et al., 2014). Lake water storage is simulated as the difference of precipitation on the lake, evapotranspiration, inflows, and outflows. Outflow is zero for end lakes like Lake Urmia. The temporal variation of lake area, affecting precipitation on and evapotranspiration from the lake, is simulated as a non-linear function of lake water storage. WGHM contains more than 20 parameters that can be potentially be adjusted by calibration (Werth and Güntner, 2010)."*

**Referee #1:** Given resonance times, how appropriate is the temporal spacing (daily) relevant to the spatial grid size?

**Response**: *In our opinion, a daily time step fits well to simulation of water flows and storage at the 0.5° grid scale and is the usual time step in global hydrological modeling. Land surface models that also simulate energy flow require a smaller time step.*

**Referee #1:** Is it computationally efficient to run a global model for 15 or 20 grid cells of interest? There needs to be a much stronger justification for why the modeling and calibration methods are the correct approaches to use to answer the motivating questions.

**Response**: *It should be noted there is no need to run the model for whole globe. The model can be run for a specific basin, in the case of Lake Urmia for just 22 grid cells. Therefore, simulations are highly efficient. We have added this explanation to section 2.1. Using WaterGAP to answer the motivating questions is also efficient in that the model for the Lake Urmia basin was already set up at the beginning of the study as WaterGAP as a global model is ready for simulating any (large enough) basin around the world. We do not think that it is necessary to further explain in the text why the modeling and calibration approach is suitable, as e.g. section 2.1 describes that WaterGAP allows to consider a complex hydrological system (including surface water, groundwater, lake, human water use, etc).*

**Referee #1:** There is a lot of focus in the text on the multiple calibration variants run with different input data sets. What was learned from this activity? How do those result effect Lake Urmia management?

**Response**: *We described the lesson we learned from this activity in details in section "3.2 What we learn from the calibration?". We learned that there is no guarantee that a single objective calibration improves the model performance with respect to the simulation of other components of hydrological system. As a result, for defining the Lake restoration plans a comprehensive modeling framework like our study is needed. Specifically, quantification of return flow from irrigation is*

5 *paramount for managing irrigation in the basin. We have added the following paragraph to the conclusion, as the last one:*

*"Our study has shown that management of the Lake Urmia basin should be based on a comprehensive assessment of all water storages and flows in the basin, including human water uses of groundwater and surface water. We recommend refining the estimated net abstractions from surface water and groundwater by a basin-wide spatially explicit quantification not only of water abstractions but also return flows to groundwater and surface water."*

10 *Also, to clarify of research objectives we have revised the objectives of the paper in the Introduction section as follows:*

*"The aim of our study was twofold. On the one hand, we wanted to quantify, by a holistic and reliable modelling approach, the contributions of climate variations and human activities to the decrease of Lake Urmia water volume and river inflows as well as, different from previous studies, to groundwater storage and total water storage in the whole Lake Urmia basin. Such a modelling approach requires the set-up of a model that is able to simulate the impact of surface and groundwater use as*

15 *well as of climate variations on these water storages and flows. The hypothesis is that if model output for all these variables fit well to observations, then the model can be used to assess the contribution of human water use by comparing the outputs of two model variants, one with human water use and one where human water use is assumed to be zero. To achieve a good fit to observations, hydrological models need to be calibrated by comparison of observations with model output variables. While hydrological models are usually calibrated only against observations of river discharge, it is well known that a good fit of*

20 *simulated and observed river discharge does not lead necessarily lead to an appropriate simulation of other flows and storages (Beven and Freer, 2001). However, in previous hydrological modeling studies of Lake Urmia basin, model calibration was either not done at all or only using a single observation type. On the other hand, using Lake Urmia basin as a test case, we wanted to explore the value of different types of observation data for adjusting a global hydrological model by multi-observation calibration. Currently, global hydrological models are mostly uncalibrated but globally available space-born*

25 *observations have increased the opportunity for model calibration at the global scale (Döll et al., 2016)."*

**Referee #1:** Also, what could one potentially learn from 4 model calibrations that use different calibration data and yield four different models?

**Response**: *It is very common to use a single or two objective calibrations for calibrating a hydrological model. In this study, we have tried to understand which level of data can reveal that our modelling is holistic. Based on the results, for a holistic*

30 *modeling, at least remote sensing, discharge and groundwater levels data are required. In addition, we have investigated with adding each in-situ data in calibration process how the model performance improved in simulating different water resources components.*

**Referee #1:** What are the limitations of this study? The discussion of uncertainty in the results needs to go much deeper and be more specific. This uncertainty is real and likely plays a large role in the interpretation of the results.

**Response**: *In the revised manuscript, we have added, as section 4.3, a short discussion.*

*"4.3 Limitations*

*Even after multi-objective calibration of a state-of-the-art comprehensive hydrological model, there remain many uncertainties that affect the accuracy of the model results. Like the results of all hydrological models, our results are affected by uncertainties in model input, model parameters, and model structure. Model parameter uncertainty was reduced by the comprehensive multi-observation calibration, albeit conditioned on just one climate input data set and using just one model (instead of the state-of-the-art multi-model ensemble approach, compare www.isimip.org, last accessed: 14 Dec. 2018). Given the low spatial model resolution ($0.5°×0.5°$), the model results are only valid for the basin as a whole and results for individual grid cells are very uncertain. Also due to a lack of data at the basin scale, the hydrogeology of the basin was not taken into account in the model. Information on irrigated area in each grid cell was taken from a global data set of areas equipped for irrigation from groundwater and surface water (Siebert et al., 2010), which was adapted in this study by scaling it by basin-wide correction factors to better capture the temporal development of irrigation. Calibrated modeling results are also affected by uncertainties of the observation data. GRACE TWSA data are more reliable for larger (100,000 km2 (Landerer and Swenson, 2012)) areas than the basin area. Estimation of groundwater storage changes based on water level data for unevenly distributed wells is rather uncertain due to the unknown heterogeneities in the subsurface. Evaluation results, here the good fit of simulated to "observed" lake water volume decline, are be affected by a likely underestimation of the actual decline by the "observed" value derived from remote sensing of lake water level elevation and lake water area by Tourian et al. (2015) assuming a constant bathymetry. However, there was an increase in the elevation of the lake bottom due to sedimentation and salt precipitation (Shadkam et al., 2016) so that the "observed" water volume decline was likely lower than the actual one, and our model would underestimate the lake storage decline, too."*

**Referee #1:** I found the writing difficult to follow in numerous places, particularly the results section. There are lots of acronyms, run-on sentences, and text that digresses from the section headers or topic sentences of paragraphs. The writing here made it difficult for me to see the main results and findings of the work.

**Response**: *We have revised whole of manuscript and reformulate the results section for the revised version.*

**Referee #1:** pp. 2-4. The first three figures recount results from prior work. I would much prefer to see figures and tables focus on new insights gained from the work. For example, new figures that show uncertainties.

**Response**: *In our opinion these figures are needed to inform the reader about the story that happened at Lake Urmia basin. Also, only Figure 2 is taken from previous studies. Due to the length of manuscript, we prefer not to add new figures.*

**Referee #1:** p. 3, line 5. I think "somewhat recovered" is overstated. Hard to tell from Figure 3. Maybe stabilized.

**Response**: *We agree with you. So, we have revised the sentence as follow:*

*"After 2015, lake extent and storage have stabilized".*

**Referee #1:** p. 3, line 18. Is the value -11.2 mm/yr correct? It seems incredibly small. In The Hashemite Kingdom of Jordan, drawdowns are 1+ m/yr, in numerous wells. In the U.S., we talk about drawdowns of ft/year.

**Response**: *The value taken from the study of Forootan et al. (2014) refers to groundwater storage, not to a drawdown of the groundwater table. And it is the average loss over the whole basin and not a drawdown in an extraction well which of course can be much higher than an average drawdown over the whole basin. Change in groundwater storage can be calculated by multiplying change in groundwater level with the specific yield.*

5 **Referee #1:** p. 6, line 5. Only the anomalies? Or at all time periods? If the former, please explain what is meant by "anomaly", how determined, and why anomaly is the appropriate frame to discuss. I would want to calibrate a model across a range of conditions some of which might include anomalies.

**Response**: *The model has no information about the real bathymetry and initial value of lake storage; it can only simulate changes as compared to some initial condition. As a result, we can compare only lake storage change or lake storage* 10 *anomalies. Lake storage anomaly at a given time is equal to the lake storage at that time minus the long-term average of lake storage. In this study, as mentioned in p. 20 line 24, anomalies were calculated with respect to the mean lake storage during 2004-2009 (baseline period used for the provided GRACE data).*

**Referee #1:** p. 6, lines 18-28. I'm not familiar with WaterGAP. How does this model actually work? Explain.

**Response**: *While the WaterGAP description in section 2.1 is brief, the reader is referred to publications that provide more* 15 *information on the model. However, as written above, we plan to extend the WaterGAP description slightly, in particular with respect of lake modeling.*

**Referee #1:** p. 7. What is total water storage anomaly (TWSA)? This term seems rather central to the paper. Please explain.

**Response**: *Total water storage (TWS) is amount of water which is stored in different components of the continents, e.g. as follows (Scanlon et al., 2018):*

$$TWS = SnWS + CWS + SWS + SMS + GWS \qquad (1)$$

20 *where SnWS is snow water storage, CWS is canopy water storage, SWS is surface water storage, SMS is soil moisture storage, and GWS is groundwater storage. Neither hydrological models nor GRACE can compute the total amount of stored water. They can only compute varisations according to a temporal average. Therefore, TWS anomalies (TWSA) are evaluated, defined as TWS(t) – mean (TWS).*

*We have inserted the following sentence as the second sentence of the GRACE section (p. 7, line 9):*

25 *"TWSA describes the total amount of water stored on the continents, including water storage in surface water bodies, groundwater and soil, as compared to the mean value of total water storage over a reference period."*

**Referee #1:** p. 8, lines 13-18. This method of applying (1- return flow multipliers) to the abstractions to estimate consumptive use assumes that water is used by only one water user. Is this a realistic assumption? If the return flow is used by another agricultural user and then again by a 3rd or 4th user, the basin-wide consumptive use fraction will be much different than the 30 values reported. The large grid size magnifies this error. Table 1. How sensitive are study results to the values in this table?

**Response**: *In our opinion due to the fact that most return flow returns to groundwater, there is no concern in this regard. On the other hand, in arid area e.g. Urmia basin the return flow to surface water in each irrigation is not too much that can be used by another user. Anyway, we do not know exactly how the authors of the values determined them. However, the study results are not sensitive to the independent return flow estimates that were used only in one variant (RS_Q_GW_NA). We have written in section 4.1:*

*"Consideration of regional estimates of human water withdrawals in a specific year as well as regional estimates of return flow fractions in variant RS_Q_GW_NA does not improve the fit to observations significantly and only leads to slight parameter adjustments. This indicates a reasonable simulation of per hectare water consumption for irrigation by the WaterGAP model."*

**Referee #1:** p. 10. Lines 5-15. So the correction factors are needed because WaterGAP does not get the underlying physical hydrology correct? The correction is linear? Is the process causing the error also linear?

**Response**: *Irrigated area in the standard version of WaterGAP is constant during the period of investigation. The correction factor based on MODIS remote sensing time series adjusts the WaterGAP value in each year homogeneously across the basin. Standard WaterGAP irrigation area is multiplied by the correction factor.*

**Referee #1:** p. 11, line 1. Which parameters were varied to calibrate this model?

**Response:** *These parameters are presented in Table 3.*

**Referee #1:** p. 11, line 4. What is meant by optimal fit?

**Response**: *Optimal fit means the best possible match between observed and simulated time series. Performance of the fitted model is quantified by three different performance indicators in Table 4.*

**Referee #1:** p. 15, line 2 and Table 3. Shouldn't these parameter values be the same across all the model variants? What is physically changing in the system that these parameter values would change across the model variants?

**Response**: *These parameters should not be the same because model calibration means change in model parameters to achieve the best fit to observations. As different observations are used in each variant, the values of calibration parameters are different, too. There is no physical change in the basin but different parameter values indicate that model parameterization cannot be uniquely determined. Calibration just to streamflow observations, as is usually done in hydrological modeling, does not assure a correct simulation of water storage changes, for example. In our study, the parameters optimized by just using remote sensing information for model calibration (variant RS) lead to an unsatisfactory simulation of inflow into the lake (see Table 4).*

**Referee #1:** p. 16, lines 2-3. There could be a net groundwater abstraction but still areas where there is recharge. Is this an issue of coarse spatial resolution?

**Response**: *Yes, there could be areas of recharge due to irrigation with surface water. However, the study just analyzes the mean behavior over the whole basin, based on the results in 22 grid cells.*

**Referee #1:** p. 18, lines 1-10. The discussion of uncertainty here is missing a fundamental point. Calibration cannot help if the model structure is uncertain (or in error) or the temporal or spatial scaling of the model is mismatched to the modeled

parameters of interest. This discussion also heads in a different direction than "what do we learn from the calibrations?" The text never explains what was learned. What was learned? Please discuss.

**Response**: *Regarding uncertainty of model structure, we plan to refer to multi-model ensembles in the new discussion section (see above). In this section we do explain what we learned from our calibration exercise or rather from adding more types of observations. It is outside the scope of the paper to discuss the impact of temporal or spatial scales on model results as we did not investigate this (or e.g. uncertainty due to the applied climate forcing). Thus, when we discuss different calibration variants we discuss how uncertainty can be reduced by additional observational data types.*

*We believe that in the "What we learn from the calibration?" section, the key findings were reported. The most important of them reveals that a single objective/observation calibration cannot capture hydrological dynamics and there is no guarantee a well-simulated model based on a tuned variable can properly simulate other components of the model. As a result, for a general statement about water resources in a given region a multi-objective calibration is required.*

**Referee #1:** p. 18, lines 11-20. These statements are better placed in the introduction to justify the use of global hydrologic models. Still, why is a global model the appropriate choice when the domain of study is limited to one hydrologic basin (Urmia)?

**Response**: *We prefer to leave this paragraph in the results section to clarify the context of the calibration exercise that is on the one hand done to efficiently analyze the specific situation in the Lake Urmia basin but also to evaluate the value of calibrating global hydrological models against multiple observation types. The reasons for using a global model have been given above.*

**Referee #1:** p. 19, line 22. What beta?

**Response:** *At the beginning of the sentence, there is the reference to Eq. 2. Beta adjusts the net abstraction from groundwater.*

**Referee #1:** p. 19, line 30. "much less overestimated" means what?

**Response:** *Its means that although the model variant still overestimates inflow to the lake, the degree of overestimation has decreased strongly.*

**Referee #1:** p. 19, line 31-32. This doesn't make sense to me. How come it is ok to change the parameter in one model variant but not others?

**Response**: *It is another thing we can learned from the calibration. As shown in Figure 6, in each variant the model is calibrated against different observations. For example, in the first variant the WGHM evaluated based on TWSA and in the second one WGHM evaluated based on TWSA and inflow into the lake. In the first variant we did not need to change the parameters which have no effect on simulated TWSA. But in the second variant we have to calibrate model against both TWSA and inflow into the lake. So, in this variant the parameters which have an effect on inflow into the lake also should be adjusted. It means that when we have a single objective calibration it is possible we reach to appropriate results via different combination of parameters values. When we add more and more objectives or observational data types in the calibration process, the number of parameters which should be changed.*

**Referee #1:** p. 20, lines 1-5. I would expect to see better calibration with more observational data (i.e., stream flows and lake levels).

**Response**: *We agree with you. Our results agree with your expectations, and this is what we express in the sentence starting in line 4: "Still, calibration to both observational data types leads to the best simulation of both annual lake inflow and lake volume anomalies."*

**Referee #1:** p. 20, line 22. I'm confused. The scenario "with reservoirs but without human water use" does not fit either of the two scenarios described in the prior sentence.

**Response**: *WHGM has the capability to assess the effect of dam building on water resources without considering human water use from reservoirs. Our results showed that the results with and without reservoirs has only 2% difference which indicated insignificant effect of reservoirs on water resources over the basin. So the run with or without reservoirs effect can be considered the same. As a result, we can consider the run without human water use as WGHM run under natural condition.*

**Referee #1:** p. 20, line 24. What is meant by anomalies? This term has still not been defined.

**Response**: *Anomalies is the difference from an average, or baseline for example in this study GRACE data are total water storage anomalies based average of its observation between 2004-2009. Added to the text when introducing GRACE (see response above).*

**Referee #1:** p. 21, line 17, "The lower lake water loss…" What are the loss terms besides evaporation? How are these other loss terms smaller when inflow is larger? Explain.

**Response**: *Evaporation is the only loss term of the lake. There was less decrease in lake storage due to more inflow into the lake. We have revised the sentence as:*

*"The smaller decreasing trend for lake water volume under naturalized conditions is clearly caused by more inflow into the lake, even though lake evaporation is somewhat higher under naturalized inflow conditions due to the larger lake extent."*

**Referee #1:** p. 21, lines 20-23. I don't follow this explanation. There are too many NAs in this sentence. What causes the difference between the naturalized and anthropogenic scenarios?

**Response**: *In naturalized runs, abstraction from the surface water or groundwater are assumed to be zero, while and water flows and storages vary only due to climate variations while in the anthropogenic scenario we simulate both the effect of water abstraction by human and of climate variations. We have deleted the sentence in lines 20-22.*

**Referee #1:** p. 21, lines 25-28. I don't follow. What is the connection between the first part of the sentence and the second part?

**Response**: *We wanted to state that inflow was less than the minimum environmental water requirements ($3,085 \cdot 10^6$ m³) and that therefore a loss of lake water volume is expected. So, the first part of the sentence indicated under anthropogenic situation the inflow into the lake since 2008 never has reached to minimum environmental water requirements. The second part is related to naturalized condition which showed that under naturalized condition only in 2008 and 2009 the inflow into the lake were less than minimum environmental water requirements. We intend to reformulate the first sentence as follows:*

*"Since 2008 the inflow into the lake has never reached 3,085·10⁶ m³/yr. This is the value estimated to be the minimum environmental water requirements that compensates the amount of annual evaporation from of the lake surface (Abbaspour and Nazaridoust, 2007). Therefore, a decrease of lake water storage can be expected for the best estimate of WaterGAP of 2,639·10⁶ m³/yr."*

5   **Referee #1:** p. 21, lines 28-32. Is a run-on sentence.

**Response**: *We agree with you. We have revised it.*

**Referee #1:** p. 21, lines 32 – 21. Put these ratios in context. What is desirable? Undesirable? What has implications for lake health? What values are acceptable?

**Response**: *We think that it does not make sense to talk about acceptable values. From the perspective of lake health, a higher*

10   *value may be more desirable, or rather that the values under anthropogenic conditions become closer to the values under naturalized conditions. However, we do not think it is necessary to state this explicitly in this scientific publication, it will be clear to the reader even without stating this explicitly.*

**Referee #1:** p. 23, line 15. How can water storage be negative?

**Response**: *Total groundwater storage is not computed in WaterGAP, only storage relative to a storage that occurs when the*

15   *heads in surface water and groundwater are the same. Negative values of groundwater storage computed by WaterGAP indicate that net abstractions from groundwater are larger than natural groundwater recharge, while baseflow is zero. Groundwater levels can be assumed to have dropped below the surface water heads. In WaterGAP, groundwater recharge from rivers is not taken into account.*

**Referee #1:** p. 23, line 18. This is an interesting result. It needs to be much more strongly emphasized. These cells are the

20   locations where groundwater declines and there could be problems.

**Response**: *As WaterGAP does not include reliable high-resolution information on irrigated areas and groundwater use infrastructure, the computed cell-specific net abstractions from groundwater are highly uncertain. This is why our study focuses on basin averages, and cell-specific results are less prominently shown.*

**Referee #1:** p. 23, lines 21-23. This qualification and limitation seems rather important. Why should the model results be

25   trusted or used if the model does not get groundwater storage correct?

**Response**: *No model is perfect, and calibration as done here is a way to compensate for a lack of process accuracy. Due to calibration, we do get groundwater storage (more or less) right.*

**Referee #1:** p. 23, lines 23-25. Run-on sentence. What is meant by the clause with maximum?

**Response**: *As mentioned in p. 23 line 22, WaterGAP cannot simulate a possible drop of the groundwater table below the*

30   *surface water level in the absence of groundwater abstractions, and groundwater storage might in reality have been lower before the start of groundwater abstraction than simulated in the naturalized run. Thus, contribution of human water use to groundwater storage decline might therefore be overestimated.*

**Referee #1:** p. 24, lines 18-19. Is this result surprising? More calibration data means a better fit model. How does this result improve understanding of the Lake Urmia system?

**Response**: *There are a few hydrological modelling studies on Lake Urmia basin but all of them (except Chaudhari et al. (2018) who has not implemented calibration at all) have only a singly observational data type/objective for calibration. So as first multi objective calibration study of Lake Urmia basin we show how a single objective calibration does not have the proper capability for a comprehensive modeling in the basin. As a result, this study provides a unique information for understanding*

5  *the Lake Urmia basin system not only the Lake Urmia which does not reported in any previous studies. In addition, if the model structure or the input data are wrong, it may not be possible to improve the simulation of an increasing number of observational data type that are used for calibration. Some trade-off may occur; for example, total water storage anomaly simulation may decrease but streamflow performance may increase if streamflow is added as a second calibration data type, in addition to total water storage anomaly.*

10 **Referee #1:** p. 24, lines 25-28. Is this finding new? If so how? I feel the Urmia Lake Recovery Program has been working under the assumption that agricultural water use was a large contributor to lake decline and that they have been taking steps in recent years to address.

**Response**: *Regarding lake inflow, Shadkam et al. (2016) reported similar results as our study. However, results regarding TWSA, lake storage and groundwater storage are the new findings.*

15 *Yes, Urmia Lake Restoration Program (ULPR) is working under the assumption that agricultural water use was a large contributor to lake decline. ULPR cannot manage climate variations or changes, so they certainly need to focus on management water use over the basin, if according to our study, human impact is significant.*

**Referee #1:** p. 24, line 29. 90% of what?

**Response**: *90% of groundwater storage losses. We have reformulated the sentence as follows:*

20 *"90% of groundwater storage loss is estimated to be caused by human water use but this value may be somewhat overestimated by WGHM because climate-driven loss under naturalized conditions may be underestimated due to the simplified representation of groundwater-surface water exchanges in the model."*

**Referee #1:** p. 24, lines 19-24. I disagree. There are lots of other similar systems in the world – Great Salt Lake, Owens Lake, Dead Sea, Ural Sea, etc. each satisfy the first two criteria listed. What of these results is generalizable?

25 **Response**: *Thank you for pointing this out. We do not know what is generalizable as we have not done this calibration exercise in other basins. We intend to add one sentence at the end of the paragraph that provides a recommendation based on our study.*

*"In basins with large lakes, and in particular with end lakes, remotely sensed time series on lake area and the elevation of the lake water table should be used to estimate time series of lake water storage as these observational data can be expected to*

30 *be of high value for understanding the freshwater system by hydrological model calibration. Groundwater storage cannot be observed from space but relies on in-situ observations on groundwater heads in wells but, as in the case of Lake Urmia basin, but such data may be crucial for a correct understanding of the freshwater system."*

**Referee #1:** p. 24, lines 31-34. How do the model results inform the 2014-2017 trends? Also, how can climate change be constrained in this basin? Explain.

**Response**: *Unfortunately, almost all the input datasets are not available from 2014 onwards. Thus we did not have the model results for recent years. With "constraining climate change" we meant the global reduction of greenhouse gas emissions, nothing at the basin scale. We have revised this part as follows:*

*"Further strengthening of efforts for decreasing human water use in the basin should be undertaken, while at the same time, global-scale mitigation of climate change by reducing greenhouse gas emissions to prevent strong decreases of precipitation and runoff."*

**Referee #1:** Figure 9. What is being shown in panels A, B, and D? The y-axis labels were mentioned in the text but never explained.

**Response**: *All panels have been explained in following pages and lines:*

*Figure 9a on p. 21 lines 1-5.*

*Figure 9b on p. 21 lines 15-16.*

*Figure 9d on p. 22 lines 1-14.*

**Referee #1:** Figure A1a. The color scheme makes it difficult to differentiate grid cells. Use only three colors to differentiate the 3 types of storage. How can storage volume be negative?

**Response:** *We have changed the colors according to your suggestion. Total groundwater storage is not computed in WaterGAP, only storage relative to a storage that occurs when the heads in surface water and groundwater are the same. Without NAg, storage can therefore never become zero or less than zero. If NAg becomes larger than groundwater recharge, storage can obtain negative values.*

**Referee #1:** Data availability. I don't follow. If the authors do not have permission to share the data, then how can they share by author request? The HydroSat site underwritten by the University of Stuttgart is neat. What is the original source data for Urmia? Also, there is no water storage anomaly data for Urmia.

**Response**: *We have no permission to share data except by personal request for research purposes. We have indicated the source of all data in the manuscript. The data for Lake Urmia was obtained from various sources. Regarding HydroSat site, website you can download original data for lake level and extend with related reference which is Tourian et al. (2015) after registration. Also about water storage anomaly data for Urmia, for the lake water storage anomaly, it should be noted that anomalies can be calculated easily by subtracting the mean lake water storage in baseline (2004-2009) from the lake water storage time series.*

*The section on data availability have been reformulated as follows:*

*"In-situ data from "Iran Water Resources Management Company" including groundwater levels, precipitation and temperature publicly are available upon request from the corresponding author. GRACE data is available through http://www2.csr.utexas.edu/grace/RL05_mascons.html (last accessed: 17 Jul. 2018). Lake water surface extents and water levels are available at http://hydrosat.gis.uni-stuttgart.de/php/index.php (last accessed: 17 Jul. 2018). All simulation results are available in the supplement."*

*We have done our best to consider all your comments and suggestions in the revised version. Below you can see the changes have applied based on your comments.*

**Referee #1:** The description of how the model simulates relevant processes is scant.

5 **Response:** *In the revised version we have described the model in more details as follows (section 2.1):*

*"WGHM simulates daily water storage as well as flows like evapotranspiration, groundwater recharge (Döll and Fiedler, 2008), runoff, and river discharge for all continents except Antarctica. Water is transported between grid cells according to the DDM30 drainage direction map (Döll et al., 2003). Water storage compartments encompass snow, canopy, soil, groundwater, rivers, lakes, wetlands, and man-made reservoirs (Eicker et al., 2014). Lake water storage is simulated as the*

10 *difference of precipitation on the lake, evapotranspiration, inflows, and outflows. Outflow is zero for end lakes like Lake Urmia. The temporal variation of lake area, affecting precipitation on and evapotranspiration from the lake, is simulated as a non-linear function of lake water storage. WGHM contains more than 20 parameters that can be potentially be adjusted by calibration (Werth and Güntner, 2010)."*

**Referee #1:** There is a lot of focus in the text on the multiple calibration variants run with different input data sets. What was

15 learned from this activity? How do those result effect Lake Urmia management?

**Response**: *We described the lesson we learned from this activity in details in section "3.2 What we learn from the calibration?". We learned that there is no guarantee that a single objective calibration improves the model performance with respect to the simulation of other components of hydrological system. As a result, for defining the Lake restoration plans a comprehensive modeling framework like our study is needed. Specifically, quantification of return flow from irrigation is*

20 *paramount for managing irrigation in the basin. We have added the following paragraph to the conclusion, as the last one:*

*"Our study has shown that management of the Lake Urmia basin should be based on a comprehensive assessment of all water storages and flows in the basin, including human water uses of groundwater and surface water. We recommend refining the estimated net abstractions from surface water and groundwater by a basin-wide spatially explicit quantification not only of water abstractions but also return flows to groundwater and surface water."*

25 *Also, to clarify of research objectives we have revised the objectives of the paper in the Introduction section as follows:*

*"The aim of our study was twofold. On the one hand, we wanted to quantify, by a holistic and reliable modelling approach, the contributions of climate variations and human activities to the decrease of Lake Urmia water volume and river inflows as well as, different from previous studies, to groundwater storage and total water storage in the whole Lake Urmia basin. Such a modelling approach requires the set-up of a model that is able to simulate the impact of surface and groundwater use as*

30 *well as of climate variations on these water storages and flows. The hypothesis is that if model output for all these variables fit well to observations, then the model can be used to assess the contribution of human water use by comparing the outputs of two model variants, one with human water use and one where human water use is assumed to be zero. To achieve a good fit to observations, hydrological models need to be calibrated by comparison of observations with model output variables. While*

*hydrological models are usually calibrated only against observations of river discharge, it is well known that a good fit of simulated and observed river discharge does not lead necessarily lead to an appropriate simulation of other flows and storages (Beven and Freer, 2001). However, in previous hydrological modeling studies of Lake Urmia basin, model calibration was either not done at all or only using a single observation type. On the other hand, using Lake Urmia basin as a test case, we wanted to explore the value of different types of observation data for adjusting a global hydrological model by multi-observation calibration. Currently, global hydrological models are mostly uncalibrated but globally available space-born observations have increased the opportunity for model calibration at the global scale (Döll et al., 2016)."*

**Referee #1:** What are the limitations of this study? The discussion of uncertainty in the results needs to go much deeper and be more specific. This uncertainty is real and likely plays a large role in the interpretation of the results.

**Response***: In the revised manuscript, we have added, as section 4.3, a short discussion.*

*"4.3 Limitations*

*Even after multi-objective calibration of a state-of-the-art comprehensive hydrological model, there remain many uncertainties that affect the accuracy of the model results. Like the results of all hydrological models, our results are affected by uncertainties in model input, model parameters, and model structure. Model parameter uncertainty was reduced by the comprehensive multi-observation calibration, albeit conditioned on just one climate input data set and using just one model (instead of the state-of-the-art multi-model ensemble approach, compare www.isimip.org, last accessed: 14 Dec. 2018). Given the low spatial model resolution (0.5°×0.5°), the model results are only valid for the basin as a whole and results for individual grid cells are very uncertain. Also due to a lack of data at the basin scale, the hydrogeology of the basin was not taken into account in the model. Information on irrigated area in each grid cell was taken from a global data set of areas equipped for irrigation from groundwater and surface water (Siebert et al., 2010), which was adapted in this study by scaling it by basin-wide correction factors to better capture the temporal development of irrigation. Calibrated modeling results are also affected by uncertainties of the observation data. GRACE TWSA data are more reliable for larger (100,000 km2 (Landerer and Swenson, 2012)) areas than the basin area. Estimation of groundwater storage changes based on water level data for unevenly distributed wells is rather uncertain due to the unknown heterogeneities in the subsurface. Evaluation results, here the good fit of simulated to "observed" lake water volume decline, are be affected by a likely underestimation of the actual decline by the "observed" value derived from remote sensing of lake water level elevation and lake water area by Tourian et al. (2015) assuming a constant bathymetry. However, there was an increase in the elevation of the lake bottom due to sedimentation and salt precipitation (Shadkam et al., 2016) so that the "observed" water volume decline was likely lower than the actual one, and our model would underestimate the lake storage decline, too."*

**Referee #1:** I found the writing difficult to follow in numerous places, particularly the results section. There are lots of acronyms, run-on sentences, and text that digresses from the section headers or topic sentences of paragraphs. The writing here made it difficult for me to see the main results and findings of the work.

**Response**: *We have revised whole of manuscript and reformulate the results section for the revised version.*

**Referee #1:** p. 3, line 5. I think "somewhat recovered" is overstated. Hard to tell from Figure 3. Maybe stabilized.

**Response**: *We agree with you. So, we have revised the sentence as follow:*

*"After 2015, lake extent and storage have stabilized".*

**Referee #1:** p. 7. What is total water storage anomaly (TWSA)? This term seems rather central to the paper. Please explain.

**Response**: *We have inserted the following sentence as the second sentence of the GRACE section (section 2.2.1):*

"*TWSA describes the total amount of water stored on the continents, including water storage in surface water bodies, groundwater and soil, as compared to the mean value of total water storage over a reference period.*"

**Referee #1:** p. 8, lines 13-18. This method of applying (1- return flow multipliers) to the abstractions to estimate consumptive use assumes that water is used by only one water user. Is this a realistic assumption? If the return flow is used by another agricultural user and then again by a 3rd or 4th user, the basin-wide consumptive use fraction will be much different than the values reported. The large grid size magnifies this error. Table 1. How sensitive are study results to the values in this table?

**Response**: *In our opinion due to the fact that most return flow returns to groundwater, there is no concern in this regard. On the other hand, in arid area e.g. Urmia basin the return flow to surface water in each irrigation is not too much that can be used by another user. Anyway, we do not know exactly how the authors of the values determined them. However, the study results are not sensitive to the independent return flow estimates that were used only in one variant (RS_Q_GW_NA). We have written in section 4.1:*

*"Consideration of regional estimates of human water withdrawals in a specific year as well as regional estimates of return flow fractions in variant RS_Q_GW_NA does not improve the fit to observations significantly and only leads to slight parameter adjustments. This indicates a reasonable simulation of per hectare water consumption for irrigation by the WaterGAP model."*

**Referee #1:** p. 21, line 17, "The lower lake water loss…" What are the loss terms besides evaporation? How are these other loss terms smaller when inflow is larger? Explain.

**Response**: *Evaporation is the only loss term of the lake. There was less decrease in lake storage due to more inflow into the lake. We have revised the sentence as:*

*"The smaller decreasing trend for lake water volume under naturalized conditions is clearly caused by more inflow into the lake, even though lake evaporation is somewhat higher under naturalized inflow conditions due to the larger lake extent."*

**Referee #1:** p. 21, lines 25-28. I don't follow. What is the connection between the first part of the sentence and the second part?

**Response**: *We wanted to state that inflow was less than the minimum environmental water requirements (3,085·10⁶ m³) and that therefore a loss of lake water volume is expected. So, the first part of the sentence indicated under anthropogenic situation the inflow into the lake since 2008 never has reached to minimum environmental water requirements. The second part is related to naturalized condition which showed that under naturalized condition only in 2008 and 2009 the inflow into the lake were less than minimum environmental water requirements. We intend to reformulate the first sentence as follows:*

*"Since 2008 the inflow into the lake has never reached 3,085·10⁶ m³/yr. This is the value estimated to be the minimum environmental water requirements that compensates the amount of annual evaporation from of the lake surface (Abbaspour*

*and Nazaridoust, 2007). Therefore, a decrease of lake water storage can be expected for the best estimate of WaterGAP of 2,639·10⁶ m³/yr."*

**Referee #1:** p. 24, line 29. 90% of what?

**Response**: *90% of groundwater storage losses. We have reformulated the sentence as follows:*

*"90% of groundwater storage loss is estimated to be caused by human water use but this value may be somewhat overestimated by WGHM because climate-driven loss under naturalized conditions may be underestimated due to the simplified representation of groundwater-surface water exchanges in the model."*

**Referee #1:** p. 24, lines 19-24. I disagree. There are lots of other similar systems in the world – Great Salt Lake, Owens Lake, Dead Sea, Ural Sea, etc. each satisfy the first two criteria listed. What of these results is generalizable?

**Response**: *Thank you for pointing this out. We do not know what is generalizable as we have not done this calibration exercise in other basins. We intend to add one sentence at the end of the paragraph that provides a recommendation based on our study.*

*"In basins with large lakes, and in particular with end lakes, remotely sensed time series on lake area and the elevation of the lake water table should be used to estimate time series of lake water storage as these observational data can be expected to be of high value for understanding the freshwater system by hydrological model calibration. Groundwater storage cannot be observed from space but relies on in-situ observations on groundwater heads in wells but, as in the case of Lake Urmia basin, but such data may be crucial for a correct understanding of the freshwater system."*

**Referee #1:** p. 24, lines 31-34. How do the model results inform the 2014-2017 trends? Also, how can climate change be constrained in this basin? Explain.

**Response**: *Unfortunately, almost all the input datasets are not available from 2014 onwards. Thus we did not have the model results for recent years. With "constraining climate change" we meant the global reduction of greenhouse gas emissions, nothing at the basin scale. We have revised this part as follows:*

*"Further strengthening of efforts for decreasing human water use in the basin should be undertaken, while at the same time, global-scale mitigation of climate change by reducing greenhouse gas emissions to prevent strong decreases of precipitation and runoff."*

**Referee #1:** Data availability. I don't follow. If the authors do not have permission to share the data, then how can they share by author request? The HydroSat site underwritten by the University of Stuttgart is neat. What is the original source data for Urmia? Also, there is no water storage anomaly data for Urmia.

**Response**: *We have no permission to share data except by personal request for research purposes. We have indicated the source of all data in the manuscript. The data for Lake Urmia was obtained from various sources. Regarding HydroSat site, website you can download original data for lake level and extend with related reference which is Tourian et al. (2015) after registration. Also about water storage anomaly data for Urmia, for the lake water storage anomaly, it should be noted that anomalies can be calculated easily by subtracting the mean lake water storage in baseline (2004-2009) from the lake water storage time series.*

*The section on data availability have been reformulated as follows:*

*"In-situ data from "Iran Water Resources Management Company" including groundwater levels, precipitation and temperature publicly are available upon request from the corresponding author. GRACE data is available through http://www2.csr.utexas.edu/grace/RL05_mascons.html (last accessed: 17 Jul. 2018). Lake water surface extents and water*

5 *levels are available at http://hydrosat.gis.uni-stuttgart.de/php/index.php (last accessed: 17 Jul. 2018). All simulation results are available in the supplement."*
* * *
Point-by-point response to the Referee #2:
* * *
**Referee #2:** It can be considered as an interesting update in vast literature of Lake Urmia studies while authors tried to use a vast variety of data between 2003 to 2013 to evaluate the situation of Lake Urmia. I think a Major revision are needed prior to

10 evaluating its technical quality. My comments are listed as bellow

**Response**: *We would like to thank you for the thorough consideration and critical comments that helped us improving the manuscript. We have tried to do our best to consider all your recommendations in the revised version. Below, we have provided a point-by-point response to your comments.*

**General comments:**

15 **Referee #2:** A technical proof reading is needed since some of the sentences are not understandable.

**Response**: *We have revised the whole manuscript and the manuscript will be checked by a native speaker.*

**Referee #2:** Your given figures and tables do not necessarily indicate to the discussion you have made.

**Response**: *We provided discussion on all figures and tables. We agree that some part of the discussion might not directly related to the figures and tables which is the lesson we learned from this study. However, we have re- written the result and*

20 *discussion section with considering your comment.*

**Referee #2:** Relative error in your models is important since figures are dimensionless. Still, given figures seems to have unacceptable errors.

**Response**: *We have calculated Relative Absolute Error (RAE). But we think you were misled because all figures have dimension. We assume that you did not follow the different calibration variants. If you see some unacceptable errors in the*

25 *figures, these errors are related to the variables which did not consider as an objective function in the calibration process (see different calibration variants).*

**Referee #2:** Methodology should be revised since it is not clear how you evaluated the figures in discussion

**Response**: *We do not understand the comment well. However, we have revised the methodology section.*

**Specific comments:**

30 **Referee #2:** You are suggesting that the Lake would have been vanished any way but there are a vast literature against your statement. What is your comments? You should also add sentences in the text about it.

**Response**: *We do not suggest at all that the lake would have vanished without human water use. For example, we stated the following:*

*"Still, even in the WGHM-NAT, the average inflow into the lake from 2009-2013 of 3,670·10⁶ m³ would have been only enough to keep the lake from further loosing volume but would not have been enough for a recovery to conditions between 2003 and 2007 (Fig. 9b),"*

*Also in the revised version section 4.2, we have indicated why some studies were not in line with our statement and vice versa as follows:*

*"In order to define the lake restoration program, it is vital to know which factors contribute how much to shrinkage of the lake. All previous studies (e.g. Hassanzadeh et al., 2012; AghaKouchak et al., 2015; Ghale et al., 2018; Chaudhari et al., 2018) agreed that shrinkage is caused by both climate variations and human activities, but there is no consensus about the relative contributions. For example, Chaudari et al. (2018) concluded that human-induced changes accounted for 86% of the lake volume decline during 1995-2010, while we determined the value of 40% for 2002-2013. According to our study, human water use was the reason for 41% inflow reduction into the lake during 2003-2013 which is similar to the values of Shadkam et al. (2016) for the years 2003-2009 (comp. their Figs. 8). Discrepancies are likely due to different analysis methods but different analysis periods, as well as different conceptualizations, make a direct comparison of the estimated relative contributions difficult.*

*While Ghale et al. (2018) seem to support the results of Chaudhari et al. (2018), as they state that 80% of drying of Lake Urmia is due to anthropogenic impacts during 1998-2010, there statistical analysis assumes that river inflow can be considered to reflect "anthropogenic impacts" while precipitation and evaporation changes reflect climatic variations while river inflow is in reality also affected by climate variations. Also using a statistical change point analysis and without modelling, Khazaei et al. (2019) stated that given the stable conditions of precipitation and temperature, climatic changes cannot explain the dramatic decline of the lake level. They did not use in-situ data (except lake water level data) for their analysis. Based on a analysis of Standardized Precipitation Index (SPI), a drought index, AghaKouchak et al. )2015) reported there was no significant trend in droughts over the basin during past three decades and concluded from this that human activities not climatic variations are the main reason lake shrinkage. Different from our study and the modelling studies of Shadkam et al. (2016) and Chaudhari et al. (2018), these three studies consider only the dynamics of monthly and annual precipitation, not taking into account the changes in the variability of daily precipitation. During the last three decades, there was a significant increase the frequency of daily precipitation of less than 5 mm and a significant decrease in the frequency of daily precipitation of 10-15 mm, suggesting a runoff reduction even in case of constant annual precipitation (Fig. 2 in Bavil et al., 2018). Hosseini-Moghari et al. (2018) showed that an increasing frequency of days with less than 5 mm precipitation in combination with decreasing monthly precipitation has lead to the observed reduced inflow into two dams in the Lake Urmia basin that are located downstream of areas with insignificant human water use. We conclude that analyses should be done at the daily time scale or smaller.*

*In addition, a comprehensive modeling approach is preferable that takes into account, for example, the impacts of changing temperatures on runoff and thus river inflow and on evapotranspiration of the lake itself. Such comprehensive modelling was done by Chaudhari et al. (2018) but their uncalibrated global hydrological model that represented the basin by 5-6 cells only was not able to simulate well the flows and storages in the basin. For example, annual inflow into the lake was estimated to*

5   *be $3,700 \cdot 10^6$ $m^3$ in 2003 (their Fig. 8) while observed inflow was much higher, $5,835 \cdot 10^6$ $m^3$. In 2009, observed inflow, with $1,036 \cdot 10^6$ $m^3$, was only half of the simulated one. Therefore, the very high human contribution to lake volume decline of 86% determined by Chaudhari et al. (2018) may arise from the poor performance of the uncalibrated model."*

**Referee #2:** One of the main factors in your study is the calibration of satellite data and application of filters on data which are main issues. E.g. In GRACE data how did you manage to use 2degree precision into such a cristal clear results?

10  **Response**: *We agree with you. We used satellite data for three objectives including the irrigated area, lake level (and extent) and TWSA. Irrigated area and lake level were taken from previous studies which they needed filters on data before application. Hence, among all used satellite data, here we only discuss GRACE data. To deal with your mentioned issue about GRACE data, we have used CSR mascon solutions product which based on Save et al. (2016) do not need any filters. Also, as we mentioned in the manuscript, we know that it is recommended GRACE data products only for areas with at least 100,000 $km^2$*

15  *(Watkins et al., 2015; Landerer and Swenson, 2012). But studies by Tourian et al. (2015) and Lorenz et al. (2014) showed that signal strength or the so-called gravimetric resolution is determining the applicability of GRACE data. In fact, Lake Urmia basin has experienced an $8 \cdot 10^9$ $m^3$ change in the water volume in the last decade, which allows the use of GRACE for monitoring the changes in water storage in the basin (Tourian et al., 2015). This fact is supported by the very small gain factor of 1.0083 for the Lake Urmia basin based on Community Land Model 4 (CLM4) for spherical harmonic solutions (Landerer*

20  *and Swenson, 2012), which is the factor with which signal attenuation due to leakage could be balanced. We can assume errors of the applied GRACE monthly time series of TWSA are small compared to the uncertainty of TWSA as computed by WGHM, such that model calibration against GRACE TWSA is meaningful.*

**Referee #2:** Why did you use a time length between 2003 to 2013? Since the decreasing trend have already started from late 90's.

25  **Response**: *The main reason was shortage of data. The GRACE data and irrigated area which play important roles in this study were not available for late 90's. As a results we faced substantial missing data before 2000.*

**Referee #2:** There are a lot of missing data in historical time series records of the region. How did you manage to remove them? are they satisfactorily acceptable methods to be applied?

**Response**: *We disagree. There are no significant missing data for 2003-2013. The method used for filling the gaps are as*
30  *follows:*

- *Irrigated area: There is no missing data in this data (except 2013).*
- *GRACE data: There are some missing data (8 months) in its observations which have filled using linear interpolation.*
- *Inflow into the Lake: There is no missing data in the dataset of annual inflow into the lake.*

- *Groundwater levels: We assessed data of 635 wells then we removed the wells with more than 12 months or six consecutive months missing values. After removing the wells with significant missing values, we have worked on 284 wells. If there are missing data in the dataset of these 284 wells, we used linear interpolation (if only one month is missing) or linear regression (with the nearest well in common period) for filling the gaps in data.*
- *Water withdrawals: There is no missing data in this data.*
- *Precipitation: Almost there is no missing data in the studied stations between 2003-2013. Few missing data filled in comparison with near stations.*
- *Temperature: There is no missing data in this data.*
- *Lake volume: There are some missing data for lake volume which have filled using linear interpolation.*

**Referee #2:** Page 9, Line 20, Calibration: You have to give the error evaluation if you have used "try and error" method.

**Response**: *We used trial and error to determine the most appropriate parameters of model in each calibration of variant based on the evaluating the model error with respect to the observations used, while trying to keep the number of adapted parameters at a minimum. We provide the final errors in Table 4. We do not think that it is interesting for the reader to provide the errors/model performances for all trials (there were many).*

**Referee #2:** Page 12, Performance criteria, Line 5: These criteria do not show relative error in your models since RMSE number is not necessarily satisfactory.

**Response**: *We have calculated Relative Absolute Error (RAE) in the revised version.*

**Referee #2:** Page 13, Figure 7: None of these figures indicate to an acceptable calibration.

**Response**: *This figure does not show any calibration results. This figure shows the inputs of model in different variants, not the model's output. Thus, there is no reason to fit the lines in this figure.*

**Referee #2:** Page 14, Figure 8d: The discrepancy and error is growing in your anomalies. How do you interpret?

**Response**: *It should be noted that Figure 8d shows change not anomalies. Anyway, we agree that the error in the second half of the time series is bigger than the first half. However, the difference in errors is minor. Also, discrepancies in groundwater storage (e.g. in peak of seasonality) can represent some minor discrepancy in groundwater storage changes. If you consider Figure 8c, which shows the normalized groundwater storage, there is no growing in discrepancy.*

**Referee #2:** Page 18-23: Your discussions are too long, yet none of them are visuable from given tables and figures. It is a very long article, yet given informations are narrative and reader should accept your sentences without having a chance to approve it.

**Response**: *We have re-written the results and discussions considering your comment.*

*We have done our best to apply all your comments and suggestions in the revised version. Below you can see the changes have applied based on your comments.*

**Referee #2:** Relative error in your models is important since figures are dimensionless. Still, given figures seems to have

5   unacceptable errors.

**Response**: *We have calculated Relative Absolute Error (RAE). But we think you were misled because all figures have dimension. We assume that you did not follow the different calibration variants. If you see some unacceptable errors in the figures, these errors are related to the variables which did not consider as an objective function in the calibration process (see different calibration variants).*

10  **Referee #2:** Methodology should be revised since it is not clear how you evaluated the figures in discussion

**Response**: *We have added the following sentences in section 2.4*

*"Trends and overall behaviour of the time series were also analysed."*

**Referee #2:** You are suggesting that the Lake would have been vanished any way but there are a vast literature against your statement. What is your comments? You should also add sentences in the text about it.

15  **Response**: *We do not suggest at all that the lake would have vanished without human water use. On page 21 line 28, for example, we state the following:*

*"Still, even in the WGHM-NAT, the average inflow into the lake from 2009-2013 of 3,670·10$^6$ m$^3$ would have been only enough to keep the lake from further loosing volume but would not have been enough for a recovery to conditions between 2003 and 2007 (Fig. 9b),"*

20  *Also in the revised version section 4.2, we have indicated why some studies were not in line with our statement and vice versa as follows:*

[revised manuscript text omitted]

*Beside reviewers' comments and suggestions, we also consider the short comments in the revised version. Below the changes have applied based on the short comments have been written.*
* * *
The changes have applied based on the comments of S. Chaudhari as short comment
* * *
**Comment:** The authors should use contrasting colors in figures. It is difficult to distinguish the WGHM-ANT and WGHM-NAT lines in Fig 9 due to similar colors.

**Response:** We have changed the color of WGHM-NAT in the revised version.
* * *
The changes have applied based on the comments of S. Khatami as short comment
* * *
**Comment:** Other metrics such as Willmott's refined index of agreement [Willmott et al., 2012] and KGE [Gupta et al., 2009] shown to be better than NSE.

**Response:** *We have calculated the KGE in the revised version.*

**Comment:** Using the term validation is both semantically and theoretically wrong. As a matter of good practice, it's been recommended to use the term evaluation instead of validation.

**Response:** *"validation" was replaced with "evaluation".*

**Comment:** You have used CC and NSE (P 9 L 11-12) to cross compare precipitation and temperature records of difference sources. It is better to use (more) resistant alternatives such as Spearman ranked correlation (instead of Pearson correlation) and Willmott's refined index of agreement [Willmott et al., 2012]

**Response:** *We have calculated the Willmott's refined index of agreement in the revised version.*

**Comment:** this is an unsubstantiated claim. As far as I know there is no (reliable) evidence on the degree of awareness regarding this issue. Please remove it, or provide the evidence.

**Response**: *We have removed the sentence.*

**Comment:** P 25 L 9: It is better to explicitly acknowledge the organisations that provided you with GRACE and climate data, and the URL links if available.

**Response**: *We have explicitly acknowledged the organizations that provided the used data. The URL links are available in the text.*

**Comment:** P 28-29: The URLs in the ULRP references are not accurate. Please update them.

**Response**: *We have updated the URLs.*

*\*\*\*\*\*\*\*\*\*\*\*\*\*\*\*\*\*\*\*\*\*\*\*\*\*\*\*\*\*\*\*\*\*\*\*\*\*\*\*\*\*\*\*\*\*\*\*\*\*\*\*\*\*\*\*\*\*\*\*\*\*\*\*\*\*\*\*\*\*\*\*\*\*\*\*\*\*\*\*\*\*\*\*\*\*\*\*\*\*\*\*\*\*\*\*\*\*\*\*\*\*\*\*\*\*\*\*\*\*\*\**

[revised manuscript text omitted]

---

## Referee Report (RR1)

Journal: Hydrology and Earth System Sciences
Title: Quantifying the impacts of human water use and climate variations on recent drying of Lake Urmia basin: the value of different sets of spaceborne and in-situ data for calibrating a hydrological model
Article Iteration: revision 1

**1 Overall assessment**

The aim of the study essential is to quantify the impact of human activities (mostly in terms of water consumption) vs climatic changes on the Lake Urmia water balance. Even though 3 of the referees provided detailed reviews and pointed out to several shortcomings of the paper mostly on model setup and uncertainty, the authors' revision is minimal and in fact insufficient as many comments are effectively ignored.

All reviewers except Chaudhari took issue with the experiment design, particularly the model set up, input data time period, and lack of adequate model calibration and evaluation. Yet, authors have not changed the experiment design, and only added two performance metrics. No uncertainty or sensitivity analysis was conducted whatsoever, which is a common analysis required for any hydrological modeling study. This is even more serious as the manuscript is in fact inconsistent on the issue of uncertainty. While authors discussed the limitations of the work such as parameter uncertainty, no account of the hydrogeology of the lake, assuming constant bathymetry, among others; not only they have not accounted for these uncertainty sources by even a simple uncertainty/sensitivity analysis, they kept pushing that their study is "*a holistic and reliable modelling approach*", "*we are confident that human water use reduced lake inflow that would have occurred without human water use during 2003-2013 by about 41%*", and "*This study proved that even without human water use Lake Urmia would not have recovered from the significant loss of lake water volume caused by the drought year 2008*", among other instances of false overpromises.

The manuscript is filled with redundant discussions (either well established in the literature or not relevant to the core research question of the paper), and is written in poor language with several typos. It is a waste of the editor's time and reviewers to resubmit a manuscript that has not been proof-read especially that most reviewers pointed this out. Further, some of the sources are either not peer-reviewed or in Persian. While local knowledge can be useful, the credibility of a non-peer reviewed source is always questionable. In such cases, authors should provide reason and demonstrate clearly instead of *just* referring to the source.

In my evaluation, the manuscript is **rejected**. If authors wish to resubmit the work, they must revise the modeling experiment to sufficiently address the comments by reviewers, and by addressing I specifically mean to change their modeling setup by providing a more robust calibration than a simple and

insufficient trial and error, transparently explaining the modeling setup to ensure (somewhat) the re-producibility of the study, and perform some sensitivity or uncertainty analysis. Also, remove all the redundant discussions from the manuscript and focus their discussion on the relevance of the results to the lake given the uncertainties. While most of the comments by reviewers still hold, here I high-lights a few urgent ones.

**1.1. Problem description**

The problem description (i.e. drivers of the lake desiccation) particularly in the introduction as well as the later discussion of the results is problematic.

It is misleading and inaccurate to stack the climatic and human drivers together (e.g. page 29 lines 1-10). *Khazaei et al.* [2019] disentangled these two: they compared the influence of these two sets of drivers for the lake drying and demonstrated the regional human activities (including water manage-ment, but not limited to) are the primary drivers compared to climatic changes (including atmospheric droughts). *AghaKouchak et al.* [2015] also argued that droughts cannot be the primary driver. These two studies, among others, are based on directly analyzing the data themselves, without relying on a model of the system which in most cases are inadequate. While these studies have their own short-comings, as any scientific study has, your modeling results are inadequate to challenge them. As pointed out by the reviewers your modeling setup has several issues. Inadequate models, regardless of the extent of their inputs and their results, are inadequate.

Authors said: "*This study proved that even without human water use Lake Urmia would not have re-covered from the significant loss of lake water volume caused by the drought year 2008*".

This conclusion is in direct contradiction with *AghaKouchak et al.* [2015] conclusion that "*a satellite-based gauge-adjusted climate record… of Lake Urmia basin's Standardized Precipitation Index… indi-cates no significant trend in droughts over the past three decades at the 0.05 significance (95% confi-dence) level... In fact, the region has experienced more severe drought events in the past (e.g., 1997–2002) that did not lead to a substantial change in the lake's surface area. Thus, we caution against overrating the role of droughts in the disruption of the lake's water balance to the extent that would cause such a massive shrinkage*". Given that *AghaKouchak et al.* [2015] directly analyzed the historic data of the lake without relying on any inadequate model of the lake system, it is reasonable to say this contradiction indicates the shortcoming and (un)reliability of the model set-up in this study. Not-withstanding the unscientific language of this sentence. Science is not in the business of proving any-thing. In science we demonstrate and approximate. This is more so the case when it comes to hydro-logical modeling with numerous types and sources of uncertainty including both model structure and data.

Also, what do you mean by "climate variations"? Are you referring to only *natural climate variability* or *climatic changes* which include both natural variability and human-induced changes?

**1.2. Introduction**

The introduction is very long, has redundant sections:

- Figure 3 has already been published and discussed by Aghakouchak and been repeated many times in the literature. (also pointed out by the referee 1)
- Citation to Zarghami (2011) on page 29 is irrelevant.
- Last 2 paragraphs are "method" material and not introduction

Another issue is that some of the sources are not peer-reviewed. Whether right or wrong, it is hard to rely on such sources. While local knowledge and literature may be a valuable source, it has to treated with caution, not to propagate any errors. So, I am hesitant to accept such discussions.

**1.3. Modeling setup and results**

**Time period:** the time period 2003-2013 is inadequate for modeling the lake dynamics. Before 2000 the lake was not as heavily impacted by over-regulation of the river flows, and also between 2000-2003 there is significant variation in the lake level and annual inflows to the lake. Therefore it is essential to include these years, for as many variable as possible. Otherwise, the model is biased and not representative of the lake dynamics.

**Model setup:** as pointed out by R1 comment 2 (and a few other comments), the model setup is not transparent and justifiable. Surely, documenting model setup (even as supplement) is more necessary than many redundant discussions. As a test, an adequate model setup should work against the updated observation (2013-present). Can the authors demonstrate this?

**Standard model:** inclusion and discussion of the standard model, as a reviewer pointed out, in unnecessary given that this model is not calibrated for this catchment.

**Model calibration and evaluation:** the model calibration is also questionable. It is only based on trial and error so the identified parameter sets are not reliable. Also, it is not clear how sensitive the model results are to these parameters. There is issue about over-parametrizing the model as in each variant new data is added (issue about correction factors pointed out by reviewers). A single model run for each variant is not sufficient, even if the calibration was done through an automatic parameter space search scheme. This is well-established in the literature and a model ensemble is required to account for the uncertainties, even tough partly. If the model is run on a daily basis, authors should be transparent and present the daily results too/

Monthly and annual performance metrics are usually high for most models. The devil is in the details though. On table 4, the flow is only calibrated on an annual scale and not evaluated at all. There is significant seasonality in this region. It is quite possible that seasonal errors are just canceling each other out and give seemingly good annual results. This can be seen on figure 8 where the model exhibits unrealistic seasonality which is not in the observed data, e.g. in Fig 8e, there is generally a negative bias in the first half of the RS_Q_GW_NA simulation, and positive bias in the second half (red line is first below the black line systematically, and then above it).

**Uncertainties:** as Referee 1 pointed out in their comment 6 and elsewhere, the uncertainties are playing a crucial role here. While authors added a section of uncertainty discussion, it is an ad-hoc discussion. While the discussion obviously undermines the experiment design and the results, they have not revised the experiment design to account for these uncertainties, and also they keep overpromising about the reliability of their results. Further, their discussion of uncertainty shows a lack of understanding about the area of model uncertainty. For instance, they said "*Model parameter uncertainty was reduced by the comprehensive multi-observation calibration*". Parameter uncertainty will not be reduced by *just* adding more data to the model; data uncertainty matters, "*garbage in, garbage out*" [*Kuczera et al.*, 2010]. The authors must demonstrate how adding input to the model reduced parameter uncertainty, while justifying the credibility of the data themselves. They have not done any uncertainty or sensitive analysis whatsoever.

**References**

AghaKouchak, A., H. Norouzi, K. Madani, A. Mirchi, M. Azarderakhsh, A. Nazemi, N. Nasrollahi, A. Farahmand, A. Mehran, and E. Hasanzadeh (2015), Aral Sea syndrome desiccates Lake Urmia: Call for action, *Journal of Great Lakes Research*, *41*(1), 307-311.

Khazaei, B., S. Khatami, S. H. Alemohammad, L. Rashidi, C. Wu, K. Madani, Z. Kalantari, G. Destouni, and A. Aghakouchak (2019), Climatic or regionally induced by humans? Tracing hydroclimatic and land-use changes to better understand the Lake Urmia tragedy, *Journal of Hydrology*, *569*, 203-217.

Kuczera, G., B. Renard, M. Thyer, and D. Kavetski (2010), There are no hydrological monsters, just models and observations with large uncertainties!, *Hydrological Sciences Journal*, *55*(6), 980-991.

---

## Referee Report (RR2)

Journal: Hydrology and Earth System Sciences (HESS)
Title: Quantifying the impacts of human water use and climate variations on recent drying of Lake Urmia basin: the value of different sets of spaceborne and in-situ data for calibrating a hydrological model

**1  Overall assessment**

First and foremost, I would like to sincerely apologize for my late reviews – due to family reasons. I have now reviewed the new manuscripts and authors' responses to previous reviews. By and large, the authors have addressed most of the comments, updated the modelling, condensed the manuscript and improved the use of language, and improved the discussion of results (particularly adding a new section on the limitations of the study). So, I would like to thank authors for their efforts in this round of the review.

I would like to also thank authors for dedicating a section on **limitations (section 4.3)**. Not only it is good practice for scientific studies, but also it directs readers to design future studies to address past limitations. That said, a few major points are missing in the discussion of limitations:

- The study period is limited and do not include some of the most important years of the lake (and the basin) old/recent history and evolution, e.g. recent changes in the lake since 2013 (beyond the study period of this manuscript). Compared to most hydro-climatology studies of the lake, this study is based on a very short period (2003-2013), and hence generalizing its results beyond this domain is difficult, particularly due to the non-stationarity of the lake system. I expanded on this in section 3 of this review.
- Role of dam constructions, groundwater withdrawals and its hydrological connectivity to the lake, and seasonal variations of the flow overlooked in the model calibration.

Moreover, **section 4.2 "*Comparison to human vs. climatic contribution as determined in previous studies*"** comes short of providing an adequate and accurate characterization of the ongoing debate within the literature:

- I acknowledge that the Lake Urmia desiccation has been an ongoing contested debate, i.e. whether the main driver of drying is management-related and human activities or climatic. This very question is indeed the crux of the matter, and hence it is in the best interest of both authors and readers to be more rigorous on this discussion. As opposed to several previous studies, this study puts more weight on the climatic drivers of the lake drying. However, the authors (in section 4.2) misrepresented or overlooked some of those studies which argued for human activities over climatic drivers. Regardless of my personal position in this debate, authors are unduly framing the results and merits of the previous studies to justify their own

side of the argument. Further, they have failed to discuss a few important studies on the lake. I demonstrate this in section 2 of this review. To help the authors, I discussed several points in details. I have also suggested few additional references and edits throughout. Therefore, in my opinion section 4.2 of the manuscript is inadequate and must be improved.

In my evaluation, the manuscript would be **accepted upon the suggested minor to moderate revisions**.

Below you could read my new reviews and replies to a few *responses of authors (in purple)* to my earlier reviews. The "*quotations*" are from the final version of the manuscript.

**2 Section 4.2 "human vs. climatic contribution"**

The discussion is not elaborative, lacks a clear discussion line, and previous studies are not appropriately represented/discussed. Here I discuss a few points in this section as an example, to help authors better discuss this important point.

Page 18 line 17-21: "*Chaudhari et al. (2018) concluded that human-induced changes accounted for 86% of the lake volume decline during 1995-2010, while we determined values of 39-43% for 2003-2013. According to our study, human water use was the reason for 39-45% inflow reduction into the lake during 2003-2013 which is very similar to the values of Shadkam et al. (2016) for the years 2003-2009 (comp. their Fig. 8). Discrepancies are likely due to different analysis methods but different analysis periods and conceptualizations make a direct comparison of the estimated contributions difficult.*" Re the underlined part: it is a general statement and not good enough to simply overlook the details leading to these difference. It is essential to discuss in more details what are the main differences between these studies e.g. in terms of data type, analysis approaches, fundamental assumptions, etc. For instance, Chaudhari et al. (2018) studied a considerably longer period. They also studied the land use changes in detail: over 1987-2016 showed ~98% and ~180% increase in agricultural lands and urban areas, respectively. They accounted for human impact during 1995-2010 (based on simulation of streamflow into the lake). Various studies identified two distinct periods of pre- and post-change in the lake dynamics, e.g. Khazaei et al. (2019) identified year 2000 as the change point and Fazel et al. (2017) identified year 2001. Given that, studies such as Chaudhari et al. (2018) take into account a wider range of the non-stationarity of the lake than the present study where only a part of the post-change period is investigated. It is plausible to expect that if your model was successfully calibrated over a longer period including years prior to 2000, it would have lead to different results.

"*While Ghale et al. (2018) seem to support the results of Chaudhari et al. (2018) as they state that 80% of drying of Lake Urmia is due to anthropogenic impacts during 1998-2010, their statistical analysis assumes that lake inflow from rivers can be considered to reflect "anthropogenic impacts" while precipitation and evaporation reflect climatic variation. However, inflow is in reality also affected by climatic variations.*" Your argument here is incomplete, as the impact of climate vs. human activities on river networks is different for headwaters and lower river reaches. Fazel et al. (2017) investigated this in detail, analyzing the flow regime changes across the lake basin (57 flow gauging stations) over the period 1949-2013 (perhaps the longest record of the basin flow studied so far). Their study showed that while "*flow regime in river headwaters appeared to be dominated by natural forces*", "*the reduction in river flow magnitude increased from headwaters to downstream reaches for all rivers*" due to dam river regulations and dam constructions. They further argued that "*Changes in river flow in the period 1965–2013 cannot be explained by climate change, the effects of which occur much more slowly than those of land use change in the region*". They concluded that "*The results showed that irrigation was by far the main driving force for river flow regime changes in the lake basin. All stations close to the lake and on adjacent plains showed significantly higher impacts of land use change than headwaters. As headwaters are relatively unaffected by agriculture, the non-significant changes observed in headwater flow regimes indicate a minor effect of climate change on river flows in the region*".

"*Using a statistical change point analysis and without modelling, Khazaei et al. (2019) stated that given the stable conditions of precipitation and temperature, climatic changes could not explain the dramatic decline of the lake level; however, they did not use in-situ data (except lake water level data) for their analysis*" Study by Khazaei et al. (2019) is more than a simple statistical change point analysis, they estimated the land use change (particularly vegetation dynamics and its associated hydrological loss in terms of evapotranspiration) and trends of various hydro-climatic variables across various time scales. While their study surely has its own limitations, lack of modeling and use of in-situ data are not the major limitations – let alone this is too generic for a scientific criticism. One of the major limitations of their work, for instance, is that they did not account for the role of groundwater dynamics in their analysis.

"*For quantifying human and climatic contributions to observed hydrological changes, a comprehensive modeling approach that takes into account, for example, the impacts of changing temperatures on runoff and thus river inflow and on evapotranspiration of the lake itself is preferable.*" Preferable to what exactly? I tend to disagree that modeling is preferable to comprehensive analysis of historical data. Modeling introduce various sources of new uncertainty to a problem (such as model structural uncertainty, parameter uncertainty, over-parameterization, parameter transferability across time and space, etc.), which are not preferable to the simplifying assumptions underlying statistical analyses (such as trend, correlation, or linear regressions). In

general, I believe, both approaches of modeling and data analysis can inform us in some ways, while each has its own shortcomings in other ways.

"*Chaudhari et al. (2018) but their uncalibrated global hydrological model that represented the basin by 5-6 cells only was not able to simulate well the flows and storages in the basin.*" This is a mischaracterization of Chaudhari et al. (2018), undermining their extensive modeling setup and evaluation. Although Chaudhari did not explicitly discuss the model setup and calibration, they demonstrated the adequacy of their model by evaluating various model outputs against available knowledge and data of the LU basin. For instance, they compared their simulation inflow to the lake with the observed inflow record (previously gathered by Hassanzadeh et al., 2012). As the figure shows it is in good agreement. I agree with the authors' intent to critically review previous studies to elaborate their shortcomings, however this must be done rigorously and accurately.

[Figure]

**Fig. 7.** Comparison of simulated river inflow to Urmia Lake from HiGW-MAT model and the inflow data from Hassanzadeh et al., (2012).

Page 19 lines 2-5: "*Hosseini-Moghari et al. (2018) showed that an increasing frequency of days with less than 5 mm precipitation in combination with decreasing monthly precipitation has led to the observed reduced inflow into two dams in the Lake Urmia basin that are located 5 downstream of areas with insignificant human water use.*" This study is not available online and it is not possible to confirm whether it is peer-reviewed or not.

"*We conclude that analyses should be done on a daily time scale or smaller.*" What type of analyses exactly? Such a generic statement. Needless to mention that the very present study of the authors is not done on a daily time scale either. The time scale of a study depends on its objective.

"*we examined the ratio of annual inflow into the lake (based on the ensemble mean) over annual precipitation during the study period. This ratio reached maximum values in 2003 (0.29 and 0.41 for the anthropogenic and naturalized conditions, respectivly) and minimum values in 2009 (0.07 and 0.15). Averaged over the period 2009-2013, these ratios are, with 0.11 (ANT) and 0.22 (NAT), much smaller than the values for 2003-2007, 0.20 and 0.32. Thus, the drought year 2008 as well as the relatively small ratio of inflow into the lake over precipitation in the last five years of the study period play a significant role in the decline of inflow and lake water storage*" There are various issues with this argument. First, the period 2009-2013 is a very short period to build a hydro-climatic analysis on, particularly for LU with remarkable non-stationarity. So, the naturalized scenario based on this

period is not reliable. Second, the considerable extraction of groundwater resources has been an additional source of water for irrigation and consequently hydrological loss in this basin. The impact of groundwater withdrawal (and its consequent hydrological loss) would have had a direct impact on the lake and possibly on streamflow generation in the basin as well (e.g. as the land coverage of the basin has changed). Urbanization in this basin (discussed by Chaudhari et al. (2018)) together with the expansion of agricultural and irrigated areas would have an impact on streamflow generation (both magnitude and generation mechanisms).

"*For quantifying human and climatic contributions to observed hydrological changes, a comprehensive modeling approach that takes into account, for example, the impacts of changing temperatures on runoff and thus river inflow and on evapotranspiration of the lake itself is preferable.*" Also, estimating the impact of land use change (e.g. urbanization and cropland expansion) on runoff generation in the basin.

**2.1.    On the role of atmospheric drought**

**Figure 2 and the last paragraph of page 2:** This figure and its associated text provide an **incomplete** overview of the lake dynamics. The decline of the lake water level started around the year 2000, which is way more abrupt than 2003 onwards.

Page 1 line 30: "*The study shows that even without human water use Lake Urmia would not have recovered from the significant loss of lake water volume caused by the drought year 2008.*" First, you have not provided any evidence that the drought year 2008 caused a significant loss in lake volume, this causal link is non-existent in your study. The authors are trying to over-emphasize the role of atmospheric droughts, specially the 2008 one. There has been stronger atmospheric droughts in previous years than year 2008. Here is a figure from Alborzi et al. (2018). The historic droughts during 80s and early 90s are more severe than the 2008 drought, yet the lake has survived (AghaKouchak et al., 2015).

[Figure]

**Figure 2.** Key attributes of the lake-basin system prior to restoration program in 2013, including observed lake level, standardized precipitation index (SPI), basin-scale naturalized runoff, and surface water withdrawal. The basin's recent wet (blue) and dry (red) periods are illustrated in SPI and naturalized runoff curves. Post-1998 drop in lake level corresponds to a substantial increase (~25%) in surface water withdrawals during the prolonged drought of 1998–2002.

**2.2. Irrigated area**

One aspect that has not been discussed in section 4.2 is the irrigated area: how it is differently estimated by different studies and its implications. Below I have extracted figures corresponding to the estimated irrigated areas by 3 different studies.

Supplement page 3 line 5: "*Considering that water management in the basin aims at preventing any increase of irrigated areas, it is assumed that the*

[Figure]

**From the present study**

**From (Alizade Govarchin Ghale et al., 2019)**

*irrigated area in 2013 remained at the 2012 value (Fig. S3)*". This assumption is questionable. For instance, Alizade Govarchin Ghale et al. (2019) estimated the irrigated lands to decrease by ~12% from 2012, but again increase in 2018-2019. Further, their estimated irrigated area is very different from the present study: year 2003 is different by ~500 km$^2$ (~12%). The trend is also different, e.g. the increase during 2007-2011, or during 2003-2005.

While both this study and Alizade Govarchin Ghale et al. (2019) estimated the irrigated area based on the overall vegetation coverage, Chaudhari et al. (2018) made a distinction between natural vegetation and cropland. They showed (see the figures below) that the while the natural vegetation has oscillated throughout the years (1998 to 2006 → decreased, 2006 to 2011 → increased, and 2011

[Figure]

**From Chaudhari et al. (2018)**

to 2016 → decreased), the cropland has continuously increased. Moreover, they estimated the

annual net irrigation requirement (NIWR) during 1980-2010 based on the crop evapotranspiration (FAO Penman Monteith approach), independent of the global hydrological model they used, and compared it with estimations based on Landsat classification (see Figure 9 in their study). So, their estimated irrigation is independent of how well or poorly their model was calibrated, and arguably more comprehensive than your study. While the present study demerited Chaudhari et al. (2018) (page 19 of the manuscript) and entirely overlooked Alizade Govarchin Ghale et al. (2019) and Fazel et al. (2017), the authors failed to acknowledge that these studies delved deep into land use changes, irrigation water requirement, and flow regime changes.

**3 Comments on "Section 4.3 Limitations"**

**3.1. Study period**

Here I would like to allude to a previous comment of the review process.

**Reviewer comment:** the time period 2003-2013 is inadequate for modeling the lake dynamics. Before 2000 the lake was not as heavily impacted by over-regulation of the river flows, and also between 2000-2003 there is significant variation in the lake level and annual inflows to the lake. Therefore it is essential to include these years, for as many variable as possible. Otherwise, the model is biased and not representative of the lake dynamics.

*Authors' response: We have considered this period due to the fact that the observed data was available for this period. We completely agree with you; it was better to consider a longer period for calibration. However, we don't prefer to reconstruct data, that is error-prone. The GRACE data and irrigated areas are not available for the period 2000-2003. Further, we don't want to use the model for out of calibration period, therefore we believe that for using the model in the calibration period there is no concern about the bias.*

**Reviewer's response:** the point I argued is not simply about the length of data and model calibration. There are major aspects of the lake dynamics (and the basin evolution) that falls outside the 2003-2013 period. While you evaluated your model within the calibration period using an independent variable, you tend to generalize your findings about the lake beyond the limited period of 2003-2013. To study the drivers of the lake drying, it is not adequate to build up your entire argument based on a limited time period that does not include the non-stationarities of the lake system: various studies identified two periods of pre- and post-change for the lake, e.g. Khazaei et al. (2019) identified year 2000 and Fazel et al. (2017) identified year 2001 as the change point. Given that, your study does not cover the pre-change period, and both anthropogenic and natural scenarios are defined based on only a sub-period (2003-2013) of the post-change period (2000 to date), which biases the scenario

analysis. Further, most recent changes in the lake system is also not discussed. The lake has experienced considerable changes since 2013, e.g. see the extensive study by (Alizade Govarchin Ghale et al., 2019) on the land use changes within the lake basin. The figure below (extracted from Alizade Govarchin Ghale et al. (2019) shows the historic surface area as well as its increase since 2013 – evidence of remarkable non-stationarity. To what extent your modelling assumptions and results are compatible with this non-stationarity, particularly the most recent changes of the lake?

[Figure]

Fig. 4. Salinization and desertification progress in Urmia Lake from 1975 to 2018.

**3.2. Other limitations and suggestions**

**Role of dams:** Another aspect of the lake system that you did not accounted for explicitly is the dam construction within the lake basin over the past decades (24 dams were constructed during 1970-2000, and 32 during 2000-2014), which studies such as Fazel et al. (2017) and Alizade Govarchin Ghale et al. (2018) accounted for explicitly.

**Seasonal flow and model calibration:** While all variables are calibrated/evaluated on a monthly basis, the streamflow is calibrated on annual scale. I suspect that it is due to the fact that the model could not adequately represent the seasonal variations in streamflow, which are significant for this basin (Alizade Govarchin Ghale et al., 2019; Fazel et al., 2017). The seasonal variations of the flow have direct implications on irrigation estimations and the lake dynamics.

**Groundwater withdrawal and its hydrological connectivity to the lake:** The groundwater withdrawal is under-estimated in the model setup (a point that has been raised by reviewers before). While authors stated that "*Observed decline of groundwater storage was 1.8·10⁹*

[Figure]

**From LURP report (in Persian, attached to this review).** Red is withdrawal from deep wells, and green is withdrawal from partial deep wells, and blue is the total extraction.

*m³, i.e. 18% of the observed total water storage loss in the basin*" (page 17 line 20), the groundwater withdrawal (including both shallow and deep wells, see the figure) shows at least 2.1 MCM withdrawal in the past 2 decades.

Also, as discussed by Danesh-Yazdi and Ataie-Ashtiani (2019) the hydrologic connectivity between the lake and groundwater remains an under-studied aspect of the lake dynamics – which is a general limitation of most studies including the present one.

Page 20 lines 1-3: re lake bathymetry please also cite the below studies:

- Sima, S., & Tajrishy, M. (2013). Using satellite data to extract volume–area–elevation relationships for Urmia Lake, Iran. *Journal of Great Lakes Research*, *39*(1), 90-99.

- Karimi, N., Bagheri, M. H., Hooshyaripor, F., Farokhnia, A., & Sheshangosht, S. (2016). Deriving and evaluating bathymetry maps and stage curves for shallow lakes using remote sensing data. *Water Resources Management*, *30*(14), 5003-5020.

**4   Minor comments**

Page 3 line 10: "*Studies on various aspects of the Lake Urmia disaster abound. With decreasing lake water volume, salt concentration has increased*". Please cite the recent study on salt concentration as a dust source:

- Boroughani, M., Hashemi, H., Hosseini, S. H., Pourhashemi, S., & Berndtsson, R. (2019). Desiccating Lake Urmia: A New Dust Source of Regional Importance. *IEEE Geoscience and Remote Sensing Letters*.

Page 3 line 11: "*Precipitation reduction, temperature increase, agricultural development including construction of man-made dams and building a causeway across the lake have been identified as the reasons for the degradation of Lake Urmia (Abbaspour and Nazaridoust, 2007; Zeinoddini et al., 2009; Delju et al., 2012; Jalili et al., 2012; Sima and Tajrishy, 2013; Fathian et al., 2014; Farajzadeh et al., 2014; Banihabib et al., 2015; AghaKouchak et al., 2015; Azarnivand and Banihabib 2017; Alizadeh-Choobari et al., 2016; Ghale et al., 2018; Khazaei et al., 2019)*". Please separate out the references and cite relevant references for each factor (underlined phrases) individually. It helps the readers to track back.

Page 4 line 25: "*a good fit of simulated and observed streamflow may not necessarily lead to an appropriate simulation of other flows and storages (Beven and Freer, 2001). Therefore, additional types of observations have to be added to avoid equifinality (Beven and Freer, 2001; Döll et al.,*

*2016).*" The second sentence does not follow the first sentence, and using "therefore" does not make sense here. Also, by adding further data types, one will not "avoid" equifinality, because equifinality is a general property of open complex systems (e.g. hydrological models) and cannot be avoided. The goal is to "reduce" equifinality when possible. Please also cite the following recent studies on equifinality which are directly relevant to the discussion:

- Kelleher, C., McGlynn, B., & Wagener, T. (2017). Characterizing and reducing equifinality by constraining a distributed catchment model with regional signatures, local observations, and process understanding. *Hydrology and Earth System Sciences*, *21*(7), 3325.

- Khatami, S., Peel, M. C., Peterson, T. J., & Western, A. W. (2019). Equifinality and flux mapping: A new approach to model evaluation and process representation under uncertainty. *Water Resources Research*, 55, 8922– 8941.

Page 12 line 17 "*We determined that the results of the naturalized run differ by less than 2% from a run with reservoirs but without human water use*". First, it is not clear 2% of what is discussed here exactly. Second, such a small difference between the two scenarios is clearly a red flag, indicating that the model setup and/or scenarios are problematic. Most of the recent studies concluded that the lake condition is heavily impacted by human water use.

Page 14 line 20 "*In this way, efficient simulation of regional water flows and storages can be achieved, possibly as an alternative to a costlier setup of a regional model*". I'm not sure if I understood this part. What is costly about a regional model that is discouraging? What do you exactly mean by "setup a regional model", do you mean to develop a model from scratch?

Page 18, reword the title of the subsection 4.2 "*Comparison to human vs. climatic contribution as determined in previous studies*", it does not read well.

Page 19 line 8: "*respectivly*" → respectively

**References**

AghaKouchak, A., Norouzi, H., Madani, K., Mirchi, A., Azarderakhsh, M., Nazemi, A., et al. (2015). Aral Sea syndrome desiccates Lake Urmia: Call for action. *Journal of Great Lakes Research, 41*(1), 307-311. doi:http://dx.doi.org/10.1016/j.jglr.2014.12.007

Alborzi, A., Mirchi, A., Moftakhari, H., Mallakpour, I., Alian, S., Nazemi, A., et al. (2018). Climate-informed environmental inflows to revive a drying lake facing meteorological and anthropogenic droughts. *Environmental Research Letters, 13*(8), 084010.

Alizade Govarchin Ghale, Y., Altunkaynak, A., & Unal, A. (2018). Investigation Anthropogenic Impacts and Climate Factors on Drying up of Urmia Lake using Water Budget and Drought Analysis. *Water Resources Management, 32*(1), 325-337. doi:10.1007/s11269-017-1812-5

Alizade Govarchin Ghale, Y., Baykara, M., & Unal, A. (2019). Investigating the interaction between agricultural lands and Urmia Lake ecosystem using remote sensing techniques and hydro-climatic data analysis. *Agricultural Water Management, 221*, 566-579. doi:https://doi.org/10.1016/j.agwat.2019.05.028

Chaudhari, S., Felfelani, F., Shin, S., & Pokhrel, Y. (2018). Climate and anthropogenic contributions to the desiccation of the second largest saline lake in the twentieth century. *Journal of Hydrology, 560*, 342-353. doi:https://doi.org/10.1016/j.jhydrol.2018.03.034

Danesh-Yazdi, M., & Ataie-Ashtiani, B. (2019). Lake Urmia crisis and restoration plan: Planning without appropriate data and model is gambling. *Journal of Hydrology, 576*, 639-651. doi:https://doi.org/10.1016/j.jhydrol.2019.06.068

Fazel, N., Torabi Haghighi, A., & Kløve, B. (2017). Analysis of land use and climate change impacts by comparing river flow records for headwaters and lowland reaches. *Global and Planetary Change, 158*, 47-56. doi:https://doi.org/10.1016/j.gloplacha.2017.09.014

Khazaei, B., Khatami, S., Alemohammad, S. H., Rashidi, L., Wu, C., Madani, K., et al. (2019). Climatic or regionally induced by humans? Tracing hydro-climatic and land-use changes to better understand the Lake Urmia tragedy. *Journal of Hydrology, 569*, 203-217. doi:https://doi.org/10.1016/j.jhydrol.2018.12.004

۳.علیرغم افزایش تعداد چاههای حفرشده در سطح حوضه، میزان تخلیه و برداشت از آنها در سالیان اخیر با روند نزولی همراه بوده است. این امر بهخوبی نشاندهنده کاهش توان آبدهی آبخوانهای حوضه میباشد.

۴. کیفیت بسیاری از آبخوانها با روند نزولی همراه بوده است و این مسئله ناشی از برداشت بیرویه از منابع آب زیرزمینی حوضه میباشد.

۵.در شکل ۴، نحوه تأثیرگذاری برداشت بیرویه از چاهها بر روانابهای رودخانهها نشان داده شده است. درواقع این شکل بهخوبی نشاندهنده تغییرات نحوه اندرکنش آب زیرزمینی با آبراههها و رودخانهها میباشد. به بیان بهتر، به دلیل افزایش برداشت از منابع آب زیرزمینی یک رودخانه تغذیهشونده از منابع آب زیرزمینی تبدیل به رودخانه تغذیهکننده گردیده و حتی دبی پایه خود را نیز ممکن است از دست دهد. افت میزان آبدهی بسیاری از رودخانههای حوضه، بهخوبی مؤید این مطلب است.

**راهکارهای مصوب در خصوص مدیریت آب زیرزمینی**

بهمنظور ایجاد پایداری در وضعیت منابع آب زیرزمینی حوضه و همچنین کاهش پیامدهای منفی ناشی از برداشت بیرویه از منابع مذکور در تداوم افت تراز دریاچه ارومیه، راهکارهای مختلفی توسط دفتر برنامهریزی و تلفیق ستاد احیای دریاچه ارومیه بررسی شد و درنهایت راهکارهای ذیل در این خصوص مورد تصویب قرار گرفته است:

– ساماندهی چاههای حوضه آبریز دریاچه ارومیه و نصب کنتورهای هوشمند و حجمی جهت کنترل برداشت در راستای افزایش میزان جریان ورودی از رودخانهها به دریاچه ارومیه

– انجام هماهنگیهای لازم با قوه قضائیه در راستای تسهیل و تسریع در اجرای قانون تعیین تکلیف چاههای فاقد پروانه بهویژه چاههای اثرگذار بر آبهای سطحی

– شناسایی محدودههای اثرگذار بر آبدهی رودخانههای اصلی منتهی به دریاچه ارومیه و تقویت آنها از طریق عملیات آبخیزداری و آبخوانداری بهمنظور افزایش حجم آب ورودی به دریاچه.

در مجموع شواهد نشان میدهد که، یکی از عوامل مهم موثر بر کاهش آبدهی رودخانههای منتهی به دریاچه ارومیه، برداشت بیرویه از منابع آب زیرزمینی حوضه از طریق حفر تعداد قابل ملاحظهای چاه مجاز و غیر مجاز میباشد. لذا بر اساس مصوبات کارگروه ملی نجات دریاچه ارومیه ساماندهی وضعیت منابع آب زیرزمینی حوضه به ویژه بیش از ۴۰۰۰۰ حلقه چاه غیرمجاز و نظارت بر وضعیت برداشت از منابع راهبردی حوضه با همکاری همه نهادهای مسئول، امری بسیار ضروری میباشد.

دبیر خانه مرکزی: خیابان آزادی، جنب دانشگاه صنعتی شریف، پلاک ۵۱۷، طبقهٔ همکف، واحد۳.

دبیرخانه استانی:
• آذربایجان غربی، ارومیه خیابان عدالت، پلاک ۱۶، دفتر استانی ستاد احیای دریاچهٔ ارومیه.
• آذربایجان شرقی، تبریز، بلوار۲۲ بهمن، ابتدای شهرک زعفرانیه، ساختمان شماره دو استانداری، مدیریت بحران، دفتر استانی ستاد احیای دریاچهٔ ارومیه.

**آب زیرزمینی و احیای دریاچه ارومیه**

[Figure]

**وضعیت پراکندگی چاه های حوضه آبریز دریاچه ارومیه**

[Figure]

**شکل ۴- اندرکنش رودخانه با آب زیرزمینی**

**اهمیت آب زیرزمینی**

آب زیرزمینی از مهمترین مؤلفههای منابع آب تجدیدپذیر هر حوضه آبریز محسوب گردیده و بهعنوان ذخایر راهبردی آب شیرین از اهمیت منحصربهفردی در سطح دنیا بـــــرخوردار میباشد. همچنین این منابع نقش بسیاری در آبدهی چشمهها، قنوات و تغذیه و تأمین آبدهی پایه رودخانهها بر عهده دارد.گرچه منابع آب زیرزمینی درصد قابلملاحظهای از منابع آب شیرین دنیا را به خود اختصاص دادهاند، اما این منابع در مقابل برداشتهای بیرویه بسیار حساس و آسیبپذیر بوده و قدرت تجدیدپذیری خود را بهسرعت از دست میدهند. برخی از لایههای آبهای زیرزمینی بهخصوص لایههایی که با حفر چاههای عمیق از آنها برداشت میشود، قرنها و حتی هزاران سال طول میکشد که تجدید شوند و تنها لایههای سطحی منابع آب زیرزمینی از قابلیت تجدید سالانه برخوردارند.

[Figure]

شکل ۱ – مدت زمان مورد نیاز جهت تشکیل لایههای مختلف آب زیرزمینی

از لحاظ برداشت از منابع آب زیرزمینی در سطح دنیا، ایران با جمعیت بهمراتب کمتر پس از کشورهای هند، چین، آمریکا و پاکستان در مقام پنجم دنیا قرارگرفته است. افزایش تـــعداد چاههای غیرمجاز کشور به بیش از ۲۵۰ هزار حلقه در سالهای اخیر و برداشت بیش از ۱۱۰ میلیارد مترمکعبی (حدود ۳۶ درصد) از ذخـایر استاتیک آب زیرزمینی شیرین کشور، شاخصهای بسیار نگرانکنندهای هستند. اعلام ممنوعیت در ۳۱۷ دشت از ۶۰۹ دشت کشور، فرونشست و ایجاد فروچاله در بسیاری از دشتها تبعات بسیار ناگوار وضعیت نابسامان و
* * *
ناپایدار منابع آب زیرزمینی در کشور است. در حوضه آبریز دریاچه ارومیه، تعدد چاههای غیرمجاز حفرشده، افت کیفیت آبخوانها و کاهش رواناب رودخانهها و آبراهههای حوضه ازجمله مهمترین پیامدهای برداشت بیرویه از منابع آب زیرزمینی این حوضه میباشد.

**نقش آب زیرزمینی در وضعیت کنونی دریاچه ارومیه**

اگرچه بررسیهای صورت گرفته توسط دفتر برنامهریزی و تلفیق ستاد احیای دریاچه ارومیه نشاندهنده عدم ارتباط مؤثر و فعال بین دریاچه ارومیه و آبخوانهای ساحلی آن میباشد، اما حفر حدود ۸۸۰۰۰ حلقه چاه در سطح حوضه تــأثیر قابل ملاحظهای بر آبدهی رودخانههای حوضه داشته و این امر منجر به کاهش قابلملاحظه رواناب ورودی به دریاچه گردیده است. با فرض مساحت ۱۲۵۰۰ کیلومتر مربعی برای دشتهای و کوهپایههای حوضه، یک محاسبه ساده نشان میدهد که متأسفانه بهطور متوسط در هر کیلومتر مربع از سطح دشتهای حوضه تعداد ۷ حلقه چاه حفر گردیده است. البته با توجه به عدم توزیع یکنواخت چاهها در سطح حوضه، تراکم چاهها در برخی از دشتها بهمانند ارومـیه، میاندوآب و تبریز بیش از این تعداد میباشد. پراکنش چاههای حفرشده در سطح حوضه بهخوبی مؤید مطلب فوق است.

[Figure]

شکل ۲ – روند تغییرات مقدار برداشت از منابع آب زیرزمینی حوضه آبریز دریاچه در چهار دهه اخیر
* * *
نکته تأسفبرانگیز در خصوص چاههای حفرشده در سطح حوضه این است که بسیاری از آنها غیرمجاز بوده و بر طبق آمار موجود تعداد آنها به بیش از ۴۰۰۰۰ حلقه میرسد. در نمودار ارائهشده در شکل ۲ روند تغییرات میزان برداشت از منابع آب زیرزمینی حوضه و در شکل ۳ روند افزایشی میزان حفر چاه ها در سطح حوضه به تفکیک عمیق و نیمه عمیق در چهار دهه اخیر نشان داده شده است.

[Figure]

شکل ۳ – روند تغییرات تعداد چاههای عمیق و نیمه عمیق حوضه آبریز دریاچه

با توجه به ارقام ارائهشده در نمودارها ذکر چند نکته زیر، نشان دهنده وضعیت منابع آب زیرزمینی در سطح حوضه میتواند باشد:

۱.تعداد کل چاههای عمیق و نیمهعمیق حفاری شده در سطح حوضه در طی چهار دهه منتهی به سال آبی ۹۲-۹۱، ۳۱ برابر و نسبت به سال آبی ۶۴-۶۳، ۴ برابر شده است.

۲.تعداد قابل ملاحظهای از چاههای حفرشده در سطح حوضه بهصورت چاههای نیمهعمیق میباشد که اکثراً در حریم رودخانهها و آبراهههای حوضه واقعشدهاند. بر طبق آمار اعلام شده تعداد چاههای نیمهعمیق در سطح حوضه در سال آبی ۹۲-۹۱ نسبت به سال آبی ۶۴-۶۳ بیش از ۴ برابر و میزان تخلیه از آنها در حدود ۶ برابر افزایش داشته است. لازم به ذکر است که بخش عمدهای از این چاهها بهصورت غیرمجاز و توسط صاحبان باغات، مزارع و ویلاها حفر شده و مورد بهرهبرداری قرار گرفته است.

---

## Author Response (AR2)

**Dear Editor and Reviewers,**

*We have revised the manuscript according to the insightful comments provided by the editor and reviewers. All recommendations have been addressed in the revised manuscript. We would like to thank you for the thorough consideration and critical comments that helped us improve our manuscript.*

5 | Editor Comments

**Editor Comment:** You received two new reviews, one minor revisions but with some critical points and one rejected (with extensive review). Both reviewers agree the manuscript is really too long, the groundwater aspect of Lake Urmia (lake-groundwater relationship) should be discussed much more in depth and the model set-up, assumptions and uncertainty need more attention. This could mean the

10 manuscripts will be even longer. See some suggestions to shorten it below.

**Response**: *We would like to thank you for your time and suggestions. We have reduced the manuscript size and provided some description of the lake-groundwater relationship, model setup, and uncertainty. We hope the changes have made the manuscript suitable for publication and we look forward to your response.*

15 **Editor Comment:** A formal uncertainty analysis is not required for me (as requested by reviewer 2), but some more quantification and discussion as requested by the second reviewer needs to be included.

**Response**: *We have changed the model set up to consider uncertainty. We have discussed the uncertainty based on different optimal parameter sets which were obtained from the optimization algorithms (GA and NSGA-II).*

**Editor Comment:** Having gone through the paper I do want to stress I think editorial or only textual changes are not sufficient, I recommend to have a fresh look at the current manuscript and bring in some more focus (choice of results to present and discussion and decide which less important results should be presented in supplement only).

25 **Response**: *We have restructured the results and discussion sections and rewriting other parts accordingly. We have put some parts of the result section related to modifying NA in the supplement.*

*Also, we have put some parts of the material and methods in the supplement. In the revised version, we have focused more on uncertainty and model setup.*

**Editor Comment:** Could you reduce the introduction, and see whether the method section really needs all info? I could imagine that the well-known WaterGAP model(s) could be summarized in terms of Urmia application only (more, extensive info in supplement). For me, the main aspect (novelty) of your article seems using the comparison of calibrating using RS without and with ground information, so keep that in main text but bring all short references to your input data to the supplement as well.

**Response**: *We have restructured the manuscript based on your suggestion and put all the detailed information in the supplement. Also, in the revised version, an automatic approach was developed for calibrating WGHM based on evolutionary optimization algorithms. To reduce the length of the manuscript, we have focused on two calibration variants i.e. calibrating using RS with and without ground information as you mentioned in the next comment. We have summarized the descriptions of the WaterGAP model in the manuscript instead we have added some descriptions for the model in the supplement.*

**Editor Comment:** The results section is very long whereas the discussion is shorter. Typically, one would like to see the reverse. Instead of giving all correction parameters and time-series, you could try to highlight 1-2 results (and present rest in supplement). Furthermore, quite some paragraphs in the results section have discussions. Please re-evaluate whether these parts could not be combined in the discussion.

**Response**: *We agree that the results section is longer than normal size. We have reduced the size of the results and put some parts in the supplement. Also, we reconsidered the result and discussion sections based on your comment (see the result and discussion sections in the revised version).*

**Editor Comment:** Note, section 3 consists only of 1 subsection (3.1).

**Response**: *In the revised version it has two subsections.*
* * *
A point-by-point response to the Referee #2
* * *
**Referee #2:** The revised version of the article is much more understandable and better. However, I still think it is too long and there are a lot of assumptions made in the text which can be criticized separately.

But, I think a minor revision would be proper for the final decision considering an additional comment below.

**Response:** *First of all, we genuinely appreciate your time in reviewing the revised version. We have reduced the manuscript size and revised it according to your comments. We hope that you are satisfied, after the changes have been made.*

**Referee #2:** I think that you are aware that the role of groundwater on the lake has been discussed by many researchers of this field. Some of these researches e.g. Amiri et al. (2016) think that there is no significant relationship between lake and groundwater in the basin but some like Ashraf et al. (2017), and Vaheddoost and Aksoy (2018) believe that there is strong evidence of a relationship between them. Since your modeling includes analysis of groundwater I think you should compare your results by these articles and try to confirm or reject obtained results by them using similarities and dissimilarities in results.

**Response:** *We run the models under two scenarios (1) there is a relationship between lake and groundwater and (2) there is no relationship between lake and groundwater. In scenario 1, the seasonality of the groundwater storage was strongly misrepresented. Therefore, we could not calibrate the model when there is a relationship between lake and groundwater. However, WGHM as a hydrological model that does not include a gradient-based groundwater model has some limitations for studying groundwater-lake water flows. We believe that there is an indirect relationship between lake and groundwater i.e. groundwater-river, river-lake as accepted by ULRP (2015). In addition, as you mentioned some studies e.g. Amiri et al. (2016) using isotopic analyses and chemical tracer rejected the significant relationship between lake and groundwater. Also, Danesh-Yazdi and Ataie-Ashtiani (2019) stated that the study by Vaheddoost and Aksoy (2018) is not reliable and there is some doubt in accepting that. Therefore, we have added the following part in the discussion of the revised version.*

*"It is worth mentioning that WGHM as a hydrological model that does not include a gradient-based groundwater model has some limitations for studying groundwater-lake water flows. We attempted to calibrate WGHM under the assumption that there are direct water flows between lake and groundwater. Under this assumption, the seasonality of the groundwater storage was strongly misrepresented. Therefore, as accepted by ULRP (2015c), we assumed there is no direct flow between the lake and groundwater. While Vaheddoost and Aksoy (2018) using traditional hydrograph separation methods*

*claimed that there is a significant relationship between the lake and groundwater, Danesh-Yazdi and Ataie-Ashtiani (2019) rejected their claim. Equally, some studies that applied isotope and chemical tracer analyses (e.g. Amiri et al. 2016) rejected any significant relationship between lake and groundwater. In conclusion, the results of this study support the idea that there are no significant direct interactions*
5 *between lake and groundwater within the basin."*

*References:*

*Danesh-Yazdi, M., and Ataie-Ashtiani, B. (2019). Lake Urmia Crisis and Restoration Plan: Planning without Appropriate Data and Model Is Gambling. Journal of Hydrology, 576, 639-651.*
10
*Amiri, V., Nakhaei, M., Lak, R., and Kholghi, M. (2016). Geophysical, isotopic, and hydrogeochemical tools to identify potential impacts on coastal groundwater resources from Urmia hypersaline Lake, NW Iran. Environmental Science and Pollution Research, 23(16), 16738-16760.*

15 *Vaheddoost, B., and Aksoy, H. (2018). Interaction of groundwater with Lake Urmia in Iran. Hydrological processes, 32(21), 3283-3295.*

*ULRP (2015), Urmia Lake - Causes of shrinkage and potential threats. 36 pp (In Persian).*

A point-by-point response to the Referee #3:

**Referee #3: Overall assessment:** The aim of the study essential is to quantify the impact of human
20 activities (mostly in terms of water consumption) vs climatic changes on the Lake Urmia water balance. Even though 3 of the referees provided detailed reviews and pointed out to several shortcomings of the paper mostly on model setup and uncertainty, the authors' revision is minimal and in fact insufficient as many comments are effectively ignored.

**Response:** *Firstly, we are thankful for your time in reviewing our manuscript. We do not agree with you*
25 *about the "many comments are effectively ignored" because referee#2 was satisfied with the revised version. In the new revised version, we have addressed the shortcomings of the model setup and uncertainty.*

**Referee #3:** All reviewers except Chaudhari took issue with the experiment design, particularly the model set up, input data time period, and lack of adequate model calibration and evaluation. Yet, authors have

not changed the experiment design, and only added two performance metrics. No uncertainty or sensitivity analysis was conducted whatsoever, which is a common analysis required for any hydrological modeling study. This is even more serious as the manuscript is in fact inconsistent on the issue of uncertainty. While authors discussed the limitations of the work such as parameter uncertainty, no account of the hydrogeology of the lake, assuming constant bathymetry, among others; not only they have not accounted for these uncertainty sources by even a simple uncertainty/sensitivity analysis, they kept pushing that their study is "a holistic and reliable modelling approach", "we are confident that human water use reduced lake inflow that would have occurred without human water use during 2003-2013 by about 41%", and "This study proved that even without human water use Lake Urmia would not have recovered from the significant loss of lake water volume caused by the drought year 2008", among other instances of false overpromises.

**Response:** *We have developed a new model set up based on an auto-calibrated approach using a genetic algorithm (GA) and Non-dominated sorting genetic algorithm II (NSGA-II). We also have discussed uncertainty arising from the different possible optimal parameters that were obtained from the optimization algorithms. About the limitations, we should mention as you know all modeling studies faced some limitations that are inevitable. About the "holistic modeling", if you review the modeling study on Lake Urmia, all modeling applied a trial and error method based on a single objective calibration. Therefore, when we consider all possible data for our model, we consider it as a holistic one that includes TWSA, inflow, groundwater, lake volume. About the "reliable modeling" we agree with you. We have removed it from the manuscript. We also replaced the "we are confident ..." and "This study proved that" with "we found ..." and "Based on the results can be claimed that…", respectively.*

**Referee #3:** The manuscript is filled with redundant discussions (either well established in the literature or not relevant to the core research question of the paper), and is written in poor language with several typos. It is a waste of the editor's time and reviewers to resubmit a manuscript that has not been proof-read especially that most reviewers pointed this out. Further, some of the sources are either not peer-reviewed or in Persian. While local knowledge can be useful, the credibility of a non-peer reviewed source is always questionable. In such cases, authors should provide reason and demonstrate clearly instead of just referring to the source.

**Response:** *We have reformatted the manuscript, results and discussions have made shorter. We have checked the writing of the manuscript. We believe that readability is sufficient now. After acceptance HESS will proof again the manuscript by native speakers. Regarding the Persian references, we agree with you but unfortunately there are no English references, however, all ULRP reports have been reviewed by the scientific committee. Besides, actually we have no other way, for example about the "water withdrawals data" how we can cite it? Or about "irrigated area"? We should either use Persian references or do not provide any sources for these cases. Moreover, referring to ULRP as the main data center for lake Urmia is common in all published papers in international journals.*

**Referee #3:** In my evaluation, the manuscript is rejected. If authors wish to resubmit the work, they must revise the modeling experiment to sufficiently address the comments by reviewers, and by addressing I specifically mean to change their modeling setup by providing a more robust calibration than a simple and insufficient trial and error, transparently explaining the modeling setup to ensure (somewhat) the reproducibility of the study, and perform some sensitivity or uncertainty analysis. Also, remove all the redundant discussions from the manuscript and focus their discussion on the relevance of the results to the lake given the uncertainties. While most of the comments by reviewers still hold, here I high-lights a few urgent ones.

**Response**: *As aforementioned, we have revised the model setup and added uncertainty analysis to the revised version. Readers might not just focus on one objective of the study (quantify the impact of human water use and climate variation on the Lake Urmia water balance). As the editor has stated, one novelty of this study is assessing the value of remote sensing data and ground information for calibrating a hydrological model. Therefore, there are not such redundant discussions as you stated. However, we have reduced discussions with more direct focus on two objectives of the study.*

**Referee #3: Problem description:** The problem description (i.e. drivers of the lake desiccation) particularly in the introduction as well as the later discussion of the results is problematic. It is misleading and inaccurate to stack the climatic and human drivers together (e.g. page 29 lines 1- 10). Khazaei et al. [2019] disentangled these two: they compared the influence of these two sets of drivers for the lake drying and demonstrated the regional human activities (including water management, but not limited to) are the

primary drivers compared to climatic changes (including atmospheric droughts). AghaKouchak et al. [2015] also argued that droughts cannot be the primary driver. These two studies, among others, are based on directly analysing the data themselves, without relying on a model of the system which in most cases are inadequate. While these studies have their own shortcomings, as any scientific study has, your modeling results are inadequate to challenge them. As pointed out by the reviewers your modeling setup has several issues. Inadequate models, regardless of the extent of their inputs and their results, are inadequate. Authors said: "This study proved that even without human water use Lake Urmia would not have recovered from the significant loss of lake water volume caused by the drought year 2008". This conclusion is in direct contradiction with AghaKouchak et al. [2015] conclusion that "a satellite based gauge-adjusted climate record… of Lake Urmia basin's Standardized Precipitation Index… indicates no significant trend in droughts over the past three decades at the 0.05 significance (95% confidence) level... In fact, the region has experienced more severe drought events in the past (e.g., 1997– 2002) that did not lead to a substantial change in the lake's surface area. Thus, we caution against overrating the role of droughts in the disruption of the lake's water balance to the extent that would cause such a massive shrinkage". Given that AghaKouchak et al. [2015] directly analysed the historic data of the lake without relying on any inadequate model of the lake system, it is reasonable to say this contradiction indicates the shortcoming and (un)reliability of the model set-up in this study. Notwithstanding the unscientific language of this sentence. Science is not in the business of proving anything. In science we demonstrate and approximate. This is more so the case when it comes to hydrological modeling with numerous types and sources of uncertainty including both model structure and data. Also, what do you mean by "climate variations"? Are you referring to only natural climate variability or climatic changes which include both natural variability and human-induced changes?

**Response**: *We agree with you about the "page 29 lines 1-10". To clarify we have removed the word "main" in line 6, now the sentence would be accurate. About two mentioned studies, Khazaei et al. [2019] and AghaKouchak et al. [2015], we discussed in "Comparison to human vs. climatic contribution as determined in previous studies" section, why our results are different with some studies and in line with some other studies. Analysis of the monthly data used in calculating SPI cannot take into account the changes in the pattern of daily precipitation. To approve this claim, we refer to two studies that they also*

*used directly in-situ data without modeling. Bavil et al. (2018) showed a significant increase in the frequency of daily precipitation of less than 5 mm and a significant decrease in the frequency of daily precipitation of 10-15 mm, suggesting a runoff reduction even in case of constant annual precipitation. Also, Hosseini-Moghari et al. (2018) showed that an increasing frequency of days with less than 5 mm*

5 *precipitation in combination with decreasing monthly precipitation has led to reductions of inflow into two dams in the Lake Urmia basin that are located downstream of the areas with insignificant human water use. Therefore, we did not rely just on our results and provided some facts from pure in-situ data which has been rarely mentioned in the previous studies. Moreover, Shadkam et al. (2016) who considered only inflow into the lake, reported the same results for the impact of human water use on the*

10 *reduction of inflow into the lake. About "climate variations" we cannot claim that it is a natural climate variability or climatic change. To speak about climate change must be conscious. It might be considered as any change or variations in climate variables data in our study.*

**Referee #3: Model calibration and evaluation:** the model calibration is also questionable. It is only based on trial and error so the identified parameter sets are not reliable. Also, it is not clear how sensitive the model results are to these parameters. There is issue about over-parametrizing the model as in each variant new data is added (issue about correction factors pointed out by reviewers). A single model run for each variant is not sufficient, even if the calibration was done through an automatic parameter space search scheme. This is well-established in the literature and a model ensemble is required to account for the uncertainties, even though partly. If the model is run on a daily basis, authors should be trans-parent and present the daily results too.

Monthly and annual performance metrics are usually high for most models. The devil is in the details though. On table 4, the flow is only calibrated on an annual scale and not evaluated at all. There is significant seasonality in this region. It is quite possible that seasonal errors are just cancelling each other out and give seemingly good annual results. This can be seen on figure 8 where the model exhibits unrealistic seasonality which is not in the observed data, e.g. in Fig 8e, there is generally a negative bias in the first half of the RS_Q_GW_NA simulation, and positive bias in the second half (red line is first below the black line systematically, and then above it).

**Response**: *We have reconsidered the model calibration by using two optimization algorithms, namely genetic algorithm (GA) and Non-dominated sorting genetic algorithm II (NSGA-II). As editor request, to keep the length of the revised version no more than the previous version, we just focus on two calibration variants using RS with and without ground information. Other variants only considered for some*

5 *discussions. In the revised version, we used an ensemble of model outputs that were obtained from different GA runs or Pareto front for NSGA-II. Therefore, we have discussed the uncertainties in the revised version. Adding daily outputs making the manuscript longer than the current version and we believe that it is redundant. Because most of the outputs are anomaly based on the monthly mean, therefore daily data could not be beneficial. However, to consider your concern about the inflow into the*

10 *lake, we have plotted the daily inflow here. About Fig 8e, it should be noted that modeling lake volume is not a simple task for a hydrological model. Therefore, we believe that the performance of the model is quite acceptable, while in the revised version the performance of the model improved in this regard.*

[Figure]

**Figure R: Time series of simulated daily inflow into the lake**

**Referee #3: Uncertainties:** as Referee 1 pointed out in their comment 6 and elsewhere, the uncertainties are playing a crucial role here. While authors added a section of uncertainty discussion, it is an ad-hoc discussion. While the discussion obviously undermines the experiment design and the results, they have

not revised the experiment design to account for these uncertainties, and also they keep overpromising about the reliability of their results. Further, their discussion of uncertainty shows a lack of understanding about the area of model uncertainty. For instance, they said "Model parameter uncertainty was reduced by the comprehensive multi-observation calibration". Parameter uncertainty will not be reduced by just

5 adding more data to the model; data uncertainty matters, "garbage in, garbage out" [Kuczera et al., 2010]. The authors must demonstrate how adding input to the model reduced parameter uncertainty, while justifying the credibility of the data themselves. They have not done any uncertainty or sensitive analysis whatsoever.

**Response**: *We have revised the model set up and added some uncertainty analysis to the manuscript. We*

10 *believe the parameter uncertainty should be reduced when a multi calibration approach is used. Because the model should satisfy more than one objective therefore change in one parameter should be done with less freedom. We already showed this issue in Table 5 where the model provides changes in total water storage through only by changing lake depth. In addition, use more observed data always help us improve the calibration uncertainty that is arising from our lack of awareness of the observed values of a given*

15 *variable.*

Thank you very much again for your time and for providing valuable comments.

[revised manuscript text omitted]

---

## Author Response (AR3)

**Dear Editor and Reviewers,**

We have revised the manuscript according to the insightful comments provided by the editor and reviewers. All recommendations have been addressed in the revised manuscript. We would like to thank you for the thorough consideration and critical comments that helped us to improve our manuscript. In the following, we have provided a point-by-point response
5 (in blue) to the comments.
* * *
Editor Comments
* * *
**Editor Comment:** Both referees agree the paper is improved a lot and methods - concepts are well explained. However, both referees are also critical on the interpretation of your own results and especially on the discussions currently in literature on the Urmia lake drying causes. I do agree with the referees that interpretations and discussion should be extensive and complete
10 and take into account various views.

Please respond in detail to the referees' reports and submit a revised version of the paper including a track changes version. I will review the paper and your responses before considering publication.

**Response:** Thank you very much for giving us this opportunity to revise the manuscript. We have revised the manuscript based on your and referee's comments and added a more extensive discussion. We hope the changes have made the manuscript
15 suitable for publication, and we look forward to your response.
* * *
A point-by-point response to the Referee #2
* * *
**Referee #2:** As I asked in the previous version, several revisions are made. I think the revised article is much better and a lot of time and effort are given to the study. As I underlined before, the drawn conclusions are over-exaggerated. For instance, the
20 reality that your model can not be calibrated using groundwater data can never be interpreted as there is no interaction between groundwater and the lake. In addition, in P 18, Line -5-10, it is better to be aware that, no modeling was done by ULRP (2015c) and, Danesh-Yazdi and Ataie-Ashtiani (2019) but they emphasized that "it is not likely" to have a relation between. Also based on the studies of Ashraf et al. (2017), and Rodell et al. (2018) there is an undeniable relationship between groundwater and lake water in Lake Urmia basin. So I recommend adding this publication as well to your discussion as the cases which
25 concluded that there is a relationship between groundwater and lake.

**Response:** Firstly, thank you very much for your time to review our manuscript. We are not sure whether we understood your point about over-exaggerated correctly. However, it seems that it related to the contribution of climatic variations on lake shrinkage. It should be noted that all studies that conducted hydrological modelling over lake Urmia support our results (except
30 Chaudhari et al. (2019) who used uncalibrated model for this purpose). Shadkam et al. (2016), who calibrated the VIC model for Urmia Basin with a longer period (1960-2010), stated: "Our results show that annual inflow to Urmia Lake has dropped by 48% over the study period. About three-fifths of this change was caused by climate change, and about two-fifths was caused

by water resources development." Also, Farokhnia et al. (2018) who developed and calibrated SWAT model for separating anthropogenic factor and climate impact on Lake Urmia during 22 years period ending in 2009 stated: "the cumulative effect of climatic and human-induced changes in reducing the inflow into Urmia Lake was almost equal, but due to the intensification of climate variability in the second half of this period, the effect of climatic factors was dominant in the rapid negative trend of lake water level." These statements show that there are some other research that emphasize the contribution of the climatic factors. Therefore, we believe that although all models suffer from uncertainties, these uncertainties cannot change the overall results.

About the interaction between groundwater and the lake, you are right that there is no modelling in ULRP (2015c), but field tests by the Japan International Cooperation Agency (JICA) Japanese company confirmed their assumption. We could not find any statement in Rodell et al. (2018) about the relationship between groundwater and lake across the Lake Urmia basin. Of course, there are some sentences about groundwater and lake in Urumqi (in China), not Urmia!

We have revised the part related to the relationship between groundwater and lake by considering your comment as follows:

"It is worth mentioning that WGHM as a hydrological model that does not include a gradient-based groundwater model has some limitations for studying groundwater-lake water flows. We attempted to calibrate WGHM under the assumption that there are direct water flows between lake and groundwater. Under this assumption, the seasonality of the groundwater storage was strongly misrepresented. Therefore, as accepted by ULRP (2015c), we assumed there is no direct flow between the lake and groundwater. This is consistent with Danesh-Yazdi and Ataie-Ashtiani (2019) who stated that a significant water exchange between the lake and groundwater is unlikely. Also, Amiri et al. (2016) based on isotope and chemical tracer analyses rejected any significant relationship between lake. However, some studies, e.g. Ashraf et al. (2017) and Vaheddoost and Aksoy (2018), stated the opposite. In conclusion, the results of this study support the idea that there are no significant direct interactions between lake and groundwater in the Lake Urmia basin."

**Referee #2:** It is better to use "water level" instead of "Water table" for the case of lake water level.

**Response:** The "lake water table" was replaced with the "lake water level" in the whole manuscript.

**Referee #2:** The statistics obtained for your models are not satisfactory especially when Genetic Programming or optimization is addressed.

**Response**: We agree with you if, for example, we have done a single calibration against Q when all observed data be used (i.e., variant RS_Q_QW_NA), the KGE would be larger than the current KGE. However, for ensemble outputs from a multi-objective calibration the KGE value of 0.86, 0.82, 0.85, and 0.89 for simulating TWSA, Q, GWSA, and LV, respectively, are quite good as Pechlivanidis et al. (2014) stated that a KGE value of greater than 0.75 is acceptable for the calibration of a hydrological model.

**Referee #3:** First and foremost, I would like to sincerely apologize for my late reviews – due to family reasons. I have now reviewed the new manuscripts and authors' responses to previous reviews. By and large, the authors have addressed most of the comments, updated the modelling, condensed the manuscript and improved the use of language, and improved the

5   discussion of results (particularly adding a new section on the limitations of the study). So, I would like to thank authors for their efforts in this round of the review.

**Response**: Thank you very much for your time to review the revised version of our manuscript. It is our pleasure that the revised version satisfied you. We have considered all your points in the new revised version. Below, we have provided a response to your comments in detail.

**Referee #3:** I would like to also thank authors for dedicating a section on limitations (section 4.3). Not only it is good practice for scientific studies, but also it directs readers to design future studies to address past limitations. That said, a few major points are missing in the discussion of limitations:

• The study period is limited and do not include some of the most important years of the lake (and the basin) old/recent history

15   and evolution, e.g. recent changes in the lake since 2013 (beyond the study period of this manuscript). Compared to most hydro-climatology studies of the lake, this study is based on a very short period (2003-2013), and hence generalizing its results beyond this domain is difficult, particularly due to the non-stationarity of the lake system. I expanded on this in section 3 of this review.

• Role of dam constructions, groundwater withdrawals and its hydrological connectivity to the lake, and seasonal variations of

20   the flow overlooked in the model calibration.

**Response**: We respond to this comment in later parts where you describe each limitation separately.

**Referee #3:** Moreover, section 4.2 "Comparison to human vs. climatic contribution as determined in previous studies" comes short of providing an adequate and accurate characterization of the ongoing debate

25   within the literature:

• I acknowledge that the Lake Urmia desiccation has been an ongoing contested debate, i.e. whether the main driver of drying is management-related and human activities or climatic. This very question is indeed the crux of the matter, and hence it is in the best interest of both authors and readers to be more rigorous on this discussion. As opposed to several previous studies, this study puts more weight on the climatic drivers of the lake drying. However, the authors (in section 4.2) misrepresented or

30   overlooked some of those studies which argued for human activities over climatic drivers. Regardless of my personal position in this debate, authors are unduly framing the results and merits of the previous studies to justify their own side of the argument. Further, they have failed to discuss a few important studies on the lake. I demonstrate this in section 2 of this review. To help

the authors, I discussed several points in details. I have also suggested few additional references and edits throughout. Therefore, in my opinion section 4.2 of the manuscript is inadequate and must be improved.

**Response**: Thank you very much for your detail points. We have expanded the discussion to apply your comments. The changes that have been made based on your points are presented in your following comments separately.

**Referee #3:** Section 4.2 "human vs. climatic contribution"

The discussion is not elaborative, lacks a clear discussion line, and previous studies are not appropriately represented/discussed. Here I discuss a few points in this section as an example, to help authors better discuss this important point.

10  **Response**: Thank you very much for the help to improve the discussion part, we have revised the manuscript accordingly.

**Referee #3:** Page 18 line 17-21: "Chaudhari et al. (2018) concluded that human-induced changes accounted for 86% of the lake volume decline during 1995-2010, while we determined values of 39-43% for 2003- 2013. According to our study, human water use was the reason for 39-45% inflow reduction into the lake during 2003-2013 which is very similar to the values of

15  Shadkam et al. (2016) for the years 2003-2009 (comp. their Fig. 8). Discrepancies are likely due to different analysis methods but different analysis periods and conceptualizations make a direct comparison of the estimated contributions difficult." Re the underlined part: it is a general statement and not good enough to simply overlook the details leading to these differences. It is essential to discuss in more detail what are the main differences between these studies e.g. in terms of data type, analysis approaches, fundamental assumptions, etc. For instance, Chaudhari et al. (2018) studied a considerably longer period. They

20  also studied the land use changes in detail: over 1987-2016 showed ~98% and ~180% increase in agricultural lands and urban areas, respectively. They accounted for human impact during 1995-2010 (based on simulation of streamflow into the lake). Various studies identified two distinct periods of pre- and post-change in the lake dynamics, e.g. Khazaei et al. (2019) identified year 2000 as the change point and Fazel et al. (2017) identified year 2001. Given that, studies such as Chaudhari et al. (2018) take into account a wider range of the non-stationarity of the lake than the present study where only a part of the post-change

25  period is investigated. It is plausible to expect that if your model was successfully calibrated over a longer period including years prior to 2000, it would have lead to different results.

**Response**: Firstly, it should be noted that some other studies like Shadkam et al. (2016) who calibrated the VIC model for Urmia Basin with a longer period (1960-2010) stated: "Our results show that annual inflow to Urmia Lake has dropped by 48% over the study period. About three-fifths of this change was caused by climate change, and about two-fifths was caused

30  by water resources development.". Also, Farokhnia et al. (2018) who developed and calibrated SWAT model for separating anthropogenic factor and climate impact on Lake Urmia during 22 years period ending in 2009 stated: "the cumulative effect of climatic and human-induced changes in reducing the inflow into Urmia Lake was almost equal, but due to the intensification of climate variability in the second half of this period, the effect of climatic factors was dominant in the rapid negative trend

of lake water level.". These studies that include the change point year (2000 or 2001) support our results. Therefore, we believe our results are valid for the studied period without considering the change point year(s).

As we mentioned in the manuscript, Chaudhari et al. (2018) used an uncalibrated global hydrological model. Although, the model performance was assessed against the inflow into the lake (inflow data from Hassanzade et al. (2012)). The used data for assessing the model is quite far from observations. For instance, simulated annual inflow into the lake was estimated to be $3,700\cdot10^6$ m$^3$ in 2003 (their Fig. 8), while observed inflow was much higher, $5,835\cdot10^6$ m$^3$. In 2009, observed inflow, with $1,036\cdot10^6$ m$^3$, was only half of the simulated one. Therefore, the source of the main differences is using the uncalibrated model in Chaudhari et al. (2018). We believe that considering a longer period and land-use change based on an uncalibrated model can not challenge our results while Farokhnia et al. (2018), who also considered the land-use change support our findings. We modified this part as follows:

"Discrepancies are likely due to different analysis methods, but different analysis periods and conceptualizations make a direct comparison of the estimated contributions difficult. Chaudhari et al. (2018) performed a comprehensive hydrological modelling of Lake Urmia basin. They also studied the land use changes in detail over 1987-2016 and determined a ~98% and ~180% increase in agricultural lands and urban areas, respectively. However, their uncalibrated global hydrological model that represented the basin by 5-6 cells only was not able to simulate well the flows and storages in the basin. For example, simulated annual inflow into the lake was estimated to be $3,700\cdot10^6$ m$^3$ in 2003 (their Fig. 8) while observed inflow was much higher, $5,835\cdot10^6$ m$^3$. In 2009, observed inflow, with $1,036\cdot10^6$ m$^3$, was only half of the simulated one. Therefore, the very high human contribution to the lake volume decline of 86% determined by Chaudhari et al. (2018) may arise from the poor performance of the uncalibrated model. In addition, Chaudhari et al. (2018) studied a considerably longer period, i.e., 1995-2010, that includes the change point of lake dynamic (the year 2000 based on Khazaei et al. (2019) and 2001 based on Fazel et al. (2017)). Although including years prior to 2000 might be lead to different results, some other studies like Shadkam et al. (2016) and Farokhnia et al. (2018), who their modelling included years 2000 and 2001, support the results of the current study. Shadkam et al. (2016) stated that climate change was responsible for three-fifths of inflow reduction into the lake, and the rest was caused by water resources development between 1995-2010. Also, Farokhnia et al. (2018) showed that during a 22 years period ending in 2009, the effect of anthropogenic and climatic factors in reducing the inflow into Lake Urmia was almost equal."

**Referee #3:** "While Ghale et al. (2018) seem to support the results of Chaudhari et al. (2018) as they state that 80% of drying of Lake Urmia is due to anthropogenic impacts during 1998-2010, their statistical analysis assumes that lake inflow from rivers can be considered to reflect "anthropogenic impacts" while precipitation and evaporation reflect climatic variation. However, inflow is in reality also affected by climatic variations." Your argument here is incomplete, as the impact of climate vs. human activities on river networks is different for headwaters and lower river reaches. Fazel et al. (2017) investigated this in detail, analyzing the flow regime changes across the lake basin (57 flow gauging stations) over the period 1949-2013 (perhaps the longest record of the basin flow studied so far). Their study showed that while "flow regime in river headwaters appeared to

be dominated by natural forces", "the reduction in river flow magnitude increased from headwaters to downstream reaches for all rivers" due to dam river regulations and dam constructions. They further argued that "Changes in river flow in the period 1965–2013 cannot be explained by climate change, the effects of which occur much more slowly than those of land use change in the region". They concluded that "The results showed that irrigation was by far the main driving force for river flow regime changes in the lake basin. All stations close to the lake and on adjacent plains showed significantly higher impacts of land use change than headwaters. As headwaters are relatively unaffected by agriculture, the non-significant changes observed in headwater flow regimes indicate a minor effect of climate change on river flows in the region".

**Response**: Fistly, please note that our paper does not make any claims about the impact of climate change, but abouth the impacts of climatic variations. Our results do not deny the impact of human water use. We considered human water use as a significant factor, while the climatic factors are also significant. Therefore, although our result is not entirely consistent with Fazel et al. (2017), it is not in conflict with it either. Contrary to your statement, the study period of Fazel et al. (2017) is filled with missing data (please see Figure 7 in their study), and the period with completed data is considerably shorter than 1949-2013. They considered the post-change period, i.e., 2001-2013, compared with 1965–1977 as the pre-change period for downstream stations, while for headwater stations, the pre-change period was 1989–2001 or 1965–1977 due to availability of data (please see Figures 4 and 5 in their study). Therefore their analysis suffers from some errors caused by the different comparison periods. However, among the 6 stations that they plotted in Figure 5 (for headwater), three of them also have a reduction in inflow. We believe that comparing a period of 12 years under this condition cannot be a useful approach for quantifying the effect of climate change over the basin. However, more flow reduction in downstream stations due to agricultural development in upstream expected in any basin. Our study also confirmed that human activities have significantly reduced the inflow in the downstream of the basin, i.e., inflow into the lake. We modified this part as follows:

"While Ghale et al. (2018) seem to support the results of Chaudhari et al. (2018) as they state that 80% of drying of Lake Urmia is due to anthropogenic impacts during 1998-2010, their statistical analysis assumes that lake inflow from rivers can be considered to reflect "anthropogenic impacts" while precipitation and evaporation reflect climatic variation. However, although inflow into the lake is surely affected by human water use in upstream, also affected by climate variations over the basin."

**Referee #3:** "Using a statistical change point analysis and without modelling, Khazaei et al. (2019) stated that given the stable conditions of precipitation and temperature, climatic changes could not explain the dramatic decline of the lake level; however, they did not use in-situ data (except lake water level data) for their analysis" Study by Khazaei et al. (2019) is more than a simple statistical change point analysis, they estimated the land use change (particularly vegetation dynamics and its associated hydrological loss in terms of evapotranspiration) and trends of various hydro-climatic variables across various time scales. While their study surely has its own limitations, lack of modeling and use of insitu data are not the major limitations – let alone this is too generic for a scientific criticism. One of the major limitations of their work, for instance, is that they did not account for the role of groundwater dynamics in their analysis.

**Response**: We agree with you that the use of global datasets is not a limitation for a study. However, using them for a small region like the Urmia basin to investigate the effect of climate variability is always questionable when their validity is not assessed over the region. Therefore, we believe that it can be a limitation and may lead to misleading results if the accuracy of the data is not reasonable. They used monthly GPCP precipitation data for assessing the trend of precipitation over the basin. However, the proportion of shared variance between GPCP and in-situ data over the basin is about 0.75 on a monthly scale (see Table 2 in Jalili et al. 2012). Therefore, their analysis includes some errors caused by the questionable quality of data. Additionally, their analyses were done on a monthly scale that neglected changes in the daily precipitation pattern. We modified this part as follows:

"Using a statistical change point analysis and without modelling, Khazaei et al. (2019) stated that given the stable conditions of precipitation and temperature, climatic variations could not explain the dramatic decline of the lake level. They also estimated the change of vegetation dynamics and its associated hydrological loss in terms of evapotranspiration. They used monthly GPCP precipitation data for assessing the trend of precipitation over the basin. However, the proportion of shared variance between GPCP and in-situ data over the basin is about 0.75 on a monthly scale (see Table 2 in Jalili et al. 2012). Therefore, their analysis suffers from the poor quality of precipitation data. Moreover, their analysis was done on a monthly scale that cannot capture the sub-monthly variability of climatic variables. Also, they did not account for the role of groundwater dynamics in their analysis."

**Referee #3:** "For quantifying human and climatic contributions to observed hydrological changes, a comprehensive modeling approach that takes into account, for example, the impacts of changing temperatures on runoff and thus river inflow and on evapotranspiration of the lake itself is preferable." Preferable to what exactly? I tend to disagree that modeling is preferable to comprehensive analysis of historical data. Modeling introduce various sources of new uncertainty to a problem (such as model structural uncertainty, parameter uncertainty, over-parameterization, parameter transferability across time and space, etc.), which are not preferable to the simplifying assumptions underlying statistical analyses (such as trend, correlation, or linear regressions). In general, I believe, both approaches of modeling and data analysis can inform us in some ways, while each has its own shortcomings in other ways.

**Response**: We generally agree with you, but statistical analyses such as trend and correlation sometimes cannot give us useful information about what would happen. For example, when there is no trend in precipitation, but having a significant trend in river flow, we conclude that human activities are the leading factor for river flow reduction (most of the trend studies over lake Urmia suffer from this issue). While, in hydrological modelling the depth of precipitation in each event, the interval between rainfall events (represented in soil moisture) and other involved elements to generate runoff would be considered. Meanwhile, we agree with you that the models have their own disadvantages, i.e., more uncertainty. In our study discussed that the main point of disagreement among researchers on the contribution of each factor to the lake shrinkage comes from this issue. All modelling studies (except Chaudhari et al. (2018) that used uncalibrated model), i.e., shadkam et al. (2016), Farokhnia et al. (2018), and also our study highlighted the impact of climate factor could not be ignored over the basin. However, trend and

correlation analysis studies like Khazaei et al. (2019) and Ghale et al. (2018) stated the climate contribution is negligible compared to anthropogenic impacts. Trend analysis of daily precipitation for different classes, i.e., tiny, moderate, and heavy like Bavil et al. (2018) can be helpful if we would like to use a trend analysis approach. Bavil et al., (2018) showed that there is a significant increase in the frequency of daily precipitation of less than 5 mm and a significant decrease in the frequency of daily precipitation of 10-15 mm, suggesting a runoff reduction even in case of constant annual precipitation. We modified this part as follows:

"For quantifying human and climatic contributions to observed hydrological changes, a comprehensive modelling approach that takes into account, for example, the impacts of changing temperatures and land use change (e.g., urbanization and cropland expansion) on runoff generation and thus river inflow and on evaporation of the lake itself is preferable to statistical analyses such as trend and correlation analysis. Such statistical analyses may be misleading about reasons for certain temporal changes. For example, when there is no trend in precipitation but a significant trend in streamflow, it may be concluded that human activities are the dominant case of streamflow reduction; most of the trend studies for Lake Urmia suffer from such a hasty conclusion. In hydrological modelling, more detailed information such as the depth of precipitation in each event, the interval between rainfall events (represented in soil moisture) and other involved elements to generate runoff are considered. All modelling studies (except Chaudhari et al. (2018) who used an uncalibrated model), i.e., Shadkam et al. (2016), Farokhnia et al. (2018) and our study, found that the impact of climatic variations could not be ignored over the basin, while, trend and correlation analysis studies such as Khazaei et al. (2019) and Ghale et al. (2018) stated the climate contribution is negligible compared to anthropogenic impacts. We suggest to do trend analysis of daily precipitation distinguishing different intensity classes (e.g. Bavil et al. 2018)."

**Referee #3:** "Chaudhari et al. (2018) but their uncalibrated global hydrological model that represented the basin by 5-6 cells only was not able to simulate well the flows and storages in the basin." This is a mischaracterization of Chaudhari et al. (2018), undermining their extensive modeling setup and evaluation. Although Chaudhari did not explicitly discuss the model setup and calibration, they demonstrated the adequacy of their model by evaluating various model outputs against available knowledge and data of the LU basin. For instance, they compared their simulation inflow to the lake with the observed inflow record (previously gathered by Hassanzadeh et al., 2012). As the figure shows it is in good agreement. I agree with the authors' intent to critically review previous studies to elaborate their shortcomings, however this must be done rigorously and accurately.

[Figure]

**Fig. 7.** Comparison of simulated river inflow to Urmia Lake from HiGW-MAT model and the inflow data from Hassanzadeh et al., (2012).

**Response**: We would not undermine their extensive modelling setup and evaluation. We reported only their shortcomings. From Chaudhari et al. (2018): "The simulations we use in this study are based on Pokhrel et al. (2015) and Felfelani et al. (2017) conducted at the global scale.". Therefore, no calibration was performed specifically for Lake Urmia. We could not find inflow data in Hassanzadeh et al. (2012). However, although their model can be simulated inflow into the lake relatively well, the problem is that this inflow data is not correct and even far from observations. As we mentioned in the manuscript, for example, simulated annual inflow into the lake was estimated to be $3,700 \cdot 10^6$ $m^3$ in 2003 (their Fig. 8) while observed inflow was much higher, $5,835 \cdot 10^6$ $m^3$. In 2009, observed inflow, with $1,036 \cdot 10^6$ $m^3$, was only half of the simulated one. We believe under this situation, their results are not reliable and at least are not comparable with other modelling studies over the basin; however, we remain at your disposal for any further information and modification.

**Referee #3:** Page 19 lines 2-5: "Hosseini-Moghari et al. (2018) showed that an increasing frequency of days with less than 5 mm precipitation in combination with decreasing monthly precipitation has led to the observed reduced inflow into two dams in the Lake Urmia basin that are located 5 downstream of areas with insignificant human water use." This study is not available online and it is not possible to confirm whether it is peer-reviewed or not.

**Response**: This paper was presented in the 8th Global FRIEND-Water Conference, Beijing, China, 6-9 November 2018. Please check the paper here http://m.iahr.org.cn/Resource/abstractinfo/3955.

**Referee #3:** "We conclude that analyses should be done on a daily time scale or smaller." What type of analyses exactly? Such a generic statement. Needless to mention that the very present study of the authors is not done on a daily time scale either. The time scale of a study depends on its objective.

**Response**: All types of analysis would be done for climate change or variability impact. Because climate change affects not only the climate variable amount but also their pattern. It should be noted that our modelling was done on a daily scale and presented on a monthly or annual scale. Daily assessment does not mean to provide results on a daily scale; it means to assess the effect of variation in the daily time scale on the results. We modified this part as follows:

"We conclude that for assessing the effect of climatic variability on hydroclimatic variables, the analyses should be done on a daily time scale or shorter to consider the change in amount and patterns of variables."

**Referee #3:** "we examined the ratio of annual inflow into the lake (based on the ensemble mean) over annual precipitation during the study period. This ratio reached maximum values in 2003 (0.29 and 0.41 for the anthropogenic and naturalized conditions, respectively) and minimum values in 2009 (0.07 and 0.15). Averaged over the period 2009-2013, these ratios are, with 0.11 (ANT) and 0.22 (NAT), much smaller than the values for 2003-2007, 0.20 and 0.32. Thus, the drought year 2008 as well as the relatively small ratio of inflow into the lake over precipitation in the last five years of the study period play a significant role in the decline of inflow and lake water storage" There are various issues with this argument. First, the period 2009-2013 is a very short period to build a hydro-climatic analysis on, particularly for LU with remarkable non-stationarity. So, the naturalized scenario based on this period is not reliable. Second, the considerable extraction of groundwater resources has been an additional source of water for irrigation and consequently hydrological loss in this basin. The impact of groundwater withdrawal (and its consequent hydrological loss) would have had a direct impact on the lake and possibly on streamflow generation in the basin as well (e.g. as the land coverage of the basin has changed). Urbanization in this basin (discussed by Chaudhari et al. (2018)) together with the expansion of agricultural and irrigated areas would have an impact on streamflow generation (both magnitude and generation mechanisms).

**Response**: Firstly, we did not do any specific hydro-climatic analysis. We mentioned the ratio of annual inflow into the lake over annual precipitation for the sub-periods. We did not state any general statement from a statistical point of view that requires much data; therefore, non-stationarity for sub-period of 5 years does not matter here. Regarding the effects of the extraction of groundwater, we agree with you. Although comprehensive modelling of the relationship between groundwater withdrawal and streamflow generation is not a simple simulation, the WGHM model can model this relationship with promising accuracy; hence there is no serious concern in this regard. Under the naturalized conditions, there is no groundwater withdrawal and land-use change, so under this condition, your mentioned point does not matter. Finally, our results support your statement, and the difference between the ratio in anthropogenic and naturalized conditions comes from the expansion of agricultural and irrigated areas.

**Referee #3:** "For quantifying human and climatic contributions to observed hydrological changes, a comprehensive modeling approach that takes into account, for example, the impacts of changing temperatures on runoff and thus river inflow and on evapotranspiration of the lake itself is preferable." Also, estimating the impact of land use change (e.g. urbanization and cropland expansion) on runoff generation in the basin.

**Response**: We have revised it as follows:

"For quantifying human and climatic contributions to observed hydrological changes, a comprehensive modelling approach that takes into account, for example, the impacts of changing temperatures and land use change (e.g., urbanization and cropland expansion) on runoff generation and thus river inflow and on evaporation of the lake itself is preferable to statistical analyses such as trend and correlation analysis."

**Referee #3:** 2.1. On the role of atmospheric drought

**Figure 2 and the last paragraph of page 2:** This figure and its associated text provide an incomplete overview of the lake dynamics. The decline of the lake water level started around the year 2000, which is way more abrupt than 2003 onwards.

**Response**: Unfortunately, the lake level data that was provided by Tourian et al. (2015) started in the middle of 2002. Therefore, we could not extend that back before 2000. We attempted to show the lake dynamic within our study period. However, we modified the text as follows:

"In the 1970s and 80s, the water level of Lake Urmia was approximately at 1,276 m above sea level and then increased to more than 1,278 m in 1995 due to a few wet years (Shadkam et al., 2016). Khazaei et al. (2019) identified the year 2000 as the change point of lake dynamics. The water level dropped to 1,274 m in 2003 because of the severe drought in 1999-2001 exacerbated by human water use (Shadkam et al., 2016). From 2003 to 2014, lake extent was approximately halved, and water level declined by another 3 m, while seasonal variability of lake water extent increased (Tourian et al., 2015)."

**Referee #3:** Page 1 line 30: "The study shows that even without human water use Lake Urmia would not have recovered from the significant loss of lake water volume caused by the drought year 2008." First, you have not provided any evidence that the drought year 2008 caused a significant loss in lake volume, this causal link is nonexistent in your study. The authors are trying to over-emphasize the role of atmospheric droughts, specially the 2008 one. There has been stronger atmospheric droughts in previous years than year 2008. Here is a figure from Alborzi et al. (2018). The historic droughts during 80s and early 90s are more severe than the 2008 drought, yet the lake has survived (AghaKouchak et al., 2015).

[Figure]

**Figure 2.** Key attributes of the lake-basin system prior to restoration program in 2013, including observed lake level, standardized precipitation index (SPI), basin-scale naturalized runoff, and surface water withdrawal. The basin's recent wet (blue) and dry (red) periods are illustrated in SPI and naturalized runoff curves. Post-1998 drop in lake level corresponds to a substantial increase (~25%) in surface water withdrawals during the prolonged drought of 1998–2002.

**Response**: Firstly, it should be noted that the significant loss in 2008 (in comparison with other years of the study period) was shown in Figure 8. About the rest of your comment, we see some issues as follows:

- According to the figure you provided, the droughts during the 80s and early 90s also reduced the lake level, but the following wet period not only compensated the drought impact but also reached the lake level to the maximum level during the past 50 years.

- As you mentioned in previous comments, there is a sharp reduction in lake level after 2000. Based on the figure you provided, the drought during 1999-2002 could be the main reason for this reduction. It should be noted that it takes time to lake loss a $30 \cdot 10^9$ m$^3$ water that had in 1995. Therefore, we should not expect during the previous drought, and in a few years, the lake level falls below the ecological level.

- The past three droughts are not comparable due to the lake had a significant volume in two first droughts, so we believe that it is hardly acceptable to conclude that because the lake survived in the past droughts, then the lake should also be resistant to other droughts.

Based on the aforementioned discussion, let us disagree with your statement that due to this fact that lake survived from the 80s and early 90s so it could survive from the 2008 drought. Because the conditions of the lake in 2008 were completely different from previous droughts. Moreover, the conditions of the basin after 2008 were different, i.e., reducing the ratios of annual inflow into the lake over annual precipitation, as we discussed above.

**Referee #3:** 2.2. Irrigated area

One aspect that has not been discussed in section 4.2 is the irrigated area: how it is differently estimated by different studies and its implications. Below I have extracted figures corresponding to the estimated irrigated areas by 3 different studies. Supplement page 3 line 5: "Considering that water management in the basin aims at preventing any increase of irrigated areas,
5  it is assumed that the irrigated area in 2013 remained at the 2012 value (Fig. S3)". This assumption is questionable. For instance, Alizade Govarchin Ghale et al. (2019) estimated the irrigated lands to decrease by ~12% from 2012, but again increase in 2018-2019. Further, their estimated irrigated area is very different from the present study: year 2003 is different by ~500 km2 (~12%). The trend is also different, e.g. the increase during 2007-2011, or during 2003-2005. While both this study and Alizade Govarchin Ghale et al. (2019) estimated the irrigated area based on the overall vegetation coverage, Chaudhari et
10  al. (2018) made a distinction between natural vegetation and cropland. They showed (see the figures below) that the while the natural vegetation has oscillated throughout the years (1998 to 2006 → decreased, 2006 to 2011 → increased, and 2011 to 2016 → decreased), the cropland has continuously increased. Moreover, they estimated the annual net irrigation requirement (NIWR) during 1980-2010 based on the crop evapotranspiration (FAO Penman Monteith approach), independent of the global hydrological model they used, and compared it with estimations based on Landsat classification (see Figure 9 in their study).
15  So, their estimated irrigation is independent of how well or poorly their model was calibrated, and arguably more comprehensive than your study. While the present study demerited Chaudhari et al. (2018) (page 19 of the manuscript) and entirely overlooked Alizade Govarchin Ghale et al. (2019) and Fazel et al. (2017), the authors failed to acknowledge that these studies delved deep into land use changes, irrigation water requirement, and flow regime changes.

[Figure]

20                                        From the present study

[Figure]

From (Alizade Govarchin Ghale et al., 2019)

From Chaudhari et al. (2018)

**Response**: Firstly, we should note that our study did not demerit Chaudhari et al. (2018); we stated that the results of an un-
calibrated model were evaluated based on wrong in-situ data would not be reliable. Now we found another issue in Chaudhari
et al. (2018). They stated, "we use the Landsat imagery only for the month of September when the crops are completely
matured in the study region." Despite their statement, some crops were harvested that month. There are two peaks for crop
area in the basin 1) May for rainfed agriculture and 2) July and August for the irrigated area (Kamali and Youneszadeh Jalili
2015). As a result, their estimation of the irrigated area is significantly less than the actual one. For example, Kamali and
Youneszadeh Jalili (2015) estimated the irrigated area for 2003 about 4,500 km$^2$ and based on FAO (WaterGAP dataset), it
was about 5,000 km$^2$ while Chaudhari et al. (2018)'s estimation was about 2,000 km$^2$ which is less than half of actual one.
As you mentioned, the irrigated were used in Alizade Govarchin Ghale et al. (2019), and our study both were estimated based
on the overall vegetation coverage. However, Alizade Govarchin Ghale et al. (2019) used April and August to the estimated
irrigated area while Kamali and Youneszadeh Jalili (2015) used July and August that lead to some differences. Also, month
April that was used by Alizade Govarchin Ghale et al. (2019) includes both irrigated and rainfed farms, the distinction between
irrigated and rainfed cultivation may also make some differences. However, due to the fact, Kamali and Youneszadeh Jalili

(2015)'s report was approved by the ULRP, we believe that the use of the official report from ULRP would be more reliable than other sources.

About our assumption regarding the irrigated area in 2013, it should be noted all the works you mentioned here published after our manuscript submission. Therefore, we had no more reliable sources to use its data for 2013. The overall result may not change significantly by considering the small difference in the estimated irrigated areas in 2013.

We have added Fazel et al. (2017) to another part of our discussion based on your previous comments and added the following explanation to section 4.2:

"As a final word, the irrigated area used in this study obtained from the official report of ULRP (Kamali and Youneszadeh Jalili 2015). However, Chaudhari et al. (2018) estimated the irrigated area significantly less than the irrigated area used in the current study (Figure S3 compared to Figure 9 in their study). They used September for estimation of the irrigated area while the crops are completely matured in July and August in the basin. As a result, some crops are harvested in September. Therefore, it could be the main reason for such a significant underestimation of irrigated areas in the basin by Chaudhari et al. (2018). Also, Alizade Govarchin Ghale et al. (2019) estimated the irrigated area in the basin. Although their result is much closer to Kamali and Youneszadeh Jalili (2015) relative to Chaudhari et al. (2018), they used April and August to the estimated irrigated area, while Kamali and Youneszadeh Jalili (2015) used July and August that lead to some differences. Also, month April that was used by Alizade Govarchin Ghale et al. (2019) includes both irrigated and rainfed farms, the distinction between irrigated and rainfed cultivation may also make some differences. However, due to the fact, Kamali and Youneszadeh Jalili (2015)'s report was approved by the ULRP; we believe that the use of the official report from ULRP would be more reliable than other sources. However, the data reported by Kamali and Youneszadeh Jalili (2015) surly suffer some uncertainties that are inevitable."

**Referee #3:** 3 Comments on "Section 4.3 Limitations"

3.1. Study period

**Reviewer comment:** the time period 2003-2013 is inadequate for modeling the lake dynamics. Before 2000 the lake was not as heavily impacted by over-regulation of the river flows, and also between 2000-2003 there is significant variation in the lake level and annual inflows to the lake. Therefore, it is essential to include these years, for as many variable as possible. Otherwise, the model is biased and not representative of the lake dynamics.

**Authors' response:** We have considered this period due to the fact that the observed data was available for this period. We completely agree with you; it was better to consider a longer period for calibration. However, we don't prefer to reconstruct data, that is error-prone. The GRACE data and irrigated areas are not available for the period 2000-2003. Further, we don't want to use the model for out of calibration period, therefore we believe that for using the model in the calibration period there is no concern about the bias.

**Reviewer's response:** the point I argued is not simply about the length of data and model calibration. There are major aspects of the lake dynamics (and the basin evolution) that falls outside the 2003- 2013 period. While you evaluated your model within the calibration period using an independent variable, you tend to generalize your findings about the lake beyond the limited period of 2003-2013. To study the drivers of the lake drying, it is not adequate to build up your entire argument based on a

5    limited time period that does not include the non-stationarities of the lake system: various studies identified two periods of pre- and post-change for the lake, e.g. Khazaei et al. (2019) identified year 2000 and Fazel et al. (2017) identified year 2001 as the change point. Given that, your study does not cover the pre-change period, and both anthropogenic and natural scenarios are defined based on only a sub-period (2003-2013) of the post-change period (2000 to date), which biases the scenario analysis. Further, most recent changes in the lake system is also not discussed. The lake has experienced considerable changes

10   since 2013, e.g. see the extensive study by (Alizade Govarchin Ghale et al., 2019) on the land use changes within the lake basin. The figure below (extracted from Alizade Govarchin Ghale et al. (2019) shows the historic surface area as well as its increase since 2013– evidence of remarkable non-stationarity. To what extent your modelling assumptions and results are compatible with this non-stationarity, particularly the most recent changes of the lake?

[Figure]

Fig. 4. Salinization and desertification progress in Urmia Lake from 1975 to 2018.

**Response**: We did not generalize our findings for the lake, beyond the period of 2003-2013 nowhere in the manuscript. As we mentioned, the results of two other modelling studies, i.e., Farokhnia et al. (2018) and Shadkam et al. (2016) that covered the pre-change period, support our results. However, we did all analyses for the calibration period and no extending for after 2013 or before 2003. We also discussed the most recent condition of water storage in the basin as: "GRACE TWSA data indicate an increasing trend in water storage in the basin during 2014-2017 due to both less water use due to water management (ULRP, 2015b) and the wet years 2015/2016. This trend is about half as strong as the decreasing trend during 2003-2013."

It should be noted, WGHM as a distributed hydrological model is less dependent on the behavior of data in the calibration period than the data-driven models. We believe that as WGHM simulated the non-stationarity behavior of the lake during the calibration period, if there are high-quality input data for running the model out of calibration period, it can simulate the water storage components within the basin with reasonable accuracy; at least for the most recent changes of the lake that the changes fall in bounds of calibration period has to be reliable. To address your comment, we have added the following sentences to the limitations section:

"Finally, the study period 2003-2013 does not include some of the years with significant changes in the dynamics of the lake and the basin (i.e., years 2000 and 2001 that identified as the change point of the lake by Khazaei et al. (2019) and Fazel et al. (2017), respectively) due to data availability. Therefore, our results cannot be generalized to previous decades."

**Referee #3:** 3.2. Other limitations and suggestions

**Role of dams:** Another aspect of the lake system that you did not accounted for explicitly is the dam construction within the lake basin over the past decades (24 dams were constructed during 1970- 2000, and 32 during 2000-2014), which studies such as Fazel et al. (2017) and Alizade Govarchin Ghale et al. (2018) accounted for explicitly.

**Response**: We have added the following sentences to the limitations section:

"We determined that the results of the naturalized run with and without reservoirs for annual inflow into the lake differ by less than 2%, whereas Fazel et al. (2017) and Ghale et al. (2018) stated that dams have a significant impact on the lake shrinkage. However, Shadkam et al. (2016) showed the role of dams in the reduction of inflow into the lake did not exceed 5% due to evaporation from reservoirs. Moreover, in this study, the inflow into the lake was assessed on an annual scale, and there is no correlation between the dams' operation and annual inflow in the basin (Fathian et al. 2014). Therefore, the error from this source to our result should be negligible."

**Referee #3:** **Seasonal flow and model calibration:** While all variables are calibrated/evaluated on a monthly basis, the streamflow is calibrated on annual scale. I suspect that it is due to the fact that the model could not adequately represent the seasonal variations in streamflow, which are significant for this basin (Alizade Govarchin Ghale et al., 2019; Fazel et al., 2017). The seasonal variations of the flow have direct implications on irrigation estimations and the lake dynamics.

**Response**: The reason was that the reliable monthly data for inflow into the lake was not available, and the ULRP's annual data was calculated based on monthly data in stations around the lake with filling several gaps. However, to consider your

concern about the seasonality of inflow into the lake, we have plotted the seasonality of simulated inflow here. As you can see, the seasonality is almost similar to the seasonality presented by Fazel et al. (2017).

[Figure]

**Figure R: Seasonality of simulated inflow into the lake**

**Referee #3: Groundwater withdrawal and its hydrological connectivity to the lake:** The groundwater withdrawal is under-estimated in the model setup (a point that has been raised by reviewers before). While authors stated that "Observed decline of groundwater storage was $1.8 \cdot 10^9$ m3, i.e. 18% of the observed total water storage loss in the basin" (page 17 line 20), the groundwater withdrawal (including both shallow and deep wells, see the figure) shows at least 2.1 MCM withdrawal in the

10   past 2 decades.

[Figure]

**From LURP report (in Persian, attached to this review).** Red is withdrawal from deep wells, and green is withdrawal from partial deep wells, and blue is the total extraction.

**Response**: Our sentence is about groundwater storage loss, not groundwater withdrawal. It should be noted some parts of withdrawal from surface water and groundwater would retune to the groundwater. The loss is the withdrawal amount minus returned amount. Therefore, it is reasonable that the withdrawal volume is more than the loss. Hence, it sounds not only there is no problem here, but also your provided reference approved the validity of our results.

**Referee #3:** Also, as discussed by Danesh-Yazdi and Ataie-Ashtiani (2019) the hydrologic connectivity between the lake and groundwater remains an under-studied aspect of the lake dynamics – which is a general limitation of most studies including the present one.

**Response**: To address your comment, we added the following sentences to the limitations section:

10    "Also, in this study, it is assumed that there is no significant direct relationship between the lake and groundwater. However, the hydrologic connectivity between the lake and groundwater remains an under-studied aspect of the lake dynamics (Danesh-Yazdi and Ataie-Ashtiani 2019)."

**Referee #3:** Page 20 lines 1-3: re lake bathymetry please also cite the below studies:

15    • Sima, S., & Tajrishy, M. (2013). Using satellite data to extract volume–area–elevation relationships for Urmia Lake, Iran. *Journal of Great Lakes Research*, *39*(1), 90-99.

• Karimi, N., Bagheri, M. H., Hooshyaripor, F., Farokhnia, A., & Sheshangosht, S. (2016). Deriving and evaluating bathymetry maps and stage curves for shallow lakes using remote sensing data. *Water Resources Management*, *30*(14), 5003-5020.

**Response**: Both references were added.

**Referee #3:** 4 Minor comments

Page 3 line 10: "Studies on various aspects of the Lake Urmia disaster abound. With decreasing lake water volume, salt concentration has increased". Please cite the recent study on salt concentration as a dust source:

• Boroughani, M., Hashemi, H., Hosseini, S. H., Pourhashemi, S., & Berndtsson, R. (2019). Desiccating Lake Urmia: A New

25    Dust Source of Regional Importance. IEEE Geoscience and Remote Sensing Letters.

**Response**: The reference was added.

**Referee #3:** Page 3 line 11: "Precipitation reduction, temperature increase, agricultural development including construction of man-made dams and building a causeway across the lake have been identified as the reasons for the degradation of Lake Urmia

30    (Abbaspour and Nazaridoust, 2007; Zeinoddini et al., 2009; Delju et al., 2012; Jalili et al., 2012; Sima and Tajrishy, 2013; Fathian et al., 2014; Farajzadeh et al., 2014; Banihabib et al., 2015; AghaKouchak et al., 2015; Azarnivand and Banihabib 2017; Alizadeh- Choobari et al., 2016; Ghale et al., 2018; Khazaei et al., 2019)". Please separate out the references and cite relevant references for each factor (underlined phrases) individually. It helps the readers to track back.

**Response**: It has been modified as follows:

"Precipitation reduction and temperature increase (Delju et al., 2012; Fathian et al., 2014; shadkam et al. 2016; Farokhnia et al. 2018), agricultural development including construction of man-made dams (Farajzadeh et al., 2014; Banihabib et al., 2015; Azarnivand and Banihabib 2017; AghaKouchak et al., 2015; Ghale et al., 2018; Khazaei et al., 2019) and building a causeway across the lake (Zeinoddini et al., 2009) have been identified as the reasons for the degradation of Lake Urmia."

**Referee #3:** Page 4 line 25: "a good fit of simulated and observed streamflow may not necessarily lead to an appropriate simulation of other flows and storages (Beven and Freer, 2001). Therefore, additional types of observations have to be added to avoid equifinality (Beven and Freer, 2001; Döll et al., 2016)." The second sentence does not follow the first sentence, and using "therefore" does not make sense here. Also, by adding further data types, one will not "avoid" equifinality, because equifinality is a general property of open complex systems (e.g. hydrological models) and cannot be avoided. The goal is to "reduce" equifinality when possible. Please also cite the following recent studies on equifinality which are directly relevant to the discussion:

• Kelleher, C., McGlynn, B., & Wagener, T. (2017). Characterizing and reducing equifinality by constraining a distributed catchment model with regional signatures, local observations, and process understanding. Hydrology and Earth System Sciences, 21(7), 3325.

• Khatami, S., Peel, M. C., Peterson, T. J., & Western, A. W. (2019). Equifinality and flux mapping: A new approach to model evaluation and process representation under uncertainty. Water Resources Research, 55, 8922– 8941.

**Response**: Both references were added. The mentioned part has been modified as follows:

"a good fit of simulated and observed streamflow may not necessarily lead to an appropriate simulation of other flows and storages (Beven and Freer, 2001). Moreover, additional types of observations have to be added to reduce the possibility of equifinality (Döll et al., 2016; Kelleher et al. 2017; Khatami et al. 2019)."

**Referee #3:** Page 12 line 17 "We determined that the results of the naturalized run differ by less than 2% from a run with reservoirs but without human water use". First, it is not clear 2% of what is discussed here exactly. Second, such a small difference between the two scenarios is clearly a red flag, indicating that the model setup and/or scenarios are problematic. Most of the recent studies concluded that the lake condition is heavily impacted by human water use.

**Response**: In this sentence, we are speaking about two scenarios of the naturalized run, not anthropogenic one. Therefore, there is no concern about the model setup. We stated that the difference between results annual inflow into the lake is less than 2% in naturalized run with and without reservoirs (there is no human water use in both scenarios). This value for Q and for a period longer than our study period estimated less than 5% by Shadkam et al. (2016). The mentioned part has been modified as follows:

"We determined that the results of the naturalized run for annual inflow into the lake differ by less than 2% from a run with reservoirs but without human water use."

**Referee #3:** Page 14 line 20 "In this way, efficient simulation of regional water flows and storages can be achieved, possibly as an alternative to a costlier setup of a regional model". I'm not sure if I understood this part. What is costly about a regional model that is discouraging? What do you exactly mean by "setup a regional model", do you mean to develop a model from scratch?

**Response**: We stated in its previous sentence, "a global hydrological model includes all data for simulating water flows and storages in specific regions of interest everywhere on the globe, and model calibration against multiple (regional) observations is a means for improving the performance of the global model regionally." Hence, a global model with a regional calibration is much preferable than setup a regional model. Set up a regional model takes more time and needs more labor cost compared with the use of an available model. "setup a regional model" could be "develop a model and its dataset from scratch" or "develop a dataset for an available hydrological model".

**Referee #3:** Page 18, reword the title of the subsection 4.2 "Comparison to human vs. climatic contribution as determined in previous studies", it does not read well.

**Response**: It has been modified as follows:

"Distinguishing the contributions of human water use and climate variability to lake shrinkage"

**Referee #3:** Page 19 line 8: "*respectivly*" → respectively

**Response**: It has been corrected.
* * *
Thank you very much again for your time and for providing valuable comments.

**References**

[revised manuscript text omitted]